# Epigenetic homogeneity in histone methylation underlies sperm programming for embryonic transcription

Mami Oikawa[1,2,9], Angela Simeone[1,2,9], Eva Hormanseder[1,2], Marta Teperek [1,2], Vincent Gaggioli[1,2], Alan O'Doherty [3], Emma Falk [4], Matthieu Sporniak [4], Clive D'Santos[5], Valar Nila Roamio Franklin[5], Kamal Kishore[5], Charles R. Bradshaw [1,2], Declan Keane[6], Thomas Freour [7], Laurent David [4], Adrian T. Grzybowski[8], Alexander J. Ruthenburg [8], John Gurdon[1,2] & Jerome Jullien[1,2,4 ✉]

Sperm contributes genetic and epigenetic information to the embryo to efficiently support development. However, the mechanism underlying such developmental competence remains elusive. Here, we investigated whether all sperm cells have a common epigenetic configuration that primes transcriptional program for embryonic development. Using calibrated ChIP-seq, we show that remodelling of histones during spermiogenesis results in the retention of methylated histone H3 at the same genomic location in most sperm cell. This homogeneously methylated fraction of histone H3 in the sperm genome is maintained during early embryonic replication. Such methylated histone fraction resisting post-fertilisation reprogramming marks developmental genes whose expression is perturbed upon experimental reduction of histone methylation. A similar homogeneously methylated histone H3 fraction is detected in human sperm. Altogether, we uncover a conserved mechanism of paternal epigenetic information transmission to the embryo through the homogeneous retention of methylated histone in a sperm cells population.

[1] Wellcome Trust/Cancer Research UK Gurdon Institute, University of Cambridge, Tennis Court Road, Cambridge CB2 1QN, UK. [2] Department of Zoology, University of Cambridge, Downing Street, Cambridge CB2 3EJ, UK. [3] UCD School of Agriculture and Food Science, University College Dublin, Dublin 4 D04 V1W8, Ireland. [4] CRTI, INSERM, UNIV Nantes, Nantes, France. [5] Cancer Research UK Cambridge Institute, University of Cambridge, Robinson Way, Cambridge CB2 0RE, UK. [6] ReproMed Ireland, Rockfield Medical Campus, Northblock, Dundrum, Dublin 16 D16 W7W3, Ireland. [7] Service de Biologie de la Reproduction, CHU Nantes, Nantes, France. [8] Department of Molecular Genetics and Cell Biology, The University of Chicago, 920 East 58th Street, Chicago, IL 60637, USA. [9] These authors contributed equally: Mami Oikawa, Angela Simeone. ✉email: jerome.jullien@inserm.fr

Fertilisation of eggs by sperm produces embryos with much higher efficiency than embryos generated by other methods, such as nuclear transfer[1]. What is the basis for such a high developmental potential of sperm? Previously we and others have shown that spermatids, the immediate precursors of sperm, with the same haploid genetic content but with a different chromatin structure, have a reduced ability to support embryonic development[2–4]. Hence, beside the delivery of the paternal genetic material, sperm also deliver epigenetic cues that are necessary for embryonic development. Such epigenetic features from the sperm have been proposed to participate in normal embryonic development through the regulation of gene expression in early embryos. This hypothesis is based on the observation that in mammals, zebrafish and frogs, developmentally important genes are marked by modified histones in sperm, a feature that correlates with their expression in the early embryos[5–8]. Importantly, global interference with sperm histone methylation during either spermiogenesis or at fertilisation appears to alter embryonic gene expression and development[2,9]. This suggests that modified histones in sperm chromatin are required for embryonic development.

In order to understand the importance of the epigenetic contribution of sperm to embryos, it is crucial to evaluate the homogeneity of chromatin composition between single-sperm cells. Indeed, intra cytoplasmic sperm injection to eggs demonstrated that almost every sperm cell, regardless of its ability to fertilise an egg, is competent to support development[10]. This implies that sperm-derived epigenetic features required for proper embryonic gene regulation must be present in every sperm cell of a population (i.e., homogeneous). The question of homogeneity in sperm chromatin composition is particularly important with regards to modified histones. Indeed, spermiogenesis is associated with a massive rearrangement of chromatin that entails loss of histones and deposition of protamines[11]. In mammals in particular, only 0.3–10% of histones found in somatic cells are retained in mature sperm[5,12,13]. Because of this loss, it has not been possible to clearly distinguish whether some histones are always retained at the same position in the genome of most sperm cells and hence could transmit epigenetic information, or if they are randomly distributed and are therefore unlikely to carry important epigenetic information[14,15]. Addressing this question of epigenetic homogeneity in sperm is therefore crucial, as it will establish whether modified histones have the required attributes to participate in a faithful transmission of epigenetic information from the sperm to the embryo.

In *Xenopus laevis*, the deposition of protamines during spermiogenesis is associated with only a partial loss of some of the core histones[16,17]. This feature places this vertebrate in an intermediate position between the situation found in mammals, where the majority of histones are replaced, and zebrafish in which sperm do not lose any histones[5,7]. Taking advantage of this unique feature of *Xenopus laevis* spermiogenesis in combination with a recently developed quantitative ChIP technology[18], we are able to provide a detailed analysis of histone and modified histone distribution on chromatin in a sperm population. We show that during spermiogenesis, the programming of sperm genes for embryonic development is associated with the formation of chromatin regions of homogeneous epigenetic constitution within a sperm population. Finally, we provide evidence of homogeneous histone methylation in human sperm, suggesting a conservation of sperm epigenetic programming mechanisms between vertebrates.

## Results

### Packaging of *Xenopus* sperm chromatin by histones. Sperm core histones and their associated post-translational modifications are

potential carriers of epigenetic information instructive for orchestrating embryonic gene expression. A prerequisite for such core histone involvement in the epigenetic programming of sperm for development is their presence at the same genomic location in most sperm cells. We therefore evaluated how and where core histones package the chromatin in sperm. We first analysed sperm chromatin core histone composition. Using quantitative western blotting, we found that the content of histone H3 and H4 in *X. laevis* sperm is comparable to that of somatic cell, whereas that of histone H2A and H2B decreases by ~60–70% (Fig. 1a) as previously reported[17]. This raises the question of how core histones are associated with sperm DNA in *X. laevis*. Indeed, the stoichiometry of core histones retained in frog sperm implies that a large fraction of histone H3/H4 cannot be associated with DNA as nucleosomes. To reveal how DNA is packaged with the core histones retained in frog sperm, we performed a micrococcal nuclease (MNase) digestion assay, as described before[2].

Similar to what is observed in somatic cells, MNase treatment of sperm chromatin generated 150 bp fragments corresponding to nucleosomes (Fig. 1b). Two additional DNA fragments with a size of ~70 and ~110 bp appear specifically after digestion of sperm chromatin. Such shorter DNA fragments could potentially correspond to MNase getting access to the DNA wrapped around nucleosomes and leading to an internal cut by the nuclease (Fig. 1c). Alternatively, shorter fragments could arise from protection of the DNA by different nucleoprotein complexes altogether. To distinguish between these two possibilities, we used sucrose gradient centrifugation to separate DNA-protein complexes (hereafter named particles) according to their size. The 70, 110, and 150 bp DNA fragments are recovered at gradually lower position on the gradient, clearly indicating that these DNA fragments are present in nucleoprotein complex of increasing size (Fig. 1d). To elucidate the composition of these particles we performed quantitative mass spectrometry (tandem mass tag, TMT) on proteins collected from the sucrose gradient fractions containing the 150 or 70 bp DNA fragments, in four biological replicates. The 110 bp particles were excluded from the analysis as we could not isolate a fraction containing only these types of DNA fragments. Using this approach, we identified 840 proteins associated with the purified chromatin particles (Supplementary Data 1). Relative abundance of proteins associated with the 150 bp versus 70b bp DNA was evaluated after normalisation to histone H4 signal. Both DNA complexes show similar association with core histones H3, H3.3 as well as linker histones and HMGN1-3 proteins ($|logFC\ (150/70)| < 1.5$ and FDR > 0.05). Interestingly, compared with the 70 bp DNA, the 150 bp DNA is enriched for all known variants of core histone H2A and H2B ($logFC\ (150/70) > 1.5$ and FDR < 0.05). By contrast, the 70 bp particles show increased association with testis-specific histone H1FX[19] and protamine (sperm basic protein 5[20]) as well as numerous proteins involved in chromatin structure such as CBX3 (also known as HP1gamma), HMGB1-3, HMGA2, WDR5 and WDR16 ($logFC\ (150/70) < -1.5$ and FDR < 0.05). We confirmed by western blot analysis that the 70 bp fragments contain a lower ratio of H2A and H2B to H3 and H4 as that found in the nucleosome particles (150 bp fragments), as well as a higher ratio of HMGB1 to H4, and a similar ratio of HMGN2 to H4 (Fig. 1e and Supplementary Fig. 1). These data show that the 110 and 70 bp fragments do not correspond to nucleosomes formed of core histone octamers, but rather correspond to DNA fragments protected from MNase digestion by core histone complexes depleted of H2A and H2B possibly as $(H3/H4)_2$ tetramers (70 bp) or $(H3/H4)_2(H2A/H2B)$ hexamers (110 bp) (Fig. 1f and refs. [21,22]).

From these analyses, we conclude that *Xenopus laevis* sperm DNA has a similar density of histone H3 and H4 as a somatic cell. However, in sperm, these two core histones are packaging DNA

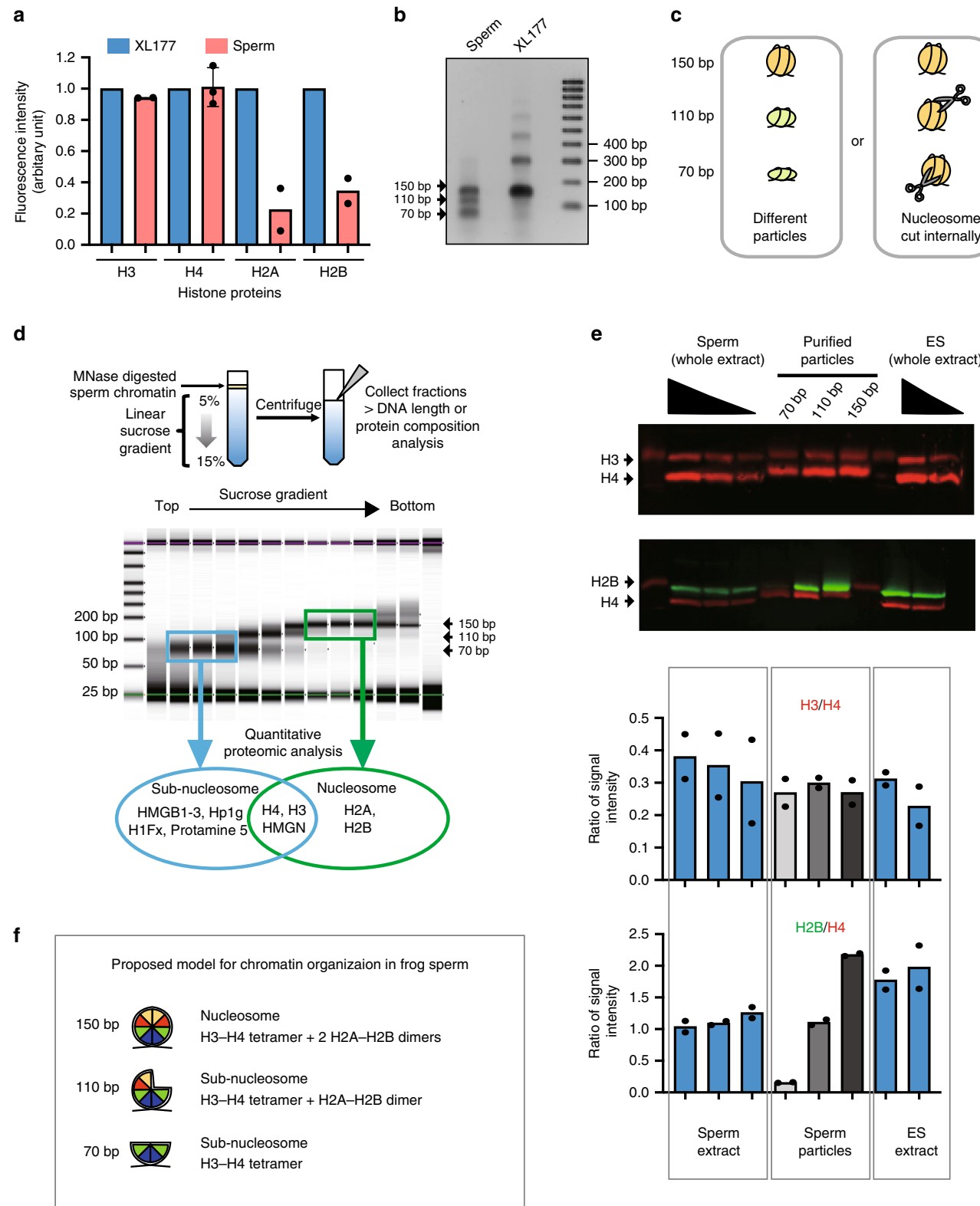

**Fig. 1 Somatic level of histone H3/H4 is retained as nuclesosomes and subnucleosomes in sperm chromatin. a** *Xenopus laevis* sperm core histones content relative to that found in a somatic cell (XL-177) as measured by quantitative WB (H2A, H2B, H3 $n = 2$, H4 $n = 3$, biologically independent samples error bar on H4 shows standard deviation). **b** DNA fragments generated by MNase digestion of *Xenopus laevis* sperm and somatic cell. **c** Schematic representation of the possible origin of subnucleosomal sized fragments generated by MNase treatment of sperm chromatin. **d** Nucleoproteic particles generated by MNase treatment of sperm are centrifuged on a sucrose gradient. Subsequently, particles isolated along the gradient are analysed for associated DNA fragment length (electrophoresis) and for associated proteins (mass spectrometry). **e** WB analysis confirms mass spectrometry analysis. Similar ratio of H3 to H4, and decreased level H2B to H4 are detected in subnucleosomes compared with nucleosomes. *Xenopus* sperm and mESCs are shown as control. Graphs below show the quantification of WB data ($n = 2$, biologically independent samples). **f** Model of core histone composition of *Xenopus laevis* sperm nucleosomal and subnucleosomal particle. Source data related to **a**, **b**, **d** and **e** are provided as Source Data files.

either as nucleosomes or as protamine-associated subnucleosomes. We next sought to estimate how the different histone H3-containing particles are distributed in the sperm genome.

**Homogeneous H3 packaging of gene regulatory regions**. To identify genomic sites where nucleosome retention or remodelling events are occurring in most sperm cells, we used MNase-seq. Indeed, because of the somatic cell-like H3 occupancy on the frog sperm genome, MNase-seq can be used to identify genomic sites where nucleosome retention or remodelling events are occurring in most sperm cells, an analysis that cannot be achieved in species where an extensive loss of core histone occurs[14]. Therefore, to reveal how the three types of H3/H4-containing particles are distributed along sperm DNA, we performed paired-end sequencing of DNA fragments generated by MNase digestion of sperm chromatin (~7 × genome coverage, Supplementary Fig. 2). We observed that ~50%, ~14% and ~36% of mapped reads correspond to the 150, 110 and 70 bp length DNA fragments, respectively, in good agreement with the relative intensity observed by electrophoresis for these fragments (Figs. 1b and 2a). When taking into account the length of DNA protected by each type of particles, this indicates that the majority of the sperm genome (66%) is protected by nucleosomes, whereas 21% and 13% is in complex with the 70 and 110 bp particles, respectively. In order to assess if spermiogenesis leads to a non-random distribution of these particles on the sperm chromatin, we applied a hidden Markov model to identify genomic regions enriched for a particular particle type (see 'Methods' for details). We observe that ~72% of the genome is heterogeneous with no enrichment for any type of particle, indicating that at these genomic locations different sperm cells retain different types of particles. By contrast, the remaining ~28% of the genome is homogeneous with regard to particle enrichment (Fig. 2b). Interestingly, the remodelling of nucleosomes into subnucleosomes is globally as frequent as nucleosome retention (Fig. 2a, left), but happens less heterogeneously (19% (17 + 2) versus 9%, Fig. 2b). Furthermore, genomic regions showing homogeneous nucleosome retention in sperm are enriched for gene regulatory regions (promoter and enhancer), supporting a possible role for sperm-derived nucleosomes in embryonic gene regulation (Fig. 2c).

To better understand how homogeneous particle composition relates to gene regulatory regions, we performed gene clustering analysis according to the extent to which the region surrounding their TSS +/−2 kb (thereafter named promoter) retains nucleosomes and/or subnucleosomes. We observe that the majority of promoters have a heterogeneous particle composition in a sperm population (Fig. 2d, cluster 6, ~59% of genes). However, in the remaining 41% of gene promoters, the particle composition is homogeneous in a sperm population with ~7% of genes retaining nucleosomes (cluster 5) and ~30% retaining subnucleosomes (cluster 2, 3 and 4). Large domain spanning most of +/−2 kb intervals surrounding the TSS appear to be homogeneous for nucleosome retention (cluster 5) or remodelling (cluster 2). Interestingly, an enrichment for function related to spermatogenesis is observed for cluster 2 where the highest level of nucleosome to subnucleosome remodelling is observed (Supplementary Data 2). Here, nucleosome remodelling of spermatogenesis-related genes regulatory regions might contribute to the programming of sperm for development by resetting chromatin structure on genes transcribed during the previous developmental phase.

We conclude from these analyses that spermiogenesis is associated with homogeneous retention or remodelling of nucleosomes on gene regulatory regions in a sperm population (Fig. 2e).

**H3K4 and H3K27 are always methylated on a sperm genes subset**. Our results so far indicate that some sperm histone H3-containing particles have the required attributes to act as carrier of epigenetic information to the next generation as they are found to be retained at the same genomic location within a sperm cell population. To further characterise histone H3-containing particles in sperm chromatin, we investigated H3 trimethylation on lysine 4 and lysine 27, two well-studied modifications associated with an active and repressed state of gene expression, respectively. We used internal standard calibrated chromatin immunoprecipitation sequencing (ICeChIP-seq[18]) to estimate the percentage of histone H3 methylation at a given locus in a sperm population (apparent histone methylation density, HMD) (Supplementary Fig. 3 and 'Methods'). We observed that most of the genomic loci with methylated histones have low levels of methylation (Fig. 3a, 0% < HMD < 80%). At these genomic sites histone methylation is therefore heterogeneous within the sperm population. However, ~0.4% and 6% of the sperm genome show an HMD > 80% for H3K4 and H3K27, respectively. These sites have histone methylation in most sperm and cover a fraction of the genome similar to that found in an ESC population[18], suggesting that modified histones might have functional relevance in sperm as is the case for ESC (Fig. 3a). When focusing on peaks of histone marks, we observed that 20% and 70% of peaks have an HMD > 80% for H3K4 and H3K27, respectively (Fig. 3b). Given the high level of retained histone H3 in *Xenopus* sperm chromatin (Fig. 1a), we conclude that, at these genomic sites, most sperm harbour a methylated H3. Such genomic sites are enriched for gene regulatory regions (TSSs and enhancers (Fig. 3c)), and appear enriched for binding motifs recognised by transcription factors implicated in early embryonic development (i.e., NFY-A/Dux[23]; Ascl1[24], ZFP281[25]; Fig. 3d). High methylation density for H3K4 is localised in the immediate vicinity of the TSSs whereas high level of H3K27 methylation is observed on most of the +/−2 kb interval surrounding the TSSs (Fig. 3e). Several gene clusters show co-occurrence of high degree of methylation density for both H3K4 and H3K27 (clusters 2, 4 and 5) and are associated with GO categories related to development, especially when H3K4 methylation cover a broader domain around the TSS (cluster 4) (Supplementary Data 3).

To conclude, specific regulatory regions near developmental genes are homogeneously methylated on H3K4 and/or H3K27 in most sperm of a population.

**Homogeneous bivalent H3 marking on sperm developmental genes**. So far, we have identified parts of the sperm genome that are homogeneous in the way histone H3 package chromatin (as nucleosomes or subnucleosomes, Fig. 2) and those that are homogeneous for the methylation of histone H3 (on Lys4 or Lys27, Fig. 3). We next investigated if homogeneity for these two types of histone H3 epigenetic features occurs at the same genomic location.

We first ask if the chromatin sites with homogeneous histone methylations (peaks of H3K4 or H3K27 with HMD > 80) are occurring at locations that always retain H3 within a nucleosome, always remodel H3 into subnucleosomes or have either of these particles in a sperm population (Fig. 4a, b). Interestingly, we observe an enrichment for homogeneous histone H3 methylation at the genomic locations where nucleosomes are always retained, and to a lesser extend at genomic locations where nucleosomes are always remodelled into subnucleosomes. In particular there is a tendency, albeit weak, for homogeneous retention of H3K4me3 in the context of a nucleosome.

As they could represent instructions for future embryonic gene expression we further characterised chromatin sites homogeneous

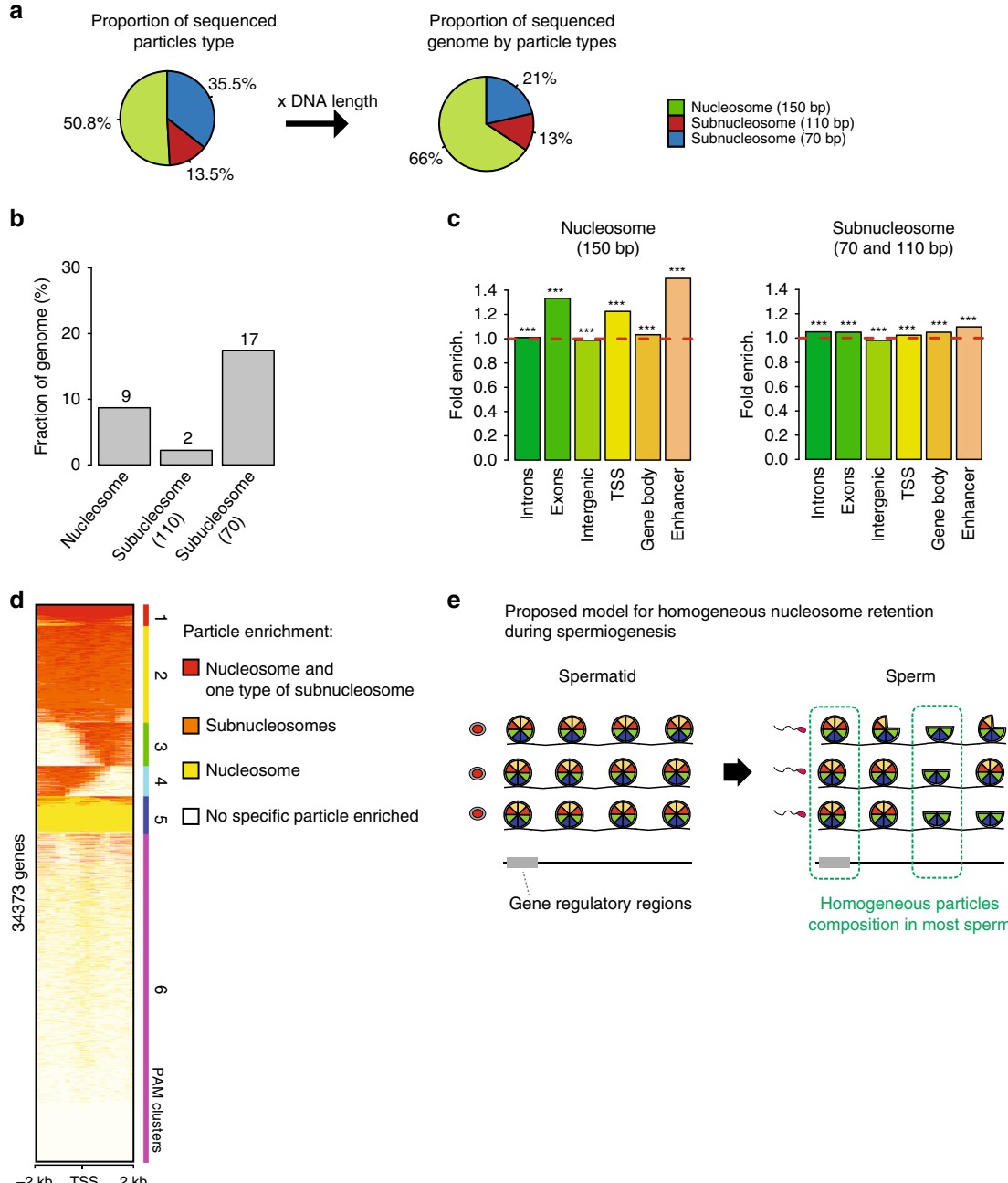

**Fig. 2 Nucleosome loss/retention associated with spermiogenesis occurs non-randomly in a large fraction of the genome. a** Relative abundance (left) and genome coverage (right) of nucleosomes and subnucleosomes in the sperm chromatin. The left pie chart reports the fraction of DNA fragments from *Xenopus Laevis* sperm corresponding to each type of particle; the right pie chart reports the fraction of genome covered by each type of particle. **b** Fraction of the genome with homogeneous particles composition. The bar graph indicates the percentage of the genome that possess nucleosomal or subnucleosomal structure across most sperm of the population sequenced (genome binned in 50 bp windows). **c** Fold enrichment (observed/random) over 1000 randomisations for homogeneous nucleosomes (left) or subnucleosomes (right) composition at the indicated genomic features; ***: empirical *p* value < 1e−3. Input data from two independent replicates were pooled. **d** PAM (partitioning around medoids) clustering of promoter (TSS +/−2 kb) according to enrichment for nucleosomes or subnucleosomes. **e** Model of nucleosomes and subnucleosomes distribution in sperm and spermatid. Source data related to **a**, **b** and **d** are provided as Source Data files.

for H3 particle type and for H3 methylation, focusing on gene regulatory regions (Fig. 4c and Supplementary Data 4). To that end, we clustered all gene TSSs according to homogeneity for the four H3 epigenetic parameters evaluated in this study (methylation of H3K4, methylation of H3K27, nucleosome retention and nucleosome remodelling). The +/−2 kb region around the TSS is divided in 200 bp bins. For each bin, the four epigenetic parameters are classified as either homogeneous or heterogeneous. Specifically, histone methylation is considered homogeneous if the bin has an HMD > 80%, and heterogeneous otherwise. In addition, a bin is classified as homogeneous for the presence of a nucleosome or for subnucleosome if an enrichment is detected (two-layered inference strategy, see M&M) and heterogeneous otherwise. In that way we obtain a

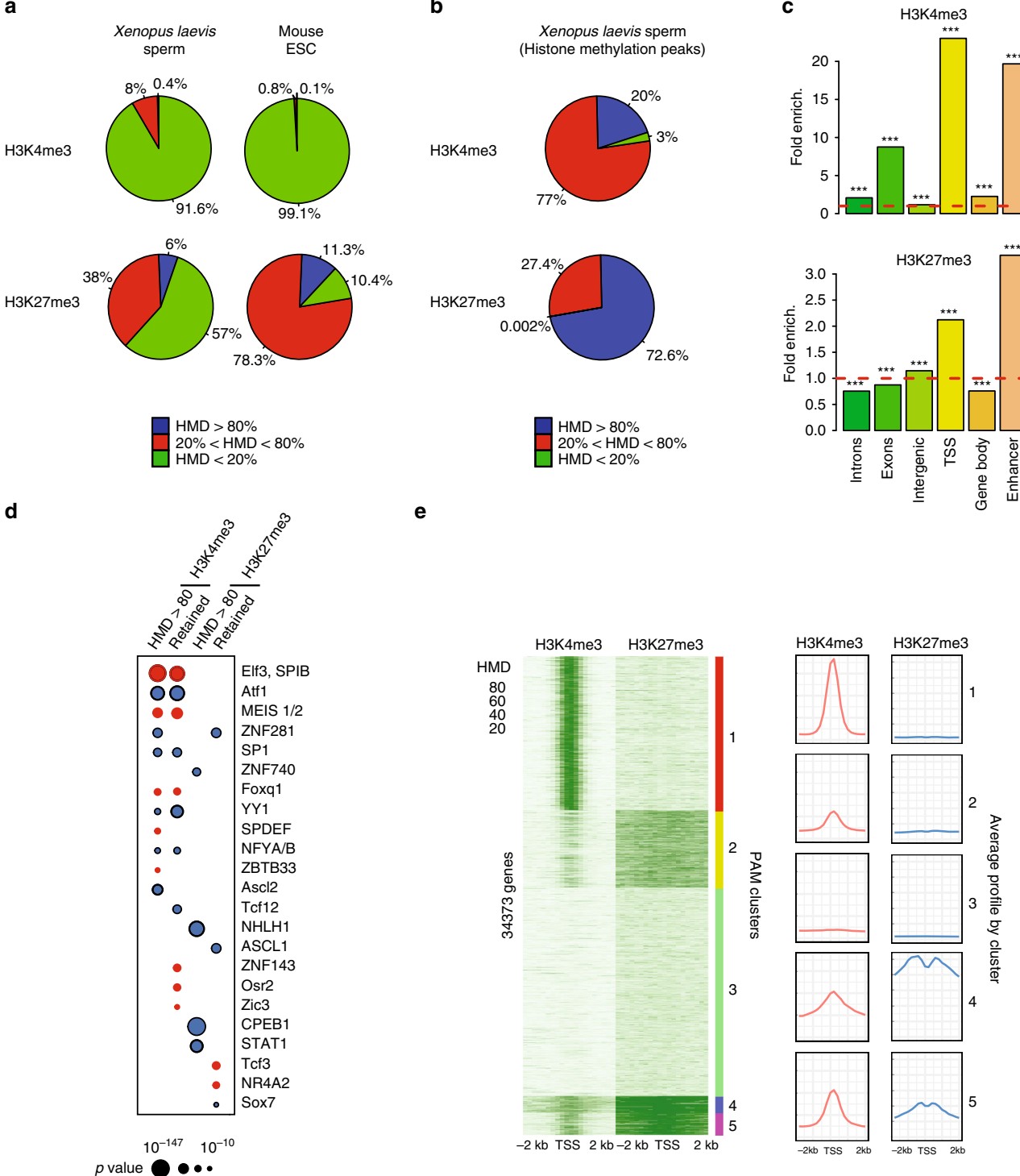

**Fig. 3 A fraction of the genome harbours methylated H3K4 and/or H3K27 at the same location in most sperm of a population. a** Percentage of the genome with different range of apparent histone H3 methylation density (HMD) on Lysine 4 and 27 in *Xenopus Laevis* sperm and mouse ESC. **b** Percentage of H3K4me3 and H3K27me3 peaks with different range of HMD in *Xenopus Laevis* sperm. **c** Fold enrichment (observed/random) over 1000 randomisations of peaks with homogeneous histone methylation (HMD > 80) at the indicated genomic features; ***: empirical *p* value < 1e−3. ICe-ChIP data from two independent replicates were pooled. **d** Dot matrix showing transcription factors with enriched motifs (*y*-axis) in the different histone methylation categories (*x*-axis). Circle size represents −log10 (*p* value) of the motif enrichment; and the circle colour indicates whether evidences exist indicating that the corresponding transcription factors is present maternally (blue) or not (red). Retained HMD > 80 peaks correspond to sperm histone methylation peaks maintained after extract treatment as in Fig. 5. **e** Heat map after PAM clustering of promoters (TSS +/−2 kb) according to histone H3-methylation density on Lysine 4 and 27. The plots on the left show the average HMD profile for each cluster. Source data related to **a**, **b**, **d**, and **e** are provided as Source Data files.

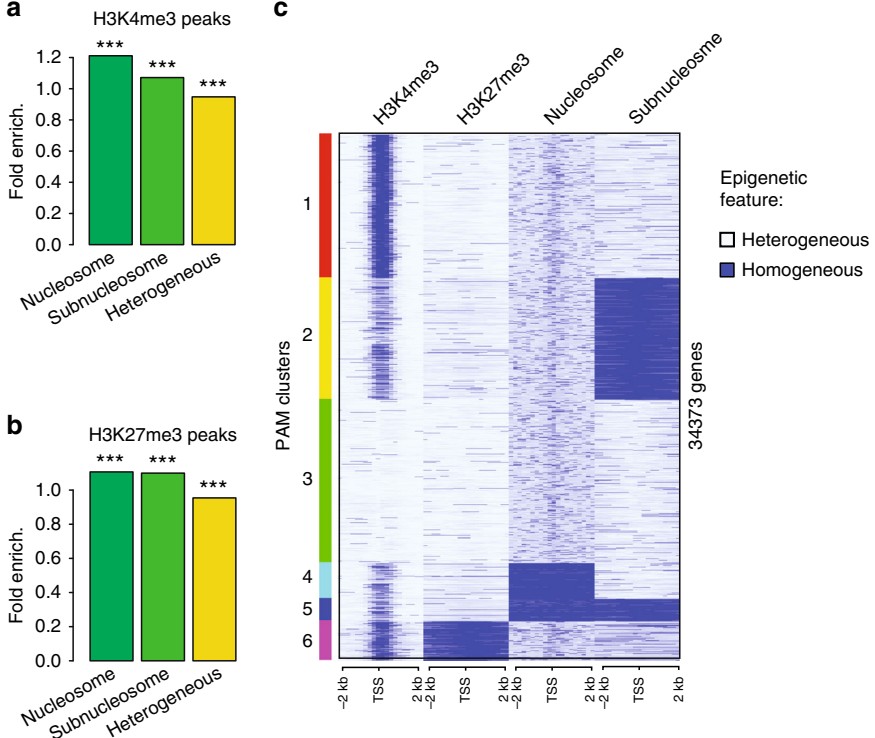

**Fig. 4 Homogeneous bivalent marking of histone H3 on sperm developmental genes. a** Fold enrichment (observed/random) over 1000 randomisations of peaks with homogeneous H3K4 methylation (HMD > 80) in regions homogeneous for nucleosomes, homogeneous for subnucleosomes or with heterogeneous particle composition in a sperm population; ***: empirical $p$ value < 1e−3. **b** Same as in **a** for H3K27 methylation. **c** PAM clustering of promoter (TSS +/−2 kb) according to homogeneity for methylation of histone H3 (HMD > 80) on Lysine 4, on Lysine 27, enrichment for nucleosomes and enrichment for subnucleosomes. Source data related to **c** is provided as a Source Data file. ICe-ChIP data from two independent replicates were pooled.

view of the sperm epigenetic landscape at TSS that reveals features conserved within the sperm population.

As expected, homogeneous H3K4me3 is mostly found in the vicinity of the TSSs where homogeneous nucleosome retention is also observed. However, this analysis also revealed that, as a whole, TSSs with the less heterogeneous histone methylation (H3K4me3 cluster 1, and H3K27me3 cluster 6) differ from TSSs characterised by homogenous block of nucleosome and/or subnucleosome (clusters 2, 4 and 5). In particular, genes with stretch of homogeneous nucleosome retention (cluster 4) or remodelling (cluster 2) surrounding the TSS are associated with reduced H3K4me3 density when compared with genes with heterogeneous particles composition (clusters 1 and 6). We also identified a set of genes with homogeneous methylation on both H3K27me3 and H3K4me3 (cluster 6). This set of genes show co-occurrence of K4 and K27 methylation in most sperm. These bivalent genes include many transcription factors involved in early embryogenesis (i.e., members of the Hox, Fox, Sox, Gata, Tbx and Pax transcription factor families). Moreover, globally, these bivalent genes are associated with GO terms related to development (Supplementary Data 4).

We conclude that, in general, homogeneity for histone methylation and histone particle composition is occurring on different group of genes. Importantly, this analysis reveals the existence of bivalent genes marking of developmental genes in all sperm.

**Sperm-methylated histones are maintained during DNA replication.** Because the epigenetic features investigated in this study are present at the same genomic location in most sperm cells, they could represent necessary information delivered by the

sperm at fertilisation to support embryonic development. Such necessary epigenetic cues would need to be transmitted to the cells of the developing embryo to exert their action. To test this hypothesis, we first incubated permeabilised sperm in egg extract to mimic the chromatin assembly and replication steps associated with the first embryonic cycle[26] (Fig. 5a and Supplementary Fig. 4). H3K4me3 and H3K27me3 ChIP-seq analysis indicated that 75% of sperm H3K4me3 peaks and 24% of H3K27me3 peaks are retained after egg extract treatment (Fig. 5b). In sperm, prior to treatment, these retained peaks had higher methylation density (i.e., less heterogeneous in the sperm population) and were larger in size than lost peaks (Fig. 5c, d). We also observed that retention of peaks is favoured over gene regulatory regions (Fig. 5e). However, the context within which high histone H3 methylation occurs (with or without enrichment for a given H3 particle) is not generally predictive of the fate of the methylated histone post replication (Supplementary Fig. 5). Finally, sperm incubation with extracts containing the DNA replication inhibitor geminin[27,28] suggests that most histone methylation peak loss following treatment is associated with chromatin assembly on sperm DNA rather than with DNA replication (Supplementary Fig. 6).

Focusing on the embryonically abundant H3K4me3, we evaluated the fate of sperm-methylated histones in vivo after several replication cycles. For that purpose, we performed ChIP-seq analysis using formaldehyde-fixed early blastula embryos (after ~8 embryonic cell cycles but before the activation of zygotic transcription) (Fig. 5a). We observed that almost all H3K4me3 peaks detected in blastulae were already present in the sperm chromatin (Fig. 5f). Similar to peak retention after egg extract treatment, we observed that H3K4me3 peaks retained in blastula

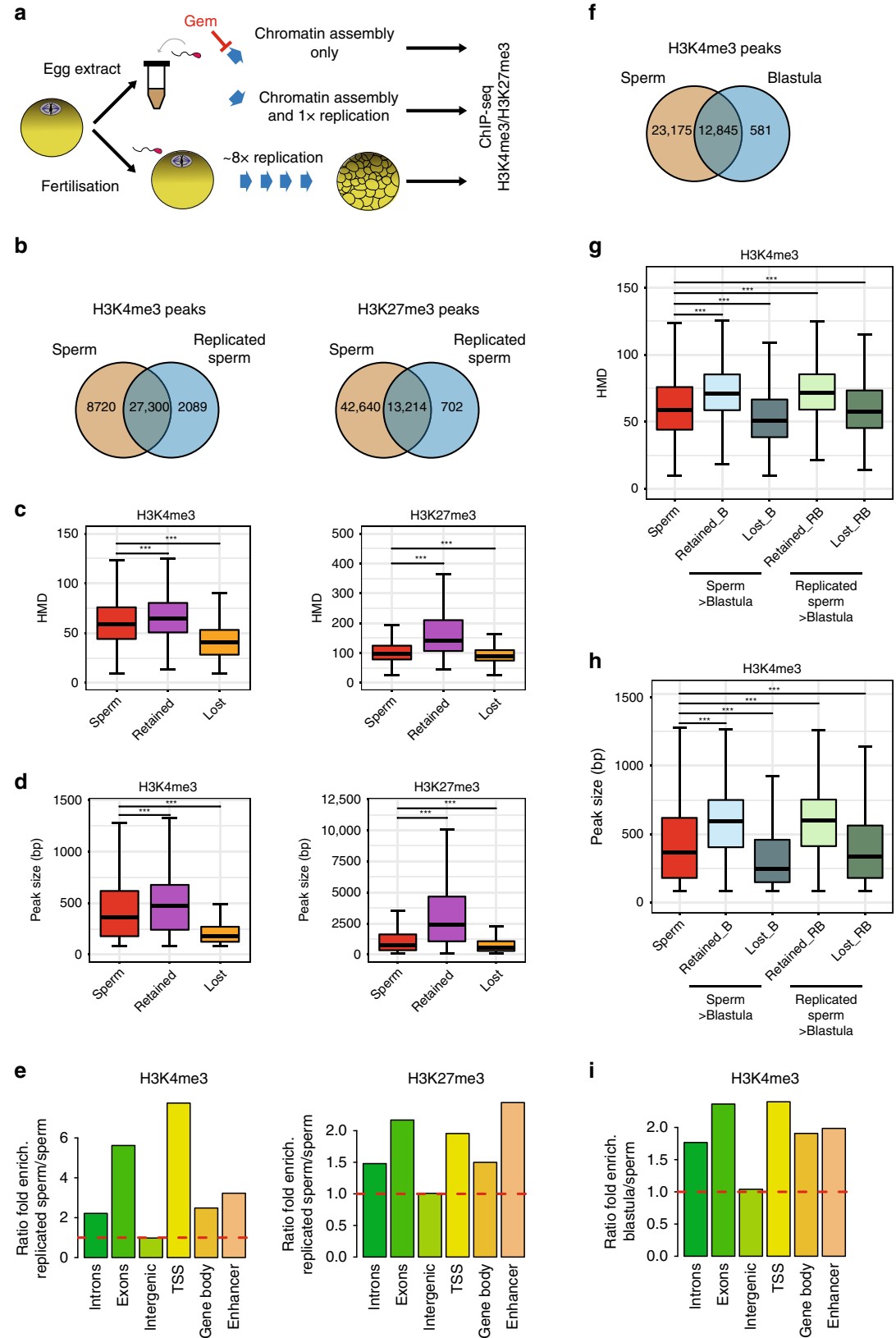

embryos corresponds to gene regulatory sites that harbour stretches of H3K4me3 in most sperm cells (Fig. 5g–i).

These observations show that the epigenetic information present at the same genomic location in most sperm cells can be faithfully transmitted to the mitotic progeny in early frog embryos.

**Sperm-methylated histone programmes embryonic gene expression.** We then thought to evaluate if the homogeneous epigenetic fraction of the sperm genome that is transmitted to the embryo contributes to the regulation of embryonic gene expression. We first compared early embryonic expression of genes with heterogenous methylation in sperm to that of genes that have a

**Fig. 5 Homogeneously methylated sperm histones are maintained during early embryonic replication. a** Experimental setup to monitor the fate of sperm-methylated histone peaks after replication. **b** Overlap between peaks of H3K4me3 and H3K27me3 before and after replication of sperm chromatin in egg extract. **c** Boxplots of HMD for all sperm peaks and for peaks that are lost or retained after replication. **d** Boxplots of the size of all sperm peaks, and of peaks that are lost or retained after replication. Data in **c** and **d** are obtained from N. sperm peaks H3K4me3: 36020; N. sperm retained H3K4me3: 27300; N sperm lost H3K4me3: 8715. N. sperm peaks H3K27me3: 55854; N. sperm retained H3K27me3: 13214; N sperm lost H3K27me3: 42635. ***p value < 1e−3 (two-sample Kolmogorov–Smirnov test). **e** Ratio of fold enrichment of peaks retained after replication over those lost after replication at indicated genomic features. Fold enrichments (observed/random) were obtained from 1000 randomisations and all instances showed an empirical p value < 1e−3. **f** Overlap between peaks of H3K4me3 in sperm and in blastula embryos. **g** Boxplots indicating HMD for all sperm peaks and for peaks that are lost or retained in blastula compared with sperm, and lost or retained in blastula compared with replicated sperm. **h** Boxplots of peak sizes for all sperm peaks and for peaks lost and retained in blastula compared with sperm, and in blastula compared with replicated sperm. In **g** and **h** ***p value < 1e−3 and are obtained by the two-sample Kolmogorov–Smirnov test. **i** Ratio of fold enrichment of peaks retained in blastula versus those in sperm at indicated genomic features. This ratio has been obtained as in **e**. HMD is from data pooled from two independent replicates. Peaks retention/lost are consensus from three independent replicates.

homogeneous methylation in sperm and that retain this methylation after replication (Supplementary Data 5). We find that the set of genes associated with homogenous methylation in sperm is enriched for genes expressed at zygotic gene activation (ZGA) (Fig. 6a). Specifically, homogeneous methylation of H3K27 in sperm is associated with ZGA genes involved in development whereas homogeneous methylation of H3K4 in sperm is associated with ZGA genes involved in housekeeping function. Importantly, genes with heterogeneous methylation in the sperm population do not show such enrichment for these ZGA gene categories. Therefore, the epigenetically homogeneous fraction of the sperm genome that is maintained after replication is strongly associated with early embryonic gene transcription.

To further test the idea that homogeneous marking in sperm is functionally linked to embryonic gene expression, we asked whether interference with H3K4 or H3K27 methylation at fertilisation, as reported in our previous work[2], preferentially affect genes with an homogeneous methylation in a sperm population. In that study we identified genes that were misregulated in gastrulae generated from egg expressing a demethylase targeting either H3K4 (Kdm5b) or H3K27 (Kdm6b) methylation prior to sperm injection. The set of genes that are misregulated upon demethylation of H3K27me3 (Kdm6b sensitive) is enriched for genes with homogeneous methylation of H3K27me3 in sperm (Fig. 6b). In addition, a significant proportion of genes sensitive to both demethylation of H3K27 and depletion for the maternally provided factor Ascl1 harbour homogeneously methylated H3K27 and the binding motif for this transcription factor (Supplementary Fig. 7). This suggests that sperm-provided modified histones might regulate maternal factor activity.

Unexpectedly, however, we observe that the set of genes misregulated upon H3K4me3 demethylation (Kdm5b sensitive) is also enriched for gene with homogeneous methylation of H3K27me3 (Fig. 6b). This could be explained if demethylation of sperm H3K4me3 affects subsequent embryonic gene expression only when it co-occurs with methylation of H3K27 (i.e., sperm bivalent genes). To test this hypothesis we first investigated the fate of the fraction of sperm genome homogeneously methylated on H3K4me3 only, H3K27me3 only or both residues (bivalent) (Fig. 6c). We find that peaks of homogeneous H3K4me3 in sperm are very well conserved after replication (>80%) whereas peaks of homogeneous H3K27me3 are poorly retained (<25%). Interestingly when co-occuring with H3K4me3, H3K27me3 peaks are better retained after replication (~50%). When focusing on the set of genes associated with these retained bivalent peaks, we observed an enrichment for genes misregulated upon H3K4 or H3K27 methylation (Fig. 6d).

We conclude that the homogenous fraction of sperm-modified histone that is propagated in early embryos prime developmental

genes for embryonic expression. We next sought to evaluate if such epigenetic regulatory principle apply in human.

**Homogeneous histone methylation in a human sperm population.** We first applied ICe-ChIP to quantify the level of methylation of the fraction of histone H3 retained in mature human sperm. To that end human sperm from a fertile individual were treated with an MNase concentration that yields nucleosome-sized fragments as well as smaller fragments (Supplementary Fig. 8). Digested chromatin was used for H3K4me3 and H3K27me3 ICe-ChIP-Seq analysis. Interestingly, in human, H3K4me3 and H3K27me3 ChIP recovers only nucleosome-sized fragment (Supplementary Fig. 8). So unlike in *Xenopus*, in human-sperm-modified histone H3 are not found associated with small DNA fragments generated by MNase digestion. We observed that part of the human sperm genome harbours a high density of H3K4me3 or H3K27me3, albeit to a lower extent than observed in *Xenopus* and in accordance with the lower histone retention in human sperm (Fig. 7a, left). Clustering promoters according to H3K4me3 and H3K27me3 density highlighted further differences between human and *Xenopus* (Fig. 7b and Supplementary Data 6). In human, H3K4me3 density is not particularly higher at the TSSs as observed in frog (compare Figs. 3e and 7b). Instead high H3K4me3 HMD is found on broad domains around TSSs of a limited number of genes (cluster 4, 1570 genes). This suggests that human spermiogenesis is associated with a removal of the majority of H3K4me3 that is usually associated with a large proportion of genes TSSs in somatic cells.

To focus on the fraction of the human sperm genome that is homogeneously methylated on histone H3, we then identified peaks of histone modifications and computed their HMD. Overall, the peaks identified are well conserved between our study and previous work[5] indicating that peak calling is consistent across a range of MNase treatments (Supplementary Data 7). Interestingly, however, our analysis shows that in human, a very large fraction of histone methylation peaks identified by conventional ChIP-seq corresponds to low HMD and are therefore heterogeneous in a sperm population (Fig. 7a, right). Nonetheless, some human sperm genes have peaks of high density of histone H3K4me3 (130 genes) or H3K27me3 (320 genes) around their TSSs (i.e., BMI1, TBX3, Fig. 7c and Supplementary Data 8). Genes associated with peaks of H3K4me3 or H3K27me3 methylation in sperm, irrespective of HMD level, show a significant overlap between human and *Xenopus* and these gene sets are enriched for genes expressed at ZGA in both species (Fig. 7d)[29]. By contrast, no conservation is observed between *Xenopus* and human orthologous genes associated with the homogeneous-methylated fractions (peaks with HMD > 80) of the sperm genome (Fig. 7d). However, in both species the set of genes with high H3K27me3 HMD is related to

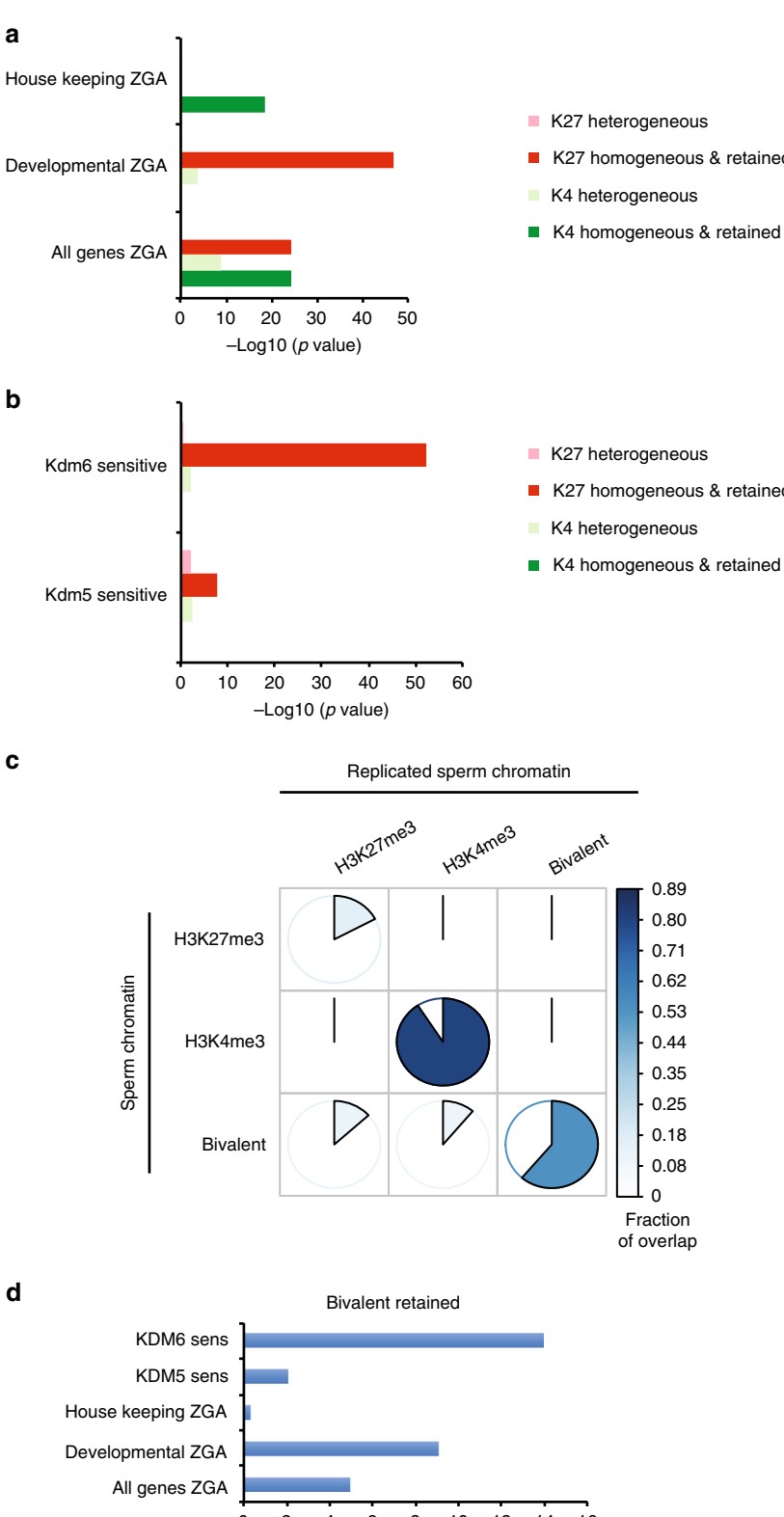

developmental functions (Supplementary Tables 3 and 7). In addition, an enrichment of high HMD peaks for ZFP281 binding sites, a transcription factor implicated in early mammalian embryonic development, is found in both human and *Xenopus*. These observations raise the possibility that the fraction of the sperm genome with homogenous histone methylation could contribute to early embryonic development in human as it is the case in *Xenopus*, but through the regulation of a different set of genes.

We observe that certain genomic sites, when retaining histones, always retain them in their methylated form. However, unlike in *Xenopus*, we cannot rule out that histone retention happens only in a subset of sperm cells in humans. DNA methylation and histone H3K27 methylation have been shown to be mutually

**Fig. 6 The homogeneously methylated histone fraction in frog sperm chromatin regulates embryonic gene expression. a** Barplot of enrichment $p$ values ($-\log 10$) in set of genes expressed at zygotic gene activation (all-, developmental- and house keeping-ZGA genes) for the presence of different type of H3K4me3 or H3K27me3 sperm peaks. pink and light green bars = enrichment $p$ value for heterogenous (HMD < 80) peaks in promoter; red and green bars = enrichment $p$ value for homogeneous (HMD > 80) peaks in promoter that are also retained after replication. $P$ values determined by the $\chi^2$ proportion test evaluating if the proportion of gene of interest is higher (alternative = 'greater') than the genome-wide proportion (i.e., expected proportion). **b** Barplot of enrichment $p$ values ($-\log 10$) in set of genes whose embryonic expression is sensitive to the presence of histone H3K4 demethylase (Kdm5b) or histone H3K27 demethylase (Kdm6b) at fertilisation[2] for the presence of the aforementioned sperm peaks categories. $P$ values are obtained as in **a**. **c** Pie chart indicating the percentage of the homogeneously methylated histone present in sperm that are retained after egg extract-mediated replication. Three type of sperm peaks are considered: homogeneous for H3K27me3 only, homogeneous for H3K4me3 only and homogeneous for H3K27me3 and H3K4me3 (bivalent). The area of the coloured sector (together with the colour) indicates the fraction of the overlap between each pair-wise comparison. **d** Barplot of enrichment $p$ values ($-\log 10$) in various set of genes for the presence of a homogeneous bivalent H3K4me3 and H3K27me3 retained after replication. $P$ values are obtained as in **a**. HMD are from data pooled from two independent replicates. Peaks retention/lost are consensus from three independent replicates. Set of kdm5 and kdm6 sensitivite genes are from three independent replicates.

exclusive on chromatin[30]. Therefore, we would expect that if peaks of high HMD methylation on H3K27 reflected the situation in most sperm cells, they would be associated with genomic regions that are uniformly DNA hypomethylated in a human sperm population. We checked this proposition using published bisulfite sequencing data from single human sperm cells[31]. This dataset allowed us to identify genomic region that harbour unmethylated DNA in all sperm. We found that high HMD peaks (HMD > 80) are indeed more likely to be associated with such unmethylated sperm DNA than low HMD peaks (0 < HMD < 80) (Fig. 7e). The observed correlation between lack of DNA methylation in all sperm and high HMD peak of H3K27 fit with the hypothesis that high HMD peaks represent methylated histones present in most human sperm cells. In addition, the set of genes with high HMD on H3K27 in sperm is enriched for genes that have a closed TSS configuration in most cells of the human embryos undergoing ZGA, as shown by ATAC-seq[31] (Fig. 7f). This suggests that methylated histones on sperm gene TSSs could regulate chromatin accessibility in embryos at ZGA.

We conclude that in human sperm, histones near some developmental genes are always retained in their methylated form. We provide indirect evidence that methylated histones might also be homogeneously distributed in a human sperm population as in *Xenopus* and could contribute to the regulation of embryonic chromatin status at ZGA.

## Discussion

Multiple lines of evidence point towards the transmission of epigenetic information from parents to offspring[32]. However, the molecular basis of this epigenetic information is unclear. In particular the notion that sperm-derived histones are randomly retained along the genome has gained support, leading to the proposal that modified histones cannot be the basis for the epigenetic information required for embryo development[14]. Here we provide evidence to the contrary. The particular core histone remodelling event occurring during *Xenopus laevis* spermiogenesis enabled us to refine epigenetic maps of the sperm by identifying genomic sites where most sperm of a population harbours a given epigenetic feature. This approach uncovers a subset of genes in the genome that retains modified histones in all sperm (Fig. 7g). Compared with genes with heterogeneous epigenetic composition, these genes are characterised by their expression at ZGA, their persistent association with modified histones after replication and their sensitivity to histone demethylation at fertilisation. Our analysis demonstrates that sperm homogenously modified histones across a cell population contribute to the transmission of epigenetic information necessary for the regulation of embryonic gene expression in *Xenopus laevis*. Moreover, we identify genomic sites where histones are always retained as

methylated in human sperm. Although we cannot assume that all sperm retain histones at these locations in human, these genomic sites represent potential candidate regions for the epigenetic programming of human sperm for embryonic expression. This suggests that a modified histone-based mechanism of inter-generational epigenetic transmission might be conserved between species.

How is the epigenetic information encoded in sperm-modified histone then transmitted in the developing embryos? In Zebrafish and *Xenopus*, 10–12 cell division occur before the major wave of zygotic genome activation. In order to affect embryonic gene expression, sperm-derived epigenetic information therefore needs to be maintained through multiple cell division. In Zebrafish, previous work indicated that modified histones were not detected at early embryonic stage, ruling out direct transmission of sperm-derived modified histones during early embryogenesis[33,34]. However more recent work indicated that sperm-derived histone variant H2A.Z acts as a placeholder in early embryos where it regulates DNA methylation required for proper transcription and development[35]. In the case of *Xenopus laevis* sperm a direct transmission mechanism of H3K27me3 is also unlikely. Indeed, we observe that a much larger fraction of H3K27 methylation than that of H3K4 methylation is lost after replication (Fig. 5b), in agreement with H3K27me3 ChIP-seq data obtained in *Xenopus tropicalis* that detected very low level of this mark in gastrulae[36]. We hypothetize that a relay mechanism (placeholder) or other histone marks associated with H3K27me3 carry the information into the embryo. By contrast, we observe that most of the H3K4 methylation peaks present in *Xenopus laevis* embryos just before ZGA could be traced back to the homogeneous fraction of H3K4 methylation identified in sperm. In human a recent report also indicates that H3K4me3 persists from fertilisation to post-ZGA stage embryos[37]. Altogether this suggests that sperm-derived modified histones contribute to the establishment of the embryonic epigenome prior to ZGA.

We find that the homogenous fraction of methylated histones in a *Xenopus* sperm population is transmitted through the first embryonic cell cycle and potentially through cell divisions of the developing embryos to regulate early embryonic gene expression. Defect in these epigenetic programming of sperm for embryonic development could be at the origin of case of idiopathic male infertility. In addition, the natural mechanisms of epigenetic information transmission from the sperm to the developing embryos might also be responsible for the maintenance of an epigenetic memory of somatic cell identity that characterise embryos generated by nuclear transfer[2,38]. By getting a better understanding of the processes underlying the transmission of paternal epigenetic information, we will be able to devise better strategies to erase somatic cell identity and improve cloning efficiency.

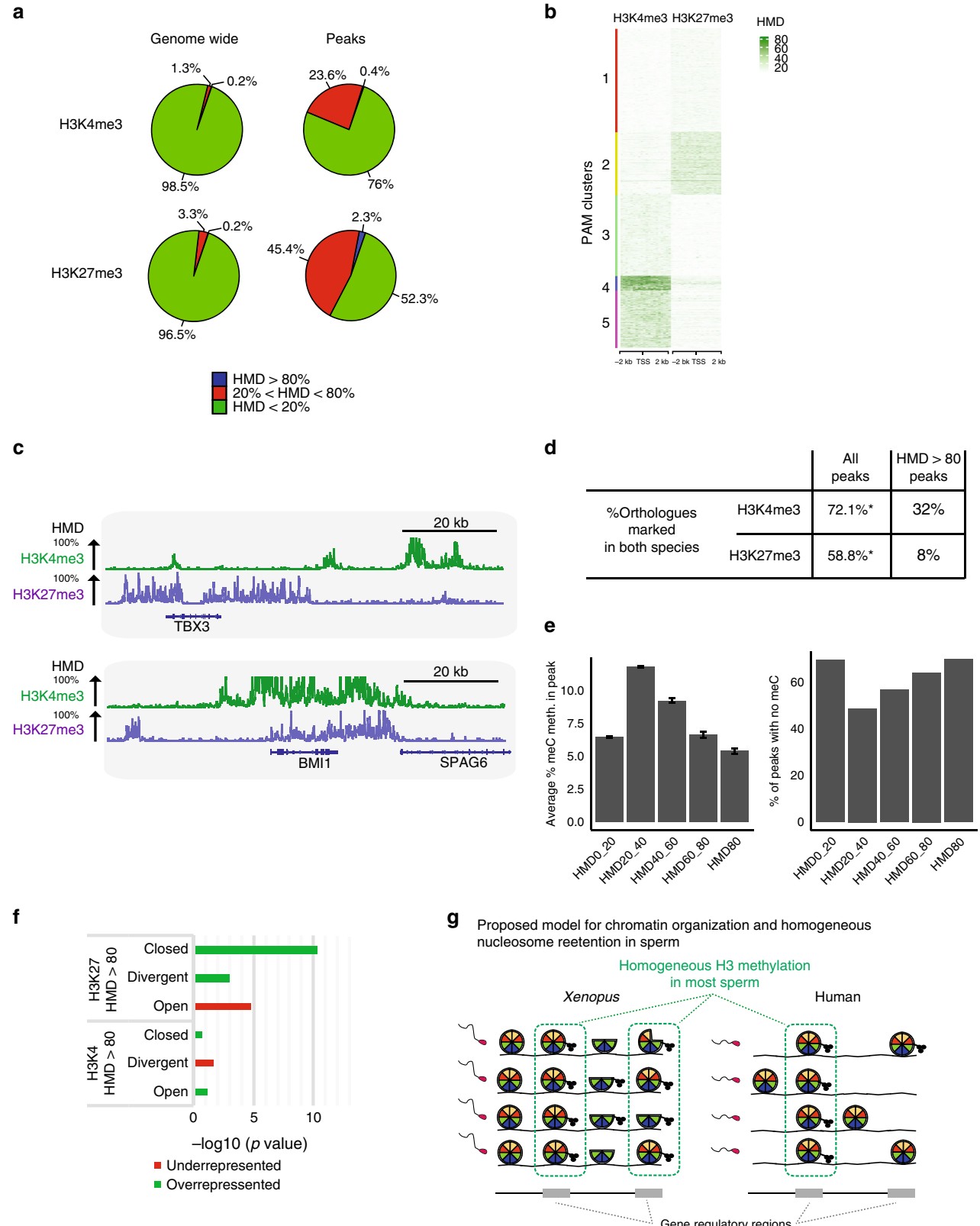

## Methods

**Experimental model and subject details**. Mature *Xenopus laevis* males were obtained from Nasco (901 Janesville Avenue, PO Box 901, Fort Atkinson, WI 53538-0901; https://www.enasco.com/xenopus). Our work with Xenopus complied with all relevant ethical regulations for animal testing and research. This work is covered under the Home Office Project License PPL 70/8591 and frog husbandry and all experiments were performed according to the relevant regulatory standards.

Animals were kept in a Marine Biotech recirculating system at a density of one adult/3l, with 10% water change per day. Water was sequentially filtered with mechanical pad sump filter, nitrifying bacteria filter, mechanical canister filter, carbon filter and UV sterilised. Water quality parameters were as follow: conductivity 1500 uS; temperature 17–22 °C; PH 6–8. Photoperiod was set to 12 h ON/ 12 h OFF. Frogs are fed twice per week with Royal Horizon 4.5 mm pellets (skretting, https://www.skrettingfishfeeds.co.uk/). Unconsumed food was removed

**Fig. 7 Homogeneous retention of methylated nucleosome in a human sperm population. a** Percentage of the genome (left) and percentage of peaks (right) with different levels of H3K4 or H3K27 methylation density in human sperm. **b** PAM clustering of H3K4 and H3K27 HMD levels at promoter regions (TSS +/−2 kb) in human sperm. **c** Genome browser screenshots of TBX3 and BMI1 HMDs in human sperm. **d** Percentage of gene orthologues with peaks of histone methylation in both human and *Xenopus* sperm. **e** Barplots of the average percentage (%) of 5mC methylation at H3K27me3 peaks stratified by methylation level (left); error bars: sem; barplot of the percentage of H3K27me3 peaks showing absence of 5mC methylation (right) (5mC methylation: single-sperm bisulfite sequencing data from ref. [31]). **f** Barplots indicating −log10 ($p$ value) for enrichment of sperm TSSs with HMD > 80 for H3K4 or H3K27 in set of genes with TSSs showing different chromatin accessibility level in eight cell embryos (open and closed corresponds to TSSs open or closed in all cells of a eight cell embryos), whereas divergent corresponds to TSSs either open or closed in different cells (ATAC-seq data from ref. [31]); $p$ values determined by $\chi^2$ proportion test evaluating if the proportion of gene of interest is higher (green) or lower (red) than the genome-wide proportion (i.e., expected proportion). **g** Model of epigenetic homogeneity in *Xenopus* and human sperm. The cartoon summarises the observed trends in retention of modified histones across the genome in those species. Source data related to **b** and **e** are provided as Source Data files.

---

10 min after the start of feeding. The researchers and the staff of the Gurdon Institute animal husbandry facility are trained in these experiments, and veterinarians monitor the health status of the animals.

**Xenopus sperm collection.** *Xenopus* sperm collection was performed as described before[2]. For each round of sperm purification, testes from six adult *Xenopus laevis* males were isolated and manually cleaned from blood vessels and fat bodies in 1× Marc's modified ringers (MMR, 100 mM NaCl, 2 mM KCl, 1 mM MgSO₄, 2 mM CaCl₂, 3 mM HEPES, pH 7.4) using forceps and paper tissues. It is crucial to clean the testes well from any non-testicular tissues, as otherwise the cells released from the tissues may negatively affect the final purity of isolated cells. Subsequently, testes were torn into small pieces with forceps and homogenised with 2–3 strokes of a dounce homogeniser (tissue from one testis at a time). The cell suspension was then filtered to remove tissue debris and cell clumps (CellTrics, cat. 04-0042-2317) and spun down at 800 rcf, 4 °C, for 20 min. Supernatant was discarded and the cell pellet was resuspended in 12 mL of 1× MMR. If any red blood cells were visible at the bottom of the pellet (a result of incomplete removal of blood vessels), only the uncontaminated part of the pellet was recovered, taking extreme care not to disturb the red blood cells. Subsequently, step gradients of iodixanol (Optiprep; Sigma, D1556; 60% iodixanol in water) in 1× MMR final were manually prepared in pre-chilled 14 mL ultra-clear centrifuge tubes (Beckman Coulter, #344060) in the following order from the bottom to the top of the tube: 4 mL of 30% iodixanol, 1 mL of 20% iodixanol, 5 mL of 12% iodixanol (all in 1× MMR) and 2 mL of cell suspension in 1× MMR on top. Gradients were spun down in a pre-chilled SW40Ti rotor at 7500 rpm (10,000 g), 4 °C, for 15 min, deceleration without brake (Beckman Coulter Ultra-centrifuge, Optima L-100XP). The pelleted fraction, containing mature sperm, was collected. Collected fractions were diluted six times with 1× MMR and collected by spinning first at 805 rcf, 4 °C, for 20 min and respinning at 3220 rcf, 4 °C, for 20 min to pellet remaining cells. Pelleted cells were subsequently permeabilized with Digitonin (10 mg/mL as a final concentration, Sigma, D141) for 5 min at room temperature (RT) as detailed before[39].

**Human-sperm collection.** All human sperm samples were processed in accordance to ReproMed Ireland's standard procedures and ethical approval was obtained from the University College Dublin Human Ethics Committee (HREC) (protocol number LS-16-53-ODoherty-Fair). The approval process entailed independent peer review along with approval from the HREC. Human-sperm samples were donated to the research project by informed consent. Written informed consent was obtained from male partners of expectant couples/couples that recently had a baby (within 6 months of sample donation) that donated fresh sperm samples following 3 days of abstinence. Before giving consent, donors were provided with all of the necessary information about the research project and contact information for the project lead. Specifically, patients signed a consent form authorising the use of their sperm samples for future research purposes, including molecular and epigenetic analyses, and for the results of these studies to be published in scientific journals. No financial inducements were offered for donation.

Chromatin was prepared from fresh ejaculates donated from men with proven fertility. Men were deemed fertile based on the following parameters; female partners are currently pregnant or have given birth within 6 months of sample production. Semen samples were collected from men aged between 30 and 35 years old. Ejaculates were maintained at 37 °C for 10 min immediately following production. A 10 µl aliquot of each fresh ejaculate was subjected to routine semen analysis prior to processing and all samples were in accordance with the World Health Organization guidelines (World Health Organization. WHO laboratory manual for the examination and processing of human semen (5th ed.), WHO Press (2010)) for normal semen samples (sperm density, total number, motility, morphology and semen volume). The recorded parameters for sperm preparations used in this study were within the following range: progressive motility (PR) = 74–81%; non-progressive motility (NP) = 10–13%; immotile (IM) = 9–11%; total motility (PR + NP) = 87–91%.

Motile spermatozoa were purified on a discontinuous Percoll gradient (Pharmacia, Uppsala, Sweden), as outlined previously[40]. Gradient-purified spermatozoa were quantified, resuspended in Hepes-buffered Tyrode's medium

(Sigma). Permeabilization of sperm nuclei was performed with Digitonin following the same procedure as for *Xenopus* samples.

**Xenopus cell culture.** Cell line XL-177 was derived from tadpole epithelium of *X. laevis*[41]. Cultured cells were maintained in medium containing L-15 (SIGMA L1518), sterile water and FBS (6:3:1vol/vol) and supplemented with 100 µl penicillin/streptomycin. Cells were grown at 23 °C in gelatin-coated dish that were sealed. Confluent culture of cells were split 1:3 to 1:6 following trypsin digestion.

**Interphase egg extract preparation.** Eggs were collected in 1× MMR, de-jellied with 0.2× MBS (Modified Barth's Saline) including 2% cysteine (Sigma, W326305) (pH was adjusted to 7.8 using 10 N NaOH) and washed with 0.2× MMR. Subsequently, eggs were activated for 3 min at RT with 0.2× MMR supplemented with 0.2 µg/mL calcium ionophore (Sigma, C7522). Eggs were rinsed with 0.2× MMR and subsequently all abnormal or not-activated eggs were removed. Eggs were washed with 50 mL of ice-cold extraction buffer (EB) (5 mM KCl, 0.5 mM MgCl₂, 0.2 mM DTT, 5 mM Hepes pH 7.5) supplemented with protease inhibitors (PIs) (Roche, 11873580001), transferred into centrifugation tube (Thinwall. Ultra-Clear™, 5 mL, 13 × 51 mm tubes, Beckman Coulter Inc., UK; 344057) and supplemented with 1 mL of EB buffer with PI and 100 µg/mL of cytochalasin B (Sigma, C2743) and placed on ice for 10 min. Subsequently, eggs were spun briefly at 350 $x$ g for 1 min at 4 C (SW55Ti rotor, Beckman Coulter Ultra-centrifuge, Optima L-100XP) and excess buffer was discarded. Eggs were then spun at 18,000 $x$ g for 10 min at 1 C, the extract was collected with a needle, transferred to a fresh, pre-chilled tube, supplemented with PI and 10 µg/mL of cytochalasin B, and respun using the same conditions. Extract was collected with a needle and used fresh for the replication assay.

**Expression and purification of Delta-Geminin.** pET28 Delta-Geminin were transformed into *E.coli* and transformants were cultured overnight in 5 ml of LB medium, containing 50 µg/ml kanamycin and 30 µg/ml chloramphenicol. Overnight cultures were transferred into 250 ml of LB medium without glucose and antibiotics and incubated for 2–2.5 h at 37. One hundred microlitres of 1 M isopropyl 1-thio-beta-D-galactopyranoside was added into culture to induce recombinant protein expression. After 2 h induction, cells were collected by centrifugation. Cell pellets were suspended into 30–40 ml of MilliQ water and transferred into 50 ml tubes. After centrifugation, supernatant was removed and pellets were stored at −80 °C. Frozen pellets were suspended into 5 ml of Dicis buffer (300 mM NaCl, 150 mM KoAC, 20 mM Tris, pH7.5, 2 mM MgCl₂, 10% glycerol, 0.01% NP40), with 0.1% NP40 and PIs, on ice. The suspensions were sonicated with Vibra-Cell Ultrasonic Processor (SONICS) for six times 15 s on and off cycles. Sonicated samples were centrifuged at 15,000 rpm for 10 min at 4 °C and supernatants were collected. Ni-NTA agarose (Qiagen, 30210) was washed with 5 ml of Dicis buffer for three times in open column. Equilibrated Ni-NTA agarose was transferred into supernatants and rotated for 1 h at 4 °C. Agarose beads were transferred into open column and washed twice with 10 ml of Dicis buffer with 20 mM imidazole. Bound protein was eluted with Dicis buffer with either 100 mM or 200 mM imidazole for three times, respectively. Three fractions with highly concentrated delta-Geminin were pooled. Fractions were applied to PD10 column (GE Healthcare) equilibrated with buffer containing 10 mM Tris-HCl, pH8.0, 0.5 M NaCl, 5% glycerol and eluted with 3 ml of buffer.

**DNA replication in egg extracts.** Freshly prepared egg extracts were supplemented with 0.005 mg/ml creatine kinase (Roche, 10127566001), 0.375 mM creatine phosphate (Roche, 10621714001), 0.05 mM ATP (Roche, 10519979001), 0.005 mM EGTA, 1 mM MgCl₂. In some experiments, delta-geminin was added (final concentration 1.5 µg/ml) to egg extract to inhibit DNA replication. Permeabilized sperm cells were added to a final concentration of 1000 nuclei/µl of extract and incubated at RT for 2 h with gentle tapping every 10 min. The reaction was stopped by adding 10 volumes of ice-cold Egg Lysis Buffer—Chromatin Isolation Buffer (ELB-CIB, 10 mM Hepes pH 7.8, 250 mM sucrose, 2.5 mM MgCl₂, 50 mM KCl, 1 mM DTT, 1 mM EDTA, 1 mM spermidine, 1 mM spermine, 0.1% Triton X-100,

10 mM sodium butyrate, 1× EDTA-free PI Cocktail (Roche, 05056489001)) following Wang et al.[42], with slight modifications. Chromatin was isolated via centrifugation at 4000 rpm for 5 min through a 0.3 mL sucrose cushion of ELB–CIB with 0.5 M sucrose underlayered in the tube. The pellet was washed once with ELB–CIB plus 250 mM KCl. Samples were aliquoted (0.5–1 million of cells per tube) and flash frozen in liquid nitrogen.

**Quantification of DNA replication activity in egg extract.** To visualise replication in individual nuclei, 33 μM rhodamine dUTP (Roche, 11534378910) was added at the indicated times (20, 40, 60, 90 min after incubation) for 5 min as indicated, and the reaction was stopped with 1× PBS and fixed in 4% paraformaldehyde. The reaction was spun through a 30% sucrose cushion in PBS. The sperm cells were rinsed twice in 1× PBS for 10 min and mounted in Vectashield mounting medium with DAPI (Vector laboratories, H-1200). Images were acquired with a Zeiss 510 META confocal LSM microscope (Zeiss). Counting of rhodamine positive cells and measurement of rhodamine signal intensity were performed in Image J software.

**Sperm chromatin immunoprecipitation (ChIP).** Chromatin fractionation and ChIP were performed as described before[2,43] with slight modifications. Eighty microlitre of Magnetic beads (M-280 Sheep anti-rabbit IgG, Invitrogen, #11204D) were used per reaction and all wash steps were carried out with a magnet. Beads were washed with 1 ml of TE pH 8.0, lysis buffer (LB)(1:1 mixture of Buffer 1 and MNase buffer, see below), and incubated in 1 ml of LB with 100 μl of 10 mg/ml BSA at 4 °C for 30 min ('pre-blocking'). Pre-blocked beads were washed with 1 ml of LB twice and resuspend in 80 μl of LB. Half of beads were incubated with antibody overnight and leftovers were stored in 4 °C for 'pre-clear chromatin' step. For ChIP, following antibodies were used: anti-H3K4me3 (Abcam, ab8580) and anti-H3K27me3 (kind gift from Dr Thomas Jenuwein). Egg extract-treated sperm chromatin was resuspended in 50 μl of Buffer 1 (0.3 M Sucrose, 15 mM Tris pH 7.5, 60 mM KCl, 15 mM NaCl, 5 mM MgCl₂, 0.1 mM EGTA, 0.5 mM DTT) and added 50 μl of Buffer 1 with detergent (Buffer 1 including 0.5% NP40 and 1% NaDOC). Samples were incubated for 10 min on ice. One hundred microlitres of MNase buffer (0.3 M Sucrose, 85 mM Tris, 3 mM MgCl₂, 2 mM CaCl₂, 2.5 U of micrococcal nuclease: Roche 10107921001) was added in to each tube (0.5 million of cells per tube). Tubes were incubated at 37 °C for 30 min in pre-warmed water bath. Reaction was stopped by adding 2 μl of 0.5 M EDTA pH 8.0 in the same order as started. Tubes were vortexed and placed on ice for at least 5 min. Supernatant and pellet were separated by centrifugation at 13,000 rpm for 10 min at RT. Beads (stored at 4 °C for pre-clear chromatin) were washed with 1 ml of LB twice. Supernatant and EDTA-free protenase inhibitor cocktail were added into the beads and rotating on the wheel at 4 °C for 60 min. Ten percent of volume of pre-cleared chromatin was taken as 'input' and stored at 4 °C. Antibody-conjugated beads were washed with 1 ml of LB twice. Pre-cleared chromatin were added to washed beads and incubated at 4 °C for 6 h. After incubation, beads were washed with 1 ml of washing buffer A (50 mM Tris-HCl pH 7.5, 10 mM EDTA and 75 mM NaCl) on the wheel at 4 °C for 5 min and buffer was discarded carefully. Subsequently, beads were washed with 1 ml of washing buffer B (50 mM Tris-HCl pH 7.5, 10 mM EDTA and 125 mM NaCl) and buffer was discarded carefully. Another 1 ml of washing buffer B was added and transferred to 1.5 ml protein lo-bind tubes (Eppendorf, 0030108116) with beads. Buffer was discarded carefully. One hundred and fifty microlitres of elution buffer (1:9 mixture of 10% SDS and TE) was added. Tubes were placed at 25 °C on the Eppendorf thermomixer and shaked at 800 rpm for 15 min. Supernatant were taken and transferred to the new tube using magnetic stand. Another 150 μl of elution buffer was added and repeated this step (in total 300 μl). Input samples were diluted to 300 μl total by adding TE. Two microlitres of 2 mg/ml RNase A (heat-inactivated or DNase-free) was added and incubated at 37 °C for 30 min. Four microlitres of 10 mg/ml Proteinase K was added and incubated at 55 °C overnight. After incubation, tubes were shortly spun. Three hundred microlitres of Phenol/chloroform were added and vortexed for 30 s. Samples were centrifuged at 13,000 rpm for 10 min at RT. One microlitre of Glycogen, 1/10 volume of NaAcetate and 2 volume of 100% EtOH were added and vortexed. Tubes were stored on dry ice for 20–30 min. After freezing, tubes were centrifuged at 13,000 rpm for 30 min at 4 °C. Supernatant was discarded carefully and 500 μl of 70% EtOH was added. Tubes were centrifuged at 13,000 rpm for 10 min at 4 °C. Pellets were dried and suspended in 17 μl of ddH₂O. DNA was excised and subjected to library preparation with a TruSeq DNA Kit (Illumina, FC-121-2001).

**Internal standard calibrated ChIP (ICe-ChIP).** Most procedures were performed following the ChIP section protocol. For internal standard calibration, semi-synthetic nucleosomes were spiked in just before MNase digestion. We used the semi-synthetic standards used in Grzybowski et al.[18] and provided by the Ruthenburg's lab. We spiked in the standards aiming for the second lowest concentration of the barcoded nucleosomes to be at the same concentration as the genome count (to that end we simply multiply the amount of the nuclei in the sample with the number of genome copies per nucleus (2.5 for dividing diploid; 2 for stationary diploid; 1 for haploid). We then add the amount of the ladder equivalent to the member representing the 5% or 10% of the ladder so that some standards will be below and above the expected genome coverage). Sperm

chromatin and semi-synthetic nucleosomes were digested with 2.5 U (Xenopus) and 0.5 U (human) of MNase for 30 min at 37 °C. The following antibodies were used: anti-H3K4me3 (Abcam, ab8580) and anti-H3K27me3 (kind gift from Dr Thomas Jenuwein). Apparent inflation of HMD values for H3K27me3 beyond what is physically possible indicates off-target binding by the antibody, most likely H3K27me2 as we have anecdotally observed in prior ICe-ChIP experiments, or a consequence of PCR-duplicate removal's bias for excluding IP versus input reads. As a consequence, we express these data as apparent HMD %.

**Embryos ChIP.** ChIP was performed as described previously[44,45] with the following modifications. Blastula (stage 7) embryos were generated by in vitro fertilisation. For each ChIP experiment, 20–30 embryos were fixed in 2 mL of 1% formaldehyde in 1× MMR for 15–25 min at RT, followed by four washes with 1 ml of 1× MMR and equilibration in 500 μl HEG solution (50 mM HEPES-KOH pH 7.5, 1 mM EDTA, 20% Glycerol) at 4 °C, then excess buffer was removed and samples were frozen at −80 °C. To extract chromatin, the samples were transferred to 2 ml tube or 15 ml Falcon tube and homogenised in 200–250 μl of sonication buffer (20 mM Tris-HCl pH 8.0, 70 mM KCl, 1 mM EDTA pH 8.0, 10% Glycerol, 5 mM DTT, 0.125% NP40, 1× complete PIs), by pipetting up and down in a 1 ml pipette tip. Two hundred and fifty microlitres of diagenode beads were transferred to diagenode falcon tube (Diagenode, C1020031). Seven hundred and fifty microlitres of embryo lysates were added and diagnode falcons were transferred to the sonicator bath (diagnode bioruptor). Sonication is carried out in 30 cycles (with 30 s on/off cycles). Sonicated embryo extract was transferred into eppendorf tubes. Chromatin was collected by centrifugation for 5 min at top speed in tabletop centrifuge at 4 °C. Chromatin extract was transferred to new tube and stored in ice. Before ChIP, 50 μl (20% of extract) of chromatin was taken as an input control and stored at 4 °C on ice. For each ChIP reaction, 200 μl of chromatin extract was mixed with 1–5 μg of antibodies and incubated for 2 h at 4 °C on a rotating wheel, in the mean time beads were prepared. Twenty microlitres of beads (Dynabeads M-280 sheep anti-rabbit IgG or Protein A/G dynabeads) were taken and washed for three times in 1× PBS containing 1% BSA and once in ChIP incubation buffer (50 mM Tris-HCl pH 8.0, 100 mM NaCl, 2 mM EDTA, 1 mM DTT, 1% NP40, 1× complete PIs). Beads were resuspended in 100 μl of ChIP incubation buffer per reaction and added to chromatin-antibody mix. Samples were incubated at 4 °C with overnight rotating. After reaction, supernatant was removed. Beads were sequentially washed with 1 ml of wash buffer 1 (50 mM Tris-HCl pH8.0, 100 mM NaCl, 2 mM EDTA, 1 mM DTT, 1% NP40, 0.1% doxycholate, 1× complete PIs) for 5 min on ice, and supernatant was removed subsequently. Washing was performed with this step: 1 ml wash buffer 2 (50 mM Tris-HCl pH8.0, 500 mM NaCl, 2 mM EDTA, 1 mM DTT, 1% NP40, 0.1% deoxycholate, 1x complete PIs), 1 ml of wash buffer 3 (50 mM Tris-HCl pH8.0, 100 mM NaCl, 250 mM LiCl, 2 mM EDTA, 1 mM DTT, 1% NP40, 0.1% deoxycholate, 1× complete PIs), 1 ml of wash buffer 1 and 500 μl of TE buffer (10 mM Tris-HCl, pH 8.0, 1 mM EDTA). After washing, supernatant was removed. Four hundred microlitres of elution buffer was added to the beads, and 350 μl of elution buffer to the input. Sixteen microlitres of 5 M NaCl was added to each tube. Cross-links were reversed by incubating for 5 h at 65 °C in a thermomixer shaking at 1000 rpm. Eight microlitres of RNaseA (DNase-free, 10 mg/ml) was added to 400 μl of reaction and incubated for 30 min at 37 °C. Four hundred microlitres of Phenol/chloroform/isoamylalcohol (25:24:1) pH 8.0 and mixed using vortex mixer for 30 s. Samples were centrifuged at RT for 3 min with top speed. This extraction step was repeated again. One hundred microlitres of chloroform was added and mixed by vortex mixer for 30 s. Samples were centrifuged at RT for 3 min at top speed. Aqueous upper phase was transferred to a new tube. For ethanol precipitation, 3 μl of glycogen (20 mg/ml), 40 μl of 3 M NaAcetate pH 5.2 and 1200 μl of ethanol (96%) were added. Precipitation was performed overnight at −20 °C or for 20–30 min on dry ice. Samples were centrifuged at top speed for 10 min. Supernatant was removed and pellet was washed with 1 ml of cold ethanol (70%). Samples were centrifuged at top speed for 10 min. This washing step was repeated again. All supernatant was removed and pellet was air-dried for 10–15 min. Pellet was dissolved in 50 μl of ddH₂O.

**Library preparation and sequencing.** ChIP samples were subjected for ChIP-seq library preparation using the TruSeq DNA kit (Illumina, FC-121- 2001) and Agencourt AMPure XP beads (Beckman coulter, A63881) with slight modifications. After end repair, A-tailing and adaptor ligation, DNA fragments were purified with two volumes of beads. Purified DNA was amplified by 18–22 cycle of PCR cycle and size selection was performed. Greater than 370 bp DNA fragments were removed with 0.7 volume of beads, these selected DNA was recovered with 1.8 volume of beads, and >250 bp DNA fragments were collected with 1.0 volume of beads.

DNA length was analysed on a D1000 screen tape (Agilent, 5067-5582) using D1000 sample buffer (Agilent, 5067-5583) on an Agilent 2200 tape station. DNA quantity was analysed on a Qubit dsDNA HS assay kit (Invitrogen, Q32851) using Qubit system (Invitrogen). Single-end 50 bp sequencing was performed for ChIP-seq libraries of egg extract-treated sperm and blastula DNA, and paired-end 32 bp sequencing for ICeChIP-seq libraries of Xenopus and human sperm DNA on a HiSeq 1500 sequencer (Illumina).

**Chromatin particles separation on linear sucrose gradient.** *Xenopus* sperm chromatin was digested with MNase as described[2]. Five and fifteen percent sucrose buffer (5%: 10 mM NaCacodylate pH7, 1 mM EDTA, 0.5 mM EGTA, 50 mM KCl, 5% (w/v) sucrose, 0.5 mM PMSG, 5 mM 2-mercaptoethanol and 1× PI) (15%: same as 5% but with 15% (w/v) sucrose instead) were prepared following Ruthernburg's lab website protocol with slight modification (http://ruthenlab.org/web/Protocols.html). 5.5 ml of 15% sucrose buffer was transferred to 14 ml tube (Beckman coulter, 344060) and 5.5 ml of 5% sucrose buffer was layered carefully. Tubes were leaned for 3 h at RT to make linear gradient. One millilitre of digested sperm chromatin was loaded on each tube. For nucleosome and subnucleosome separations, tubes were centrifuged at 36,000 rpm for 15–16 h at 4 °C. After centrifugation, 250 μl of each fraction was collected carefully starting from the top of the gradient. Fifty microlitres of each fraction was collected for DNA extraction. DNA was extracted by the Quiaquick DNA extraction kit (Qiagen, 28304), and DNA length was analysed on D1000 screentape (Agilent, 5067-5582), according to manufacturer instructions. Fraction containing 150 bp length DNA was treated as 'nucleosome fraction' and 70–110 bp length DNA was treated as 'subnucleosome fraction'. The remaining part of the fractions was pooled depending on DNA length. Proteins were precipitated by incubating with 20% TCA on a rotating wheel for 3 h at 4 °C. Proteins were pelleted by centrifugation at 10,000 rpm for 15 min at 4 °C. Pellets from each biological replicate were washed with ice-cold acetone twice and dried. Five subnucleosome and nucleosome fractions were taken for mass spectrometry analysis. Out of five, one replicate for each condition is doubled up in order to increase the signal.

**Quantitative mass spectrometry.** TCA precipitated protein pellets of fractions were suspended in buffer containing 100 mM Triethylammonium bicarbonate/0.1%, heated at 90 °C for 5 min followed by bath sonication for 30 s. Proteins were reduced with 2 μl of 50 mM tris-2-caraboxymethyl phosphine for 1 h at 60 °C followed by alkylation with 1 μl of 200 mM methyl methanethiosulfonate for 10 min at RT. Proteins were digested overnight at 37 °C using trypsin (Thermo Scientific) asprotease at ratio protein/trypsin ~1:30. Protein digests were labelled with the TMT-10plex reagents (Thermo Scientific; Lot #SA239882A) for 1 h. The reaction was quenched with 8 μl of 5% hydroxylamine (Thermo Scientific) for 15 min at RT. Samples were mixed and subsequently fractionated using the basic pH Reversed-Phase Peptide Fractionation kit (Thermo Scientific). Fractions were dried and each fraction was reconstituted in 0.1% formic acid for liquid chromatography tandem mass spectrometry (LC–MS/MS) analysis.

**LC-MS/MS**

*Peptide fractions were analysed on a Fusion Lumos Orbitrap.* Mass spectrometer (Thermo Scientific) was coupled with RSLC nano Ultimate 3000 system. Peptides were trapped on a 100 μm ID × 2 cm microcapillary C18 column (5 μm, 100 A) followed by 2 h elution on a 75 μm ID × 25 cm C18 RP column (3 μm, 100 A) with 5–45% acetonitrile gradient in 0.1% formic acid at 300 nl/min flow rate. In each data collection cycle, one full MS Orbitrap scan (380–1500 m/z) was acquired at 120 K resolution, automatic gain control (AGC) setting of $3 \times 10^5$ and maximum injection time (MIT) of 100 ms. Subsequent MS2 scans were acquired with a top speed approach using a 3-s duration. The most abundant ions were selected for fragmentation by collision induced dissociation (CID) with a collision energy of 35%, an AGC setting of $1 \times 10^4$, a quadrupole isolation window of 0.7 Da and MIT of 35 ms and detected in ion trap. Previously analysed precursor ions were dynamically excluded for 45 s. During the MS3 analyses for TMT quantification, precursor ion selection was based on the previous MS2 scan and isolated using a 2.0 Da m/z window. MS2–MS3 was conducted using sequential precursor selection methodology with the top 10 setting. HCD was used for the MS3, using 55% collision energy. Reporter ions were detected using the Orbitrap at 50 K resolution, an AGC setting of $5 \times 10^4$ and MIT of 86 ms.

**Quantitative mass spectrometry data processing.** The collected CID tandem mass spectra were processed with the SequestHT search engine against the *Xenopus laevis* proteome database on the Proteome Discoverer 2.1 software for peptide and protein identifications. The node for SequestHT included the following parameters: precursor mass tolerance 20 ppm, FRagment Mass tolerance 0.5 Da, dynamic modifications were methionine oxidation (+15.995 Da), aspartamine and glutamine deamination (+0.984 Da) and Static Modifications were TMT6plex at any N-Terminus, K (+229.163 Da) for the quantitative data. Methylthio at C (+45.988) was included for the total proteome data. The Reporter Ion Quantifier node included a TMT 6plex (Thermo Scientific Instruments) Quantification Method, for MS3 scan events, HCD activation type, integration window tolerance 20 ppm and integration method most confident centroid. The consensus workflow included S/N calculation for TMT intensities, and the level of confidence for peptide identifications was estimated using the Percolator node with decoy database search. Strict FDR was set at $q$ value < 0.01. Quantitative proteomics data were analysed by qPLEXanalyzer, an R Bioconductor package[46]. The raw peptide intensities were normalised by median-scaled protein histone H4 (Xelaev18026404m.g) intensities. Statistical analysis of differentially expressed protein was carried out using Limma moderated *t*-test.

**Protein extraction and WB analysis.** For protein extraction, 80 μl of EB (500 mM Tris pH6.8, 500 mM NaCl, 1% NP40, 0.1% SDS, 1% beta-Mercaptoethanol and PI cocktail) was used to homogenise 1 million of sperm per sample. Lysates were centrifuged for 10 min at 4 °C and 16,000 g. The supernatant was transferred to a fresh tube, mixed with 20 μl of 5× loading buffer (0.3 M Tris-HCl pH 6.8, 50% (v/v) glycerol, 1% (w/v) SDS, 0.05% (w/v) bromophenol blue and 3.2% (v/v) beta-mercaptoethanol) and boiled at 100 °C for 5 min. All samples were centrifuged for 3 min at RT (16,000 g). The supernatant was then transferred to a new tube. Polyacrylamide gels (12% SDS) were loaded with 10–40 μl of this supernatant. Gel electrophoresis and western blots were performed according to standard protocols. For blotting PVDF membranes, a semi-dry transfer system was used (30–40 min, 25 V). For the protein detection, following primary antibodies were used: anti-H3K4me3 (1:1000, Abcam, ab8580), anti-H3K27me3 (1:1000, cell signalling, 9733), anti-H4 (1:1000, Abcam, ab31830), anti-H3 (1:1000, Cell signalling, 14269), anti-H2A (1:1000, Millipore, 07-146), and anti-H2B (1:2000, Abcam, ab1790), anti-XHMGN2 (1:1000, kind gift from Dr Robert Hock), anti-HMGB1 (1:1000, Sigma, H9664). Blots were incubated with primary antibodies overnight at 4 °C. For the secondary reactions, anti-rabbit (1:25000, Thermo Fisher, A21076) or anti-mouse IgG Alexa Fluor 680 (1:25000, Thermo Fisher, A21058), Goat anti-rabbit Alexa 800 (1:25,000, Thermo Fisher, A32735), IRDye 800CW Goat Anti-Mouse IgG (H + L) (LI-COR, 926-32210) or anti-rabbit IgG Horseradish peroxidase conjugated antibodies (Thermo Fisher, #31460, Cat #31466) were used and incubated for 1 h at RT. The signals were captured using the imaging system Odyssey (LI-COR) or X-ray film after incubating with ECL Western Blotting Detection Reagents (GE healthcare, RPN2109).

**Sequencing data processing.** See supplementary methods and Supplementary data 9.

**Reporting summary.** Further information on research design is available in the Nature Research Reporting Summary linked to this article.

## Data availability

The data that support this study are available from the corresponding author upon reasonable request. The *Xenopus laevis* ChIP-seq have been deposited on GEO with accession number GSE125982. Customised codes and scripts are available on GitLab. The mass spectrometry proteomics data have been deposited to the ProteomeXchange Consortium via the PRIDE[47] partner repository with the dataset identifier PXD012853. Publicly available data were obtained from GEO with the following accession numbers: GSE75164, GSE76915, GSE73430 and GSE100272. The source data underlying Figs. 1a, b, d, e, 3a, b, d, e, 4c and 7b, e and Supplementary Figs. 1a–c, 2a, c, g, 3c, d, 4c, e and 8a are provided as a Source Data file. Source data are provided with this paper.

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

## Acknowledgements

J.J. is funded from a Wellcome Trust grant (101050/Z/13/Z). M.O is funded from a Wellcome Trust (101050/Z/13/Z), MRC (MR/K011022/1) grant and postdoctoral fellowship from the Japan Society for the Promotion of Science (JSPS). A.S. is funded from a Wellcome Trust (101050/Z/13/Z) and MRC (MR/K011022/1) grant. All members of the Gurdon group acknowledge the core support provided by the Gurdon Institute core grant from Cancer Research UK (C6946/A14492) and the Wellcome Trust (092096/Z/10/Z). A.O.D. is supported by a grant from the Science Foundation Ireland Industry Fellowship (SFI:15/IFB/350). C.D.S. and V.N.R.F. are supported by Cancer Research UK. The Fusion Lumos Orbitrap mass spectrometer was purchased with the support from a Wellcome Trust Multi-user Equipment Grant (Grant #108467/Z/15/Z). This study was supported by the National Institutes of Health (R01-GM115945) to A.J.R. We would like to thank the following people for providing reagents used in this study: Dr Julian Blow (pET28 Delta-Geminin plasmid), Dr Atsuya Nishiyama (help with geminin purification), Dr Robert Hock (anti-XHMGN2 antibody) and Dr Thomas Jenuwein (anti-H3K27me3 antibody).

## Author contributions

Conceptualisation: J.J.; methodology: J.J., M.O., M.T., A.S., T.F., L.D. and C.D.S.; validation: J.J., M.O. and A.S; formal analysis: A.S. and K.K.; investigation: M.O., J.J., E.H., M.T., V.G., E.F., M.S. and V.N.R.F.; resources: A.O.D., D.K., A.T.G. and A.J.R.; data curation: C.R.B. and A.S.; writing original draft: J.J.; writing review and editing: J.J., M.O., A.S., E.H., A.O.D., A.J.R., J.G. and V.G.; visualisation: J.J., M.O. and A.S.; supervision: J.J.; project administration: J.J.; funding acquisition: J.J., M.T. and J.G.

## Competing interests

The authors declare no competing interests.
