## [Peer Review File · Nature Communications]

Reviewers' comments:

Reviewer #1 (Remarks to the Author):

Please see the file attached (Word file).

Reviewer #2 (Remarks to the Author):

Nature Communications manuscript NCOMMS-19-01134-T

Investigation of epigenetic landmarks in sperm histones has been a great interest in the field not only for understanding the principles of histone-protamine exchange but also for elucidating their potential transgenerational effects. There have been several studies reporting the unique features of sperm-retained histones mainly in mammals, and some of the studies have demonstrated the enrichment of sperm histones in gene coding regions while others proposed the opposite. In addition, inter-sperm heterogeneity of histone marks is never investigated so far despite of its biological importance.

In this study, authors successfully “quantified” the percentage of retained histones at certain loci in frog and human sperm by using IceChIP-seq. They observed highly methylated domains (HMDs) of either or both H3K4me3 and H3K27me3 in certain sets of genes including ZGA, suggesting the uniform retention of histones in these genes, and also a possibility of faithful transmission to the progeny.

【Major concerns】

1) One of the major concerns is quantification of retained histones by IceChIP. It may not be very surprising that modified H3s (i.e. H3K4me3 and H3K27me3) are uniformly retained at high levels in certain promoters because i) in frog sperm, H3 and H4 are escaped from histone-protamine exchange, thus almost 100% of H3/H4 are maintained in the sperm chromatin as shown in Fig. 1A, and ii) using MNase for sperm chromatin solubilization causes the artificial enrichment of histones at TSSs in mice (Carone et al., 2014, and Yamaguchi et al., 2018). On the other hand, IceChIP sometimes exhibits the value (HMDs) of >100% (Fig. 4, Fig. 7). This problem was also pointed out in the original report by Grzybowski et al., 2015. Although authors in this study speculated that it is due to the cross-reactivity of anti-H3K27me3 antibody to H3K27me2, this over-representation appears quite problematic especially in Fig. 4C, as HMDs of H3K27me3 are over 500%, suggesting that ~80% (400/500) of the signal was non-specific. Because this data is critical to support authors' claim, changing the antibody to more specific ones or other experimental validation are essential. This antibody issue may be related to other problem described in 3) below.

2) Another important finding in this study is to demonstrate the retention (or re-establishment) of sperm histone marks in embryos even after DNA replication. If I understood the experimental procedure correctly, Fig. 5B-D are the comparison between “untreated sperm” and “sperm with chromatin assembly and 1X replication in egg extract”. If so, is certain normalization required to compare the HMDs between haploid and diploid chromatin? In addition, according to the Method, authors treated all the frog samples (i.e. sperm, egg extract-treated sperm, and blastula) with the same MNase condition (2.5 U, 37 C, 30 min). It might be too harsh for egg extract-treated sperm and blastula, as their chromatins are somatic-like, thus the MNase treatment can cause over-digestion (and it could be the reason why blastula has only 1/2 of H3K4me3 peaks in Fig. 5F). Validation of proper chromatin extraction and DNA fragment size are needed.

3) In the entire manuscript, peak pattern of H3K27me3 seems broader compared to previous studies by other groups. In Fig. 3E, for instance, only cluster 5 exhibit peak-like enrichment at

TSSs, but it's not obvious in other clusters. Also in Fig. 3E (right), how was the relative enrichment calculated, as if cluster 1, 2, and 3 were compared, cluster 2 possess more H3K27me3 according to the heatmap, but it wasn't clear in the histogram? And by judging from Fig. 3E, there is no gene with H3K27me3 alone; all the genes were clustered either in H3K4me3 alone or H3K4me3/K27me3 bivalency. Is it really so, or could it be due to the failure of peak calling of H3K27me3?

4) Unlike sperm of model animals, there must be substantial person-person and sperm-sperm heterogeneity in human in terms of morphology and likely other factors, although authors mentioned that they were donated by fertile men, and their sperm passed quality checks, and are purified by Percoll gradient method. This could be one of the causes of quite low % of HMD>80% in human sperm (Fig. 7A) compared with those in frog (Fig. 3AB). Is it possible for authors to provide data how homogeneous their human sperm sample is? What if they analyzed the individual sperm specimen without pooling them?

5) Are there differences of H3K4me3 and H3K27me3 HMDs between nucleosome and subnucleosome? If the nucleosomes and subnucleosomes were analyzed separately, what do the Fig. 3A (left) and Fig. 3B look like?

6) In Fig. 3D, authors listed transcription factor-coding genes related to embryonic development and looked at the enrichment of H3K4me3 and H3K27me3. They also divided the TFs to maternal factor and non-maternal/no evidence-of-origin factor. I could not understand authors' intention to perform this analysis. Is it also related to Fig. S7 (maternal gene regulation)?

7) Related to Fig. 4, 6, and partly 7, Erkek et al., 2013 previously reported the transgenerational effect of sperm H3K27me3 on gene expression in early embryos, while authors in this study emphasized the implication of H3K4me3/H3K27me3 bivalency in transgenerational regulation during early embryogenesis including ZGA. When H3K27me3 alone (if it existed) and H3K4me3/H3K27me3 bivalency are compared, which one is more predominantly retained at developmental genes in this study? Because according to Table S4, H3K27me3 also exhibits significant enrichment in developmental genes, some of which are more significant compared to those in H3K4me3/K27me3 bivalency. In addition, according to Fig. 4, it is still unclear to me whether these modified histones are preferentially retained in nucleosomes or subnucleosomes.

8) By comparison between frog and human data, authors discovered that enrichment of HMDs in developmental genes including ZGA are commonly observed in both species, while no common (= ortholog) genes were detected. Is it because the gene profile composing ZGA is different between frog and human? Please explain.

【Minor concerns】

9) # replication of each ChIP-seq samples were not described in the manuscript.

10) In Fig. S1A, western blot of H4 lacks linearity (whereas the one in S1B looks good).

11) Fig. 1E, lower panels (bar graphs). Please indicate error bars if authors performed the experiment more than three times.

12) Related to Fig. 3A, authors mentioned that "these sites have histone methylation in most sperm and cover a fraction of the genome similar to that found in an ESC population, suggesting that modified histones might have functional relevance in sperm as is the case for ESC". What kind of genes are in common between frog sperm and mESCs?

13) In Fig. 4A, fold enrichments are calculated by $HMD_{>80}/HMD_{<80}$, not by $HMD_{>80}/total$. I assume $HMD_{>80}/HMD_{<80}$ makes the difference more obvious, but is it really appropriate?

14) What is the exact property of "None" in Fig. S5, as there must be certain types of histones. Please specify.

15) In line 389-392, authors introduced ref# 13 and 32 after the sentence of "modified histones cannot be the basis for the epigenetic information required for embryo development". However, at least #32 demonstrated the enrichment of H3K4me3 at developmental genes, and also presented the consistency of their data with Erkek et al., 2013. Thus, putting #32 here seems irrelevant. Additionally, as authors may realize, there is a report demonstrating the reset of paternal H3K27me3 at fertilization in mice (Zheng et al., 2016, PMID: 27635762). Some discussion is highly recommended.

Reviewer #3 (Remarks to the Author):

This paper examines the epigenetic states of mature sperm DNA using omics approaches. The authors suggest that 1) the programming of sperm genes for embryonic development is associated with the formation of chromatin regions of homogeneous epigenetic constitution, and that 2) there is a conservation of sperm epigenetic programming mechanisms between human and frogs. Unfortunately, the paper suffers from a number of issues. While the paper makes some interesting observations, these observations are often not clearly explained and not integrated well in the text. In addition, this paper does not integrate other relevant literatures. Consequently, the paper suffers from the lack of clarity as described below.

Ln 66-67: "In *Xenopus laevis*, the deposition of protamines during spermiogenesis is associated with only a partial loss of some of the core histones." Does this mean that one can have less than an octamer? Or does it mean that the lost core histone is just replaced with a variant histone subunit?

Ln 66-67: "This feature places this vertebrate in an intermediate position between the situation found in mammals where the majority of histones are replaced, and zebrafish sperm that do not lose any histones", citing reference 7. If fish don't lose histones, and have respectably sized genomes, what's the reason to do this increasingly more as animals move to amphibia and then eventually mammals? Cairns paper (ref 7), describes that H3K27me3 and H3K4me3, among other marks, are found on zebrafish sperm developmental genes. This suggests that the frog story herein might be evolutionarily conserved with zebrafish. Is this correct?

Ln 104-106: "Two additional DNA fragments with a size of ~70 bp and ~110 bp appear specifically after digestion of sperm chromatin." In Fig1B, XL177 seems to generate DNA fragments corresponding to 110bp and 70bp, albeit weakly. Is it possible that sperm chromatin produce these smaller fragments simply by "overdigestion" of sperm sample, relative to the somatic sample? Would a longer digestion or a higher concentration of MNase produce these fragments from somatic chromatin?

Ln128-131, Figure 1E and Figure S1. It is not clear to me how one can conclude the stoichiometry shown in histograms in Fig1E. The H3/H4 ratio does not equal 1 in ESC extract, for example. Intensity of H4 is significantly higher than that of H3 in the top panel. In the 150bp particles, which are claimed to be "standard" nucleosomes with an H3-H4 tetramer, the ratio seems to be around 0.6. Does this mean it is nearly twice as much H4 as H3? Fig1F suggests a loss of an H2a/b dimer from the standard histone octamer and the data (Fig1E) does not seem to match the model. Additionally, the molecular weights of H4 protein are not identical between purified particles relative and whole extracts. Is this a gel artifact?

Ln 154-156: "When taking into account the length of DNA protected by each type of particles, this indicates that the majority of the sperm genome (66%) is protected by nucleosomes".

Does this say a majority of genome is protected by these classes in these ratios? Or does it say that, OF THE DNA THAT IS PROTECTED, it's in these ratios? Is the entire genome inside nucleosomes or is some of it is unprotected as it's in between nucleosomes? Please clarify.

Ln 177-178. The statements in the paper like "Interestingly, an enrichment for function related to spermatogenesis is observed for cluster 2" are difficult to verify because the suppl table isn't laid out in a way that makes this obvious. Table S2 contains a list of genes for cluster 2 and another tab with GO terms for cluster 2. How can I determine from this list that spermatogenesis genes are enriched? There is another tab called GO filtered cluster 2. Is this what I should be looking at? There is no explanation provided to the reader. All data suppl files needs to be made more transparent and accessible.

Ln 188-190: "Our results so far indicate that some sperm histone H3 containing particles have the required attributes to act as carrier of epigenetic information to the next generation as they are found to be retained at the same genomic location within a sperm cell population." This statement is a bit unclear. The patterns in Fig2D seem to indicate that large blocks of DNA are held within certain categories of subnucleosomal particles. The phrase "genomic location", in the abstract and throughout the text, gave me the impression that the paper was going to be talking about phased nucleosomes. In other words, nucleosomes that sit on very precise sites relative to gene enhancers/promoters etc. But Fig2D doesn't show this. It seems to show that large blocks/DNA regions are decorated with specific nucleosomal types in various patterns. I think a change in terminology is needed to more precisely convey what is observed and to avoid confusion with a nucleosomal phasing type of model. The depicted figures also do not help clarifying this concept.

Ln194-196: I imagine there is some variance in the pipetting and quantifying when mixing sperm chromatin with methylated nucleosomal ladder standard for this quantitation. What is this error range?

Ln207-209: "...importantly, are enriched for binding motifs recognized by transcription factors implicated in early embryonic development (i.e. NFY-A/Dux 22; Ascl1 23, ZFP281 24; Figure 3D)." There is no dux gene in frogs. Since frog uses human gene/protein symbols, zfp281 should be znf281. Most of the znf281 study has been reported in mammalian pluripotency, not Xenopus. Since Nfya and Dux belong to completely different transcription factor (TF) DBD families, its more likely that the motif found corresponds to Nfya, or maybe some other HD protein, but not Dux. Additionally, there is little effort to interpret these findings other than to say 3 out of the 26 motifs might represent TFs known to be involved in early development in some vertebrate. Lastly, what is the statistical significance of the motif analysis? I ask this question because motif analyses frequently use a collection of more narrowly defined DNA regions. However, in this analysis, the authors seem to use much broader DNA fragments. What is the evidence that such an analysis can generate a statistically meaningful output?

Ln 227-231 Fig4A: By looking at histogram for K27me3, it is unclear whether 1.1, 1.05 and 1.0 are statistically significantly different. Please show p values.

Ln 254-258, Table S4, Fig4C. Is cluster 6 the only cluster with developmentally interesting genes? How many genes are in this category? Perhaps the authors can state it in the text. I also don't fully understand the way this table is organized. For instance, while there are 2450 entries in cluster 6, it is unclear how many of these are the same as another entry. For instance, what is the difference between ZNF800_Kwon201107_XENLA_00069622 and ZNF800_Taira201203_XENLA_tissue_00006407? How about ZNF536_Taira201203_XENLA_tissue_00229075 and ZNF536_Taira201203_XENLA_tissue_00250614?

Fig4C: I don't think the authors referred to this panel in any significant way and I do not know the

significance of the panel.

Ln284-286 "For that purpose, we performed ChIP-seq analysis in early blastula embryos (after ~10 embryonic cell cycles but before the activation of zygotic transcription)." In *Xenopus*, 10 cycles is around the middle of stage 9 and transcription has started earlier based on various RNA-seq data. While this does not change the overall conclusion of the paper, correct stages should be reported.

Fig5F: 97% of K4me3 blastula peaks (13,161 vs. 422 peaks) are in sperm. However, only 37% of sperm peaks (13,616 out of 36,015 peaks) are still present in blastula. What is the interpretation of this result?

Ln300-302 and subsequent paragraphs. I'm confused. Fig6A seems to show that developmental genes at ZGA are not bivalent as bivalency is on "all genes" as these are marked as "K27 homogeneous & retained" and "K4 homogeneous & retained". I also do not fully understand this entire section. In Fig4B the authors state that bivalent is where the developmental genes are, in cluster 6. Then, in paragraph below, on this page, the authors start again talking about ICeChIP finding. I am unable to follow the argument.

Ln312-313: "Genes that are misregulated upon demethylation of H3K27me3 (Kdm6b sensitive)" How was this experiment done? The experimental approach is not in materials and methods. However, based on reference 2, I assume that the mRNA injections were done in 1-cell stage embryos. If so, the statement that the authors are looking at interference with H3K27 or H3K4 methylation "at fertilization" is not possible as the injections are occurring post fertilization (and also there must be some lag between mRNA injection and build up of translated protein).

Ln317-318: "Genes sensitive to demethylation of H3K27 or H3K4 at fertilization are enriched for such bivalent genes." The authors say that developmental genes are bivalent, but that's not what's shown in Fig6A.

Ln319-320: "...depletion for the maternally provided factor Ascl1 harbour homogeneously methylated H3K27 on the binding site for this transcription factor (Figure S7). This suggests that sperm-provided modified histones might regulate maternal factor activity". It seems that the authors are implying that Ascl1 is driving H3K27me3 here. And how does this suggest that sperm-provided modified histones regulate Ascl1? Isn't it more likely that the maternal factor is affecting the retention of H3K27me3 on the sperm? I don't fully understand the authors' model and the description is vague.

Ln 358-359 "However, in both species the set of genes with high H3K27me3 HMD are related to developmental functions." Previously, the authors stated that there are no conservation and now it is stated, "however in both". Please clarify your idea.

Ln 360: How is znf281 implicated in early development? What system and how? It seems the protein controls pluripotency in mice and maybe nodal signaling later.

Ln 375-376: "We found that high HMD peaks (HMD>80) are indeed more likely to be associated with such unmethylated sperm DNA than low HMD peaks (0<HMD<80)." Are these data statistically significant? Doesn't the comparison of signal strength (Y axis values) in Fig7E right vs. left suggest that all categories of HMD are more highly associated with no meC than regions with meC? It also seems that HMD 0-20 is as enriched in "no meC"/"meC" as the HMD>80 category. The text related to this figure panel is vague.

Ln378-379: "Additionally, genes with high HMD of H3K27 in sperm are enriched for genes that have a closed TSS configuration". The statement, "genes with high HMD are enriched for genes", do you mean "the gene set with high HMD"?

Ln 382-383: "We conclude that in human sperm, histones near some developmental genes are always retained in their methylated form." No gene numbers were given. How many genes are like this?

Discussion: There is no attempt to explain how the data in this paper fits together with Akkers et al 2009 paper that talked about acquisition of H3K4me3 and K27me3 post ZGA, which doesn't coincide with the current story. What's the relationship between the observations here and their findings? Sperm marks presumably dissipate during cleavage and then re-occur in late blastula. That is also what was reported in the Cairns zebrafish paper cited herein as ref.7. If so, what would be the notable findings of this paper?

Ln908-909, "ChIP was performed as described previously with the following modifications. Blastula (stage 7) embryos were generated by in vitro fertilization." The authors stated that they have used embryos after 10 divisions. If this is what they used, stage 7 is incorrect.

The figure legend is kept to a minimum. This is acceptable only if the main text describes each figure carefully. However, the current description of individual figure panels is insufficient for the reader to fully comprehend these figures.

Sometimes error bars and the statistical significance of data are missing, and it is unclear how many times each experiment was repeated.

Ln 1068-1070. "Data Availability: The *Xenopus laevis* ChIP-seq data reported in this paper will be deposited on GEO once the manuscript is accepted for publication. Accession number will be provided in this section." I was unable to determine the quality of their data.

Reviewer #1 Comments:

In this manuscript, Oikawa et al. asked how histone modifications in sperm contributes to the embryonic development. The authors identified the H3K4me3 and H3K27me3 peaks at the specific loci in the *Xenopus* sperm genome, which appears to be maintained during early embryonic replication. They also showed that embryonic developmental genes are enriched in such methylated histone fraction resisting postfertilisation reprogramming. Essentially, this study can contribute to understanding the mechanism of paternal epigenetic information transmission to the embryo which is now interesting topics. However, I have some criticisms shown below, which the authors need to reply.

Major concerns

1. Confusion by using the phrase “the retention of methylated histones --- in every sperm cell” and “homogeneously methylated histone” in ABSTRACT and Text. Such phrase may induce misunderstanding that the data were obtained using single-cell analysis. The authors probably used such phrase as the opposite phrase of “random distribution of histones”, which was used to state the histone distribution without peak at the gene-poor intergenic region in the previous works. However, the authors cannot conclude that such histone modifications are localized in every sperm without single-cell analysis. I would recommend to use other phrase, such as “the histone localization at the specific loci”.
2. Author tried to estimate the level of methylated histone by quantitative ChIP-seq, ICe-ChIP-seq. HMD computed by input/IP counts of spiked tags should be typically 0-100% (Grzybowski, A.T. et al., Mol. Cell, (2015)). In some raw ChIP-seq data (e.g. Fig. 4C), it's out of the range. What's the reason? (this is closely related to reliability of quantification of histone methylation level) Author says “the retention of methylated histone H3 at the same genomic location in every sperm cell”, but it seems to be overstatement from only the observation of “HMD>80%”. Some experiments at the single cell level are required to say it.
3. Two controversial results of MNase-seq in mouse sperm were reported; distributed in promoter vs along intergenic regions (Saitou, M. & Kurimoto, K., Dev Cell (2014)). It may be caused by different dose of MNase. Author used relatively higher dose of MNase for frog sperm (2.5U per 10 M cells) relative to previous reports for mouse sperm. How they decided to use this concentration? The efficiency of histones (H3) solubilization was checked after MNase treatment like DNA amounts (Fig. S2A)?

4. Recent report using mouse sperm indicate the fraction of swim-up sperm contains about 10% of immature sperm (Yoshida, K. et al., Nat Comms (2018)). Authors have used motile sperm as mature human sperm. The population of mature/immature sperm should be examined by FACS (Evenson, D. et al., Methods Mol. Biol. (2013)), because immature human sperm may contain much more amounts of histones than mature, like mouse. If so, histone ChIP-seq data using immature sperm-contaminated sample should reflect histone profile in immature sperm.

Minor points

1. Line 55: The fraction of mouse mature sperm (Histone-replace completed sperm) after removal of immature sperm contains only 0.3% of histone (Yoshida, K. et al., Nat Comms (2018)).
2. Fig. 1A: Cleaved form of H3 can be detected in mouse sperm (Yamaguchi, K. et al., Cell Rep. (2018)). Similar form was also observed in frog sperm? How each histone were quantified and normalized? Presentation for raw data of western film may be better.
3. Fig. 2A: The nucleosome/subnucleosome genomic region was judged by insertion size of PE 36bp sequence data. Distribution of insert size in the data may be important to claim the reliability.
4. Fig. 2C: What's the definition of enhancer regions (from previous data)?
5. Fig. 2E, 3E: Labels along x/y-axis were needed in some heat maps.
6. Fig. 2E: What's the evidence of nucleosome distributions in spermatid. Proper reference paper or their speculation should be described.
7. line 374: In addition to high HMD peaks, low HMD peaks (HMD 0-20) also seem to be associated with un-methylated sperm DNA.
8. Fig. 7G: The proper explanation of cartoon should be described in figure legend. In *Xenopus*, "all" gene regulatory regions on genome has nucleosome or sum-nucleosome

with histone modification? In human, nucleosome positions around the regulatory regions? However, author didn't perform MNase-seq using human sperm.

Revision of "Epigenetic homogeneity in histone methylation underlies sperm programming for embryonic transcription"

Please find below a point by point answer to reviewers' comments. In order to make it easier to survey, we have highlighted each of the raised points in bold and our answers in regular font type.

Reviewers' comments:

Reviewer #1:

In this manuscript, Oikawa et al. asked how histone modifications in sperm contributes to the embryonic development. The authors identified the H3K4me3 and H3K27me3 peaks at the specific loci in the *Xenopus* sperm genome, which appears to be maintained during early embryonic replication. They also showed that embryonic developmental genes are enriched in such methylated histone fraction resisting postfertilization reprogramming. Essentially, this study can contribute to understanding the mechanism of paternal epigenetic information transmission to the embryo which is now interesting topics. However, I have some criticisms shown below, which the authors need to reply.

Major concerns

1. Confusion by using the phrase “the retention of methylated histones --- in every sperm cell” and “homogeneously methylated histone” in ABSTRACT and Text. Such phrase may induce misunderstanding that the data were obtained using single-cell analysis. The authors probably used such phrase as the opposite phrase of “random distribution of histones”, which was used to state the histone distribution without peak at the gene-poor intergenic region in the previous works. However, the authors cannot conclude that such histone modifications are localized in every sperm without single-cell analysis. I would recommend to use other phrase, such as “the histone localization at the specific loci”.

To avoid possible confusion we have modified the text as follow:

- We mention calibrated ChIP directly in the abstract. **Page1 Line 19-22** “*Using calibrated ChIP-seq, we show for the first time that remodelling of histones during spermiogenesis results in the retention of methylated histone H3 at the same genomic location in most sperm cell*”

-We modified the abstract by changing “...same location in every sperm cell” to “...same location in *most* sperm cell” as this indeed reflect better the conclusions obtained from our analysis. **Page1 Line 21-22**

-Lastly, we include a statement indicating that “*Using quantitative ICeChIP-seq¹⁵ on bulk sperm sample we could infer that at certain genomic loci histone H3 is methylated on K4 and/or K27 in most sperm.*” **Page 3 Line 74-75**

2. Author tried to estimate the level of methylated histone by quantitative ChIP-seq, ICe-ChIP-seq. HMD computed by input/IP counts of spiked tags should be typically 0-100% (Grzybowski, A.T. et al., Mol. Cell, (2015)). In some raw ChIP-seq data (e.g. Fig. 4C), it's out of the range. What's the reason? (this is closely related to reliability of quantification of histone methylation level).

The calibration approach used in the ICe-Chip-sequencing technique relies on the signal measured across a given ladder of semi-synthetic nucleosomes. Specifically, after pull-down and input processing, the spiked-in sequences corresponding to the ladder for a given individual histone modification are used to estimate the pull-down efficiency (E_{ff}). Such efficiency term is then used to rescale and normalize the signal measured genome-wide or at individual peaks (enrichment sites).

The formula proposed in Grzybowski *et al.*, Mol Cell, 2015 and used also in this study is:

$$HMD = 100 * \frac{C_{IP}}{C_{INPUT}} * \frac{1}{E_{ff}}$$

Where:

- C_{IP} is the coverage in the pull-down experiment
- C_{INPUT} is the coverage in the input experiment.

When introduced, the HMD is an estimate of the global level of methylation at a specific locus across all the cells – chromatin – available. This means that HMD=0 represent complete lack of methylation, HMD=100 represents the case when, in the specific locus, all the probed chromatin show methylation and finally HMD=50 would reflect an intermediate scenario when only half of the chromatin used is methylated. This last case can also be interpreted in terms of cells across the sample screened: half of the cells reports the histone modification at the specific locus.

The behavior of this formula at the extreme cases is as follow:

- HMD=0

Such case can be generated easily when $C_{IP} = 0$ *i.e.* when nothing is measured in the pull down experiment.

-HMD=100

To have this case, it is necessary that the fraction $\frac{C_{IP}}{C_{INPUT}}$ equalize exactly E_{ff} .

So HMD=100 is reached when the ratio of the coverages equals the efficiency.

-HMD>100

Exceeding 100 happens technically when IP is E_{ff} times higher than the input. This can happen when the efficiency of the pull down of sperm chromatin is higher than the efficiency of the ladder pull-down. There are a number of possible explanations for this occurrence, inherent to the ICe-ChIP procedure (Shah *et al.*, Mol Cell 2018; Grzybowski *et al.*, Mol Cell, 2015). One would be errors in input or IP sampling. Another one would be the sensitivity of antibody pull down efficiency to neighboring histone modifications or nucleosome composition. We have ruled out that the antibodies used in ICe-chip present a differential pulldown efficiency to modified H3 associated to the different chromatin particles characteristic of frog sperm (**Figure S3 C-D**), and see reviewer 2 point 1 for further details). This is also reflected in the fact that the range of HMD values for H3K4 and H3K27 methylation are in the same

range when occurring within the context of nucleosome or in the context of subnucleosome (**Figure S5 E-F**).

Most importantly we now provide a more detailed analysis of the HMD distribution and we can show that HMD exceeding 100% represent a minor fraction of the data (**Figure S3 E, G-H**). This indicates that most values obtained using this quantification strategy are within a biologically interpretable range, therefore validating the approach deployed in this work. Lastly the range of HMD value generated on sperm in this work is very similar to that obtained in a previous study using ES cell and different antibodies against the same histone modifications (Gryzbowski *et al.*, Mol Cell, 2015) **Figure S3 F, I-J**.

To avoid confusion we define HMD as “apparent histone methylation density” (**Page 6 Lines 196-197**) to take into account the fact that in a limited number of instances the HMD values can exceed 100.

Please see reviewer 2 point 1 for additional details about HMD levels.

Author says “the retention of methylated histone H3 at the same genomic location in every sperm cell”, but it seems to be overstatement from only the observation of “HMD>80%”. Some experiments at the single cell level are required to say it.

Unfortunately, single sperm ChIP analysis with sufficient genome sampling is not technically possible at the moment. Hence, to overcome this limitation our strategy is to use ICe-ChIP as a proxy for single sperm ChIP and to quantify the epigenetic conformation of a given locus at a cell population level. We agree with the reviewer that we cannot conclude from the ICe-ChIP data that *all* sperm have a modified histone at a given locus, rather we can infer based on high HMD value that a given genomic location harbor a modified histone in *most* sperm. We have modified the text accordingly (*i.e.* **Page1 Line 21-22** “... results in the retention of methylated histone H3 at the same genomic location in most sperm cell” ;**Page 3 Line 74-75** “...we could infer that at certain genomic loci histone H3 is methylated on K4 and/or K27 in most sperm”). This conclusion is in line with the measurement of the developmental potential of sperm. Indeed ICSI experiment in various species also lead to the conclusion that *most* sperm are competent to support development (*i.e.* Morozumi *et al.*, PNAS 2006).

3. Two controversial results of MNase-seq in mouse sperm were reported; distributed in promoter vs along intergenic regions (Saitou, M. & Kurimoto, K., Dev Cell (2014)). It may be caused by different dose of MNase. Author used relatively higher dose of MNase for frog sperm (2.5U per 10 M cells) relative to previous reports for mouse sperm. How they decided to use this concentration? The efficiency of histones (H3) solubilization was checked after MNase treatment like DNA amounts (Fig. S2A)?

As mentioned in our introduction frog sperm is rather different from that of a mouse. In our previous work we tested a range of MNase digestion conditions and determined optimal histone solubilization conditions specifically for frog sperm chromatin (Teperek *et al.*, genome research, 2016).

Indeed, using such MNase treatment, histone H3 is efficiently solubilized, in the same range as the solubilization of DNA (**new Figure S2 C-D**). See also reviewer 3 point 3 for a detailed explanation of MNase treatment protocols.

4. Recent report using mouse sperm indicate the fraction of swim-up sperm contains about 10% of immature sperm (Yoshida, K. et al., Nat Comms (2018)). Authors have used motile sperm as mature human sperm. The population of mature/immature sperm should be examined by FACS (Evenson, D. et al., Methods Mol. Biol. (2013)), because immature human sperm may contain much more amounts of histones than mature, like mouse. If so, histone ChIP-seq data using immature sperm-contaminated sample should reflect histone profile in immature sperm.

Mouse is a very extreme case of histone removal (99.7%, Yoshida, K. *et al.*, Nat Comms (2018)) compared to human (90%, Hammoud *et al.*, Nature 2009). To our knowledge there is no report of the presence of immature “histone rich” sperm in human as it has been shown in mouse. As a consequence, there is no protocol to perform this analysis in human. Our ICe-CHIP analysis combined with published sperm DNA methylation profiles and embryos ATAC-seq data (Figure 7) indicates that the identified methylated histones loci in human sperm reflect the situation in most sperm and are therefore unlikely to result from a putative subpopulation of immature sperm in our sample.

Minor points

1. Line 55: The fraction of mouse mature sperm (Histone-replace completed sperm) after removal of immature sperm contains only 0.3% of histone (Yoshida, K. et al., Nat Comms (2018)).

We thank the reviewer for pointing out this updated quantitation. We corrected the numbers and included reference to Yoshida, K. *et al.*, Nat Comms (2018).

2. Fig. 1A: Cleaved form of H3 can be detected in mouse sperm (Yamaguchi, K. et al., Cell Rep. (2018)). Similar form was also observed in frog sperm? How each histone were quantified and normalized? Presentation for raw data of western film may be better.

Raw data showing the full scan are now included in the figure (**Figure S1**). We have found no evidence of cleaved histones in our analysis. Histones were normalized using relative signal intensity in quantitative WB analysis (LICOR). Additionally, to exclude the possibility that histone variant not recognized by our antibody would explain the apparent loss of H2A/H2B in sperm, we performed unbiased (*i.e.* non antibody dependent) quantitative Mass Spectrometry analysis (**Figure 1D** and **Table S1** and **dataset PXD012853** on the PRIDE repository)

3. Fig. 2A: The nucleosome/subnucleosome genomic region was judged by insertion size of PE 36bp sequence data. Distribution of insert size in the data may be important to claim the reliability.

Our unsupervised assessment of nucleosome and subnucleosome enrichment uses a Hidden Markov Model. Our methodology takes into account the size and local distribution of fragments in the input samples (see M&M section **Page 35 Lines 1194-1222**).

We now include the histogram of fragment sizes in frog sperm input samples: 3 main sizes are observed around 70, 110 and 150 bp (**Figure S2 F**).

4. Fig. 2C: What's the definition of enhancer regions (from previous data)?

Enhancer definition is based on a published embryonic p300 ChIP-seq dataset (Session *et al.*, nature 2016). GEO number: [GSE76059](https://www.ncbi.nlm.nih.gov/geo/query/acc.cgi?acc=GSE76059)

5. Fig. 2E, 3E: Labels along x/y-axis were needed in some heat maps.

We apologize for the missing information on these panels.

We have updated the heatmap including better labeling of the axes (**Figure 2E&3E**)

6. Fig. 2E: What's the evidence of nucleosome distributions in spermatid. Proper reference paper or their speculation should be described.

In our previous work we have shown that unlike the sperm but similar to somatic cell (in that case XL177 cells), MNase digestion of spermatid only produces 150 bp fragments (Teperek *et al.*, Genome Research, 2016).

Figure S5 from Teperek *et al.*, 2016:

7. line 374: In addition to high HMD peaks, low HMD peaks (HMD 0-20) also seem to be associated with un-methylated sperm DNA.

This is indeed the case that a similar fraction of high (>80) and low (<20) HMD fragments are associated with unmethylated DNA (~70%). We observe a significant enrichment for HMD>80 peaks with region of unmethylated DNA when compared to peak in the HMD 20-80 range but not when comparing to those in the low range (HMD 0-20). One explanation for this phenomenon could be that we are focusing on low level of H3K4 or H3K27 methylation but are blind to other chromatin modifications that might drive the exclusion of DNA methylation.

8. Fig. 7G: The proper explanation of cartoon should be described in figure legend. In *Xenopus*, “all” gene regulatory regions on genome has nucleosome or sum-nucleosome with histone modification? In human, nucleosome positions around the regulatory regions? However, author didn’t perform MNase-seq using human sperm.

The cartoon aims at illustrating the general properties of modified histones retained at the same location in most sperm. Hence in *Xenopus* we indicate that modified histone can be retained as nucleosome or subnucleosome and that they tend to be homogeneously retained around gene regulatory regions. The same is true in human (the ICe-Chip data we present does indeed correspond to MNase-seq, see M&M **Page 28 Lines 969-971** ” *Sperm chromatin and semi-synthetic nucleosomes were digested with 2.5 U (*Xenopus*) and 0.5 U (human) of MNase for 30 min at 37°C*”) The data supporting enrichment of homogeneously retained modified histone at gene regulatory regions in human sperm were indeed missing from our initial submission. We thank the reviewer for pointing out this omission. Supporting data are now added as **Figure S9**.

Reviewer #2 :

Investigation of epigenetic landmarks in sperm histones has been a great interest in the field not only for understanding the principles of histone-protamine exchange but also for elucidating their potential transgenerational effects. There have been several studies reporting the unique features of sperm-retained histones mainly in mammals, and some of the studies have demonstrated the enrichment of sperm histones in gene coding regions while others proposed the opposite. In addition, inter-sperm heterogeneity of histone marks is never investigated so far despite of its biological importance.

In this study, authors successfully “quantified” the percentage of retained histones at certain loci in frog and human sperm by using ICeChIP-seq. They observed highly methylated domains (HMDs) of either or both H3K4me3 and H3K27me3 in certain sets of genes including ZGA, suggesting the uniform retention of histones in these genes, and also a possibility of faithful transmission to the progeny.

【Major concerns】

1) One of the major concerns is quantification of retained histones by ICeChIP. It may not be very surprising that modified H3s (i.e. H3K4me3 and H3K27me3) are uniformly retained at high levels in certain promoters because i) in frog sperm, H3 and H4 are escaped from histone-protamine exchange, thus almost 100% of H3/H4 are maintained in the sperm chromatin as shown in Fig. 1A, and ii) using MNase for sperm chromatin solubilization causes the artificial enrichment of histones at TSSs in mice (Carone et al., 2014, and Yamaguchi et al., 2018). On the other hand, ICeChIP sometimes exhibits the value (HMDs) of >100% (Fig. 4, Fig. 7). This problem was also pointed out in the original report by Grzybowski et al., 2015. Although authors in this study speculated that it is due to the cross-reactivity of anti-H3K27me3 antibody to H3K27me2, this over-representation appears quite problematic especially in Fig. 4C, as HMDs of H3K27me3 are over 500%, suggesting that ~80% (400/500) of the signal was non-specific. Because this data is critical to support authors' claim, changing the antibody to more specific ones or other experimental validation are essential. This antibody issue may be related to other problem described in 3) below.

We agree with the reviewer that HMD >100% would exceed the biological expectation. This is why we referred to our measurement as “apparent histone methylation density” (Page 6 Lines 196-197).

However, we recognize the need to better account for the range of apparent HMD obtained in our analysis and we now further characterise HMD distribution in **Figure S3**.

Based on the way HMD is calculated (see reviewer1 point 1) values over 100% can arise from either sparse sampling in the data, or from an increased antibody pulldown efficiency of the chromatin at a specific locus compared to that of the spiked-in synthetic nucleosome reference. We minimized the former by deep sequencing of the data (~7X average coverage of the genome, **Figure S2 E**). The later might occur because of the particular composition of the sperm chromatin at this locus. This could for example occurs when additional histone tail modification around the nucleosome would potentiate the pull down of the targeted histone modification. As such it is not

an indication that the antibody is not specific, rather than its pulldown efficiency might be influenced by other characteristics of the chromatin architecture in addition to the presence of the target antigen.

The observations listed below indicates that antibody *specificity* is unlikely to account for the HMD values >100% in our dataset:

1. When comparing two completely independent ChIP-seq data set (*i.e.* our study *versus* that of Hammoud *et al.*, 2006: two different H3K27me3 antibodies used in different labs, on different sperm samples, using different digestion protocol) we observed a very large degree (77%) of conservation in terms of overlapping peaks (**Figure S8 C-D**).
2. We do not see a correlation between the off-target binding of the antibodies used and the proportion of apparent HMD value >100% (*i.e.* on the panel of modifications tested, H3K4me3 ab seems to have more off target binding than the H3K27me3 antibody (**Figure S3 B**), yet we observe less HMD>100% genome wide for H3K4me3 than H3K27me3 (0,24% and 4,4%see below)). Obviously the off-target binding assessment depends on the panel of modification tested and we cannot exclude that other modifications untested in our panel (*i.e.* H3K27me2) might be picked up by our antibody. Still, our approach goes beyond standard practise of antibody off target evaluation which traditionally uses assay that might not reflect antibody behaviour in ChIP-seq conditions (see Shah *et al.*, Mol Cell 2018 for comparison of peptide array versus ICeChIP).

Another possible explanation for HMD>100% could be that modified histones in subnucleosomal particles would be pulled down with higher efficiency than when in the context of a nucleosome (as is the case for the spiked-in reference). We have ruled out this explanation as:

1. The pull down efficiency in nucleosome and subnucleosome is similar whether using the H3K4me3 or the H3K27me3 antibodies (**Figure S3 C-D**);
2. HMD values are similar in region of the genome that have only nucleosome or have only subnucleosome (**Figure S5 E&F**).

Based on these observations we conclude that the fraction of genomic sites with HMD>100% is neither explained by the lack of specificity of our antibodies nor by the subnucleosomal/nucleosomal chromatin structure typical of frog sperm.

We propose that apparent HMD>100% occurs either when antibody pull down of its targeted modification is potentiated by additional chromatin modifications or because of sampling error in IP and input (see also Shah *et al.*, Mol Cell 2018).

Most importantly we note that HMD>100% represent a minor fraction of cases (*i.e.* 0,4% and 4,4% of the genome for H3K4me3 than H3K27me3 respectively **Figure S3 E**). HMD values within the same range are observed on mouse ES cell (dataset from Grzybowski *et al.*, 2015) (0,04% and 4,35% of the genome with HMD>100% for H3K4me3 than H3K27me3 respectively **Figure S3 F**).

Genome wide distribution of HMD values is similar in frog sperm and mouse ESC (**Figure S3 G-J**).

2) Another important finding in this study is to demonstrate the retention (or re-establishment) of sperm histone marks in embryos even after DNA replication. If I understood the experimental procedure correctly, Fig. 5B-D are the comparison between “untreated sperm” and “sperm with chromatin assembly and 1X replication in egg extract”. If so, is certain normalization required to compare the HMDs between haploid and diploid chromatin? In addition, according to the Method, authors treated all the frog samples (i.e. sperm, egg extract-treated sperm, and blastula) with the same MNase condition (2.5 U, 37 C, 30 min). It might be too harsh for egg extract-treated sperm and blastula, as their chromatins are somatic-like, thus the MNase treatment can cause over-digestion (and it could be the reason why blastula has only 1/2 of H3K4me3 peaks in Fig. 5F). Validation of proper chromatin extraction and DNA fragment size are needed.

We apologize for the lack of clarity in this section. As mentioned in the M&M and main text we evaluated the presence of peaks of histone modification in blastula embryos by regular ChIP-seq analysis in this case. For that purpose, we performed the IP on chromatin obtained by crosslinking followed by fragmentation through sonication (fragment of ~300 bp are produced). This is stated in the relevant M&M section (**Page 28 line 981** “...embryos were fixed in 2 mL of 1% Formaldehyde...”) and we have now also clarified this point in the main text (*‘For that purpose, we performed ChIP-seq analysis using formaldehyde fixed early blastula embryos’* **Page 8 Line 284-286**).

In the case of egg extract we could evidence that MNase treatment of sperm chromatin following incubation in egg extract in the presence (chromatin assembly but no replication) or absence (chromatin assembly +replication) of geminin produces expected DNA fragments of nucleosomal or di-nucleosomal sizes whereas the starting sperm material exhibits nucleosomal (150bp) and subnucleosomal fragments (**Figure S4 E**). Therefore, MNase treatment of extract treated chromatin produces the expected fragment sizes.

3) In the entire manuscript, peak pattern of H3K27me3 seems broader compared to previous studies by other groups. In Fig. 3E, for instance, only cluster 5 exhibit peak-like enrichment at TSSs, but it’s not obvious in other clusters. Also in Fig. 3E (right), how was the relative enrichment calculated, as if cluster 1, 2, and 3 were compared, cluster 2 possess more H3K27me3 according to the heatmap, but it wasn’t clear in the histogram?

Figure 3E is not reporting peak size but is showing HMD values in 50bp bins +/-2kb around the TSS. The graphs on the left are generated by computing the average HMD value across the genes in the cluster (now indicated in the figure). With this representation Cluster 4 does show peak-like distribution, but has a broader distribution than cluster 5.

To further characterize peaks of H3K27me3 in frog sperm, we plotted the peak sizes from our study (“frog sperm”) and that from others (frog testis , frog embryos (blastula and gastrula), human ESC and testis, mouse testis and sperm), see figure below. We observe that compared to those other organism/cells the *X. Laevis* sperm cells do not behave like an outlier.

And by judging from Fig. 3E, there is no gene with H3K27me3 alone; all the genes were clustered either in H3K4me3 alone or H3K4me3/K27me3 bivalency. Is it really so, or could it be due to the failure of peak calling of H3K27me3?

Please refer to the section above for validation of our peak calling strategy in frog samples. Additionally, we observe a large overlap between the peaks we detected in human sperm and those previously reported in Hammoud *et al.*, Nature 2009, (**Figure S8 D**). This provides an independent validation of our peak detection strategy therefore confirming that the pipeline is robust.

As mentioned above, figure 3E show the distribution of methylation density, not peaks. It indicates that in general methylation of H3K27 around the promoter is broader than it is in the case of H3K4. This correlates with the peaks sizes: H3K27me3 peaks are broader than H3K4me3 peaks.

When considering peaks, we detected a large number of peaks that corresponds to specific H3K4 methylation, or specific H3K27 methylation, and a relatively smaller fraction of peaks that have H3K4me3 together with H3K27me3 (“bivalent”). Please refer to the new “Histone methylation peaks” tab in **Table S3** for the corresponding numbers. We observed that regardless of the set of peaks considered (*i.e.* all peaks or peaks with HMD>80%) there are indeed peaks that have H3K27 methylation only. When considering genes whose +/-2kb interval is associated with peak of H3K4me3 only; H3K27 only, or H3K4me3 together with H3K27me3 (“bivalent”) we also observed that all these categories are well represented (See “genes” tab in **Table S3** for the corresponding numbers).

4) Unlike sperm of model animals, there must be substantial person-person and sperm-sperm heterogeneity in human in terms of morphology and likely other factors, although authors mentioned that they were donated by fertile men, and their sperm passed quality checks, and are purified by Percoll gradient method. This could be one of the causes of quite low % of HMD>80% in human sperm (Fig. 7A) compared with those in frog (Fig. 3AB). Is it possible for authors to provide data how homogeneous their human sperm sample is? What if they analyzed the individual sperm specimen without pooling them?

First, we would like to argue that if, as the reviewer suggests, there must be difference between individual, then the use of frog which is not an inbred specie, is very likely to recapitulate individual to individual variation. So the difference between the extend of high HMD between frog and human would be unlikely to stem from a stronger heterogeneity in between individual from the two species. Additionally, the human sperm ICe-ChIP does not correspond to a pooled sample so the observed low HMD is not the result of averaging between different individuals. We have now clarified this point (**Page 10 Line 340-341**: *'To that end human sperm from a fertile individual was treated with an MNase concentration...'*). Lastly, the overlap between peaks identified by our analysis and that of another lab (**Figure S8 C-D**) suggest that there is homogeneity, between individuals, in the location where methylated histones are retained.

5) Are there differences of H3K4me3 and H3K27me3 HMDs between nucleosome and subnucleosome? If the nucleosomes and subnucleosomes were analyzed separately, what do the Fig. 3A (left) and Fig. 3B look like?

This is a very good point. We compared HMD levels in peaks of methylated H3K4 or H3K27 embedded in nucleosome versus sub-nucleosome and we found a very similar distribution. This result is now included in **Figure S5 E&F**. We conclude that globally the H3 methylation level in sperm is not different whether H3 is embedded within a nucleosome or a subnucleosome.

6) In Fig. 3D, authors listed transcription factor-coding genes related to embryonic development and looked at the enrichment of H3K4me3 and H3K27me3. They also divided the TFs to maternal factor and non-maternal/no evidence-of-origin factor. I could not understand authors' intention to perform this analysis. Is it also related to Fig. S7 (maternal gene regulation)?

In Fig3D we do not report TF genes but instead TF binding sites motifs that are enriched in regions of the genome with high HMD on H3K4 or H3K27. The rationale behind this analysis is to survey the TF that could be controlled by sperm derived methylated histones. We then indicate the instance in which the presence of the corresponding TF in the egg has been documented.

The analysis shown in FigS7 aims at evaluating whether sperm derived methylated histones could regulate maternal transcription factor activity. In this analysis we evaluate whether altering methylated histones that decorate a TF binding site or altering the supply of this maternal TF (in that case *Ascl1*) leads to misregulation of a similar set of genes.

7) Related to Fig. 4, 6, and partly 7, Erkek et al., 2013 previously reported the transgenerational effect of sperm H3K27me3 on gene expression in early embryos, while authors in this study emphasized the implication of

H3K4me3/H3K27me3 bivalency in transgenerational regulation during early embryogenesis including ZGA.

When H3K27me3 alone (if it existed) and H3K4me3/H3K27me3 bivalency are compared, which one is more predominantly retained at developmental genes in this study? Because according to Table S4, H3K27me3 also exhibits significant enrichment in developmental genes, some of which are more significant compared to those in H3K4me3/K27me3 bivalency.

Our previous work comparing sperm/spermatid provided evidence of the involvement of bivalent marking of gene in the male germ cell and embryonic expression (Teperek *et al.*, genome research 2016). In that work we observed that genes that exhibit bivalent configuration in spermatid but H3K27me3 marking only in sperm show abnormally high expression in spermatid- compared to sperm- derived gastrula embryo.

We thank the reviewer for suggesting this analysis of the fate of bivalent *versus* H3K27 only marked genes. We provide this reorganised fate analysis in **Figure 6 C**. Focusing on peaks with HMD>80 we observe that after replication there is a very high retention of H3K4me3 only peaks (~80%) and bivalent peaks (~ 60%) and relatively poor retention of H3K27me3 only peaks (~20%). This strengthen our conclusion that the homogeneously bivalent marking of genes in sperm (particularly present on developmental genes) is involved in the regulation of expression of these genes in embryos.

In addition, according to Fig. 4, it is still unclear to me whether these modified histones are preferentially retained in nucleosomes or subnucleosomes.

As a whole we observe a tendency for modified histones to be better retained during spermiogenesis in chromatin region that homogeneously retain nucleosome or homogeneously acquire subnucleosome (fold enrichment value>1 in Figure 4A&B).

8) By comparison between frog and human data, authors discovered that enrichment of HMDs in developmental genes including ZGA are commonly observed in both species, while no common (= ortholog) genes were detected. Is it because the gene profile composing ZGA is different between frog and human? Please explain.

To look into this aspect we performed an orthology analysis using Inparanoid 8. Frog and human are not closely related species and we found that 4256 genes are orthologs between the two species (1249 frog genes have a human orthologs, and 2169 human genes have a frog ortholog). The number of ZGA genes that are orthologs between frog and human are only 49.

Genes with high HMD are enriched for GO terms related to development in both species. However very few ZGA orthologs have high HMD between the two species. We conclude from this observation that epigenetic programming of developmental genes through sperm methylated histones appear to be a conserved mechanism between frogs and human but that the genes affected are specie-specific. This is not so surprising as the same developmental genes are deployed in different way between different species during early development (for example between mouse and human germ cell specification Irie *et al.*, Cell 2015), and even more likely between more distantly related specie as frog and human.

【Minor concerns】

9) # replication of each ChIP-seq samples were not described in the manuscript.

The GEO file associated with the paper lists all replicates used in this study. If the reviewer deems it necessary we can add another supplemental table separately listing the replicates.

10) In Fig. S1A, western blot of H4 lacks linearity (whereas the one in S1B looks good).

Please find below LicoR measurement for the panel S1A. We observe linearity for the H4 dilution in both panels.

11) Fig. 1E, lower panels (bar graphs). Please indicate error bars if authors performed the experiment more than three times.

Error bars (standard deviation) are now added in figure 1E (n=2).

12) Related to Fig. 3A, authors mentioned that “these sites have histone methylation in most sperm and cover a fraction of the genome similar to that found in an ESC population, suggesting that modified histones might have functional relevance in sperm as is the case for ESC”. What kind of genes are in common between frog sperm and mESCs?

The point we wanted to illustrate here is that the extent of the genome that show high HMD level is similar in frog sperm and ES cell. Since histone methylation have been shown in numerous instances to affect the ES cell gene expression pattern, this suggest that the observed sperm methylation level is compatible with a role in gene expression.

13) In Fig. 4A, fold enrichments are calculated by $HMD > 80 / HMD < 80$, not by $HMD > 80 / total$. I assume $HMD > 80 / HMD < 80$ makes the difference more obvious, but is it really appropriate?

We apologize for the discrepancy between the figure and legend. The data shown in Figure 4A is fold enrichment of $HMD > 80$ peaks over random distribution (now clearly indicated in Figure 4A and B legends). Additional comparison (ratio fold enrichment over all peaks or over peaks of $HMD < 80$) are now provided as a separate

tab (“enrichment values related to fig 4A&B”) in **table S4**.

14) What is the exact property of “None” in Fig. S5, as there must be certain types of histones. Please specify.

“None” corresponds to peaks of histone methylation that occur in region of the genome where there is not retention of a specific H3 particles (*i.e* the HMM analysis did not identify enrichment for either nucleosome or subnucleosome structure at these locations). This is indicated in Figure S5 legend **lines 631-633 page 19**: “*genomic location ... with enrichment for: nucleosome; one type of subnucleosome (70 or 110 bp), a mixture of subnucleosomes (70 and 110 bp), or no particular type of particle (none)*”

15) In line 389-392, authors introduced ref# 13 and 32 after the sentence of “modified histones cannot be the basis for the epigenetic information required for embryo development”. However, at least #32 demonstrated the enrichment of H3K4me3 at developmental genes, and also presented the consistency of their data with Erkek et al., 2013. Thus, putting #32 here seems irrelevant. Additionally, as authors may realize, there is a report demonstrating the reset of paternal H3K27me3 at fertilization in mice (Zheng et al., 2016, PMID: 27635762). Some discussion is highly recommended.

We apologize for this inaccuracy in referencing. We corrected the reference to leave only reference 13 (Carone *et al.*, Dev cell, 2014). We now also comment on report of germ cell derived histone methylation fate in early human embryos (Xia *et al.*, science 2019). **lines 432-433 page 13** “*In human a recent report also indicates that H3K4me3 persists from fertilisation to post-ZGA stage embryos⁴⁸.*”

Reviewer #3:

This paper examines the epigenetic states of mature sperm DNA using omics approaches. The authors suggest that 1) the programming of sperm genes for embryonic development is associated with the formation of chromatin regions of homogeneous epigenetic constitution, and that 2) there is a conservation of sperm epigenetic programming mechanisms between human and frogs. Unfortunately, the paper suffers from a number of issues. While the paper makes some interesting observations, these observations are often not clearly explained and not integrated well in the text. In addition, this paper does not integrate other relevant literatures. Consequently, the paper suffers from the lack of clarity as described below.

We apologise for our lack of clarity and we are very grateful to this reviewer for pointing out the section of our manuscript that needs improvement. We explain below how we have modified the current manuscript based on the reviewer comments:

Ln 66-67: “In *Xenopus laevis*, the deposition of protamines during spermiogenesis is associated with only a partial loss of some of the core histones.” Does this mean that one can have less than an octamer? Or does it mean that the lost core histone is just replaced with a variant histone subunit?

Previous data indicated that only histone H2A and H2B are reduced during *Xenopus* spermiogenesis while H3 and H4 are retained. As the reviewer points out, this implies that either canonical H2A /H2B are replaced by sperm specific H2A/H2B in histone octamer OR that histone H3/H4 package chromatin without H2A/H2B in non-nucleosomal particles. Our mass spectrometry analysis and MNase characterisation of sperm chromatin shows that the latter occurs with the presence of both nucleosomal and sub-nucleosomal H3 containing particles. However, this does not exclude the presence of variant histone in the sperm chromatin and our current mass spectrometry indeed indicates the presence of histone variants (**Table S1** and **dataset PXD012853** on the PRIDE repository).

This is reported in the manuscript **page 3 lines 99-101**: “*This raises the question of how core histones are associated with sperm DNA in X. Laevis. Indeed, the stoichiometry of core histones retained in frog sperm implies that a large fraction of histone H3/H4 cannot be associated with DNA as nucleosomes.*” and **page 4 lines 132-136**: “*These data show that the 110bp and 70bp fragments do not correspond to nucleosomes formed of core histone octamers, but rather correspond to DNA fragments protected from MNase digestion by core histone complexes depleted of H2A and H2B possibly as (H3/H4)₂ tetramers (70bp) or (H3/H4)₂(H2A/H2B) hexamers (110bp) (Figure 1F and ^{20, 21}).*”

Ln 66-67: “This feature places this vertebrate in an intermediate position between the situation found in mammals where the majority of histones are replaced, and zebrafish sperm that do not lose any histones”, citing reference 7. If fish don't lose histones, and have respectably sized genomes, what's the reason

to do this increasingly more as animals move to amphibia and then eventually mammals? Cairns paper (ref 7), describes that H3K27me3 and H3K4me3, among other marks, are found on zebrafish sperm developmental genes. This suggests that the frog story herein might be evolutionarily conserved with zebrafish. Is this correct?

We do not investigate “why” there is an increased tendency to lose histones when considering fish/frogs/mammals. However, we point out that the unique situation in frogs (see point 1 of the reviewer) allowed us to evaluate the retention of methylated histone in a sperm population in a situation where extensive remodelling of nucleosome occurs.

Ln 104-106: “Two additional DNA fragments with a size of ~70 bp and ~110 bp appear specifically after digestion of sperm chromatin.” In Fig1B, XL177 seems to generate DNA fragments corresponding to 110bp and 70bp, albeit weakly. Is it possible that sperm chromatin produce these smaller fragments simply by “overdigestion” of sperm sample, relative to the somatic sample? Would a longer digestion or a higher concentration of MNase produce these fragments from somatic chromatin?

We have four lines of evidence demonstrating that 70bp and 110bp DNA fragments generated by MNase treatment are not the results of overdigestion of nucleosomes:

1. The somatic cell and sperm cell shown in fig 1B where digested using the same MNase conditions so 70 and 110 bp fragments appearance is sperm specific not MNase treatment specific.
2. A nucleosomal ladder spiked in a sperm sample prior to MNase treatment produces only 150 bp fragment (as judged by paired end analysis of the resulting digested DNA) while the xenopus sperm chromatin within the same sample yield fragment of 150, 110 and 70bp.
3. The sucrose gradient analysis of MNase treated sperm chromatin shows that 150 and 110/70bp DNA fragments corresponds to nucleoparticles of different sizes and (Fig1D)
4. The mass spectrometry analysis of protein associated with 150bp fragment identifies all histones of a nucleosome while the protein associated with the 70bp fragment are depleted of histone H2A and H2B (**Figure 1 D and Table S1 and dataset PXD012853 on the PRIDE repository**).

Ln128-131, Figure 1E and Figure S1. It is not clear to me how one can conclude the stoichiometry shown in histograms in Fig1E. The H3/H4 ratio does not equal 1 in ESC extract, for example. Intensity of H4 is significantly higher than that of H3 in the top panel. In the 150bp particles, which are claimed to be “standard” nucleosomes with an H3-H4 tetramer, the ratio seems to be around 0.6. Does this mean it is nearly twice as much H4 as H3?

Quantitative WB analysis does not allow us to measure the number of molecules present in each sample. Rather what is plotted (y-axis: ratio of signal intensity) is the fluorescent signal intensity obtained from the anti-H3 antibody detection to that obtained with the anti H4 antibody. We have now changed the axis legend in fig1E to: "Ratio of signal intensity".

Fig1F suggests a loss of an H2a/b dimer from the standard histone octamer and the data (Fig1E) does not seem to match the model.

The data does fit the model as we see a much lower H2B/H4 ratio in the 70bp particles than in the 150bp particle (WB in **Figure 1E** as well as mass spectrometry analysis (**Table S1** and **dataset PXD012853** on the PRIDE repository).

Additionally, the molecular weights of H4 protein are not identical between purified particles relative and whole extracts. Is this a gel artifact?

Yes indeed, the difference in histone size arise from the presence of sucrose in the samples collected from sucrose centrifugation and absent from the whole sperm extract.

Ln 154-156: "When taking into account the length of DNA protected by each type of particles, this indicates that the majority of the sperm genome (66%) is protected by nucleosomes".

Does this say a majority of genome is protected by these classes in these ratios? Or does it say that, OF THE DNA THAT IS PROTECTED, it's in these ratios? Is the entire genome inside nucleosomes or is some of it is unprotected as it's in between nucleosomes? Please clarify.

The sequencing data represents DNA fragments protected from MNase treatment by nucleoproteic particles (nucleosomes and subnucleosome). Figure 2A left indicates the percentages of fragments 150bp (50.8%), 110bp(13.5%) and 70bp (35.5%) long. Figure 2A left shows the fraction of the genome protected by those types of particles of different sizes. The fragment sizes captured by NGS reflect the sizes observed by gel electrophoresis (**Figure S2F** and **Figure 1B**, respectively).

As shown in **Figure S2 -A&C** (DNA) and **-D** (histones), MNase treatment release a large fraction of DNA and histone from nuclei. This release happens to similar extent in sperm and somatic cell (XL177 and erythrocyte).

Ln 177-178. The statements in the paper like "Interestingly, an enrichment for function related to spermatogenesis is observed for cluster 2" are difficult to verify because the suppl table isn't laid out in a way that makes this obvious. Table S2 contains a list of genes for cluster 2 and another tab with GO terms for cluster 2. How can I determine from this list that spermatogenesis genes are enriched? There is another tab called GO filtered cluster 2. Is this what I should be looking at? There is no explanation provided to the reader. All data suppl files needs to be made more transparent and accessible.

We have now expanded the description of the data in the supplemental table: Specifically to Table S2 the legend reads: "*The table contains for each cluster*"

corresponding to figure 2D: (i) the list of genes in that cluster, (ii) the full GO enrichment analysis, and (iii) a filtered GO term enrichment set (retaining only the GO categories for which a significant p-value is reached in all three statistical tests (classic fischer, Classic KS, and elimKS))”.

Ln 188-190: “Our results so far indicate that some sperm histone H3 containing particles have the required attributes to act as carrier of epigenetic information to the next generation as they are found to be retained at the same genomic location within a sperm cell population.” This statement is a bit unclear. The patterns in Fig2D seem to indicate that large blocks of DNA are held within certain categories of subnucleosomal particles. The phrase "genomic location", in the abstract and throughout the text, gave me the impression that the paper was going to be talking about phased nucleosomes. In other words, nucleosomes that sit on very precise sites relative to gene enhancers/promoters etc. But Fig2D doesn't show this. It seems to show that large blocks/DNA regions are decorated with specific nucleosomal types in various patterns. I think a change in terminology is needed to more precisely convey what is observed and to avoid confusion with a nucleosomal phasing type of model. The depicted figures also do not help clarifying this concept.

Our analysis is not focusing on nucleosome positioning but on nucleosome occupancy. More specifically, we wanted to categorize regions of the genomes with enrichment for the 3 possible chromatin organization patterns: nucleosome, sub-nucleosomes and no defined particle but we did not estimate the exact genomic location of individual nucleosomes.

We thank the reviewer for pointing out the confusion in the way we report this analysis. It is indeed not our intention/findings to identify nucleosomes that are precisely phased around the TSS. Our HMM analysis only identify genomic region where certain type of particles (nucleosome or subnucleosome) are enriched but does not evaluate positioning. To avoid confusion we now state this precisely **Page 35 lines 1219-1221**: “ *It is important to note that the regions identified represent genomic locations where a type of particle is likely to be found but it does not imply specific positioning (i.e. “phasing”) of the particle considered.*”

Ln194-196: I imagine there is some variance in the pipetting and quantifying when mixing sperm chromatin with methylated nucleosomal ladder standard for this quantitation. What is this error range?

We used micropipettes that are calibrated regularly. We have not measured our pipetting error but not expect it to be of concern. Indeed, the normalisation to the spike-in ladder is internal to each sample and rely on measurement of the IP efficiency by quantifying the numbers of fragments mapping to the ladder in the IP and Input samples. As such the absolute amount of ladder material spiked-in the sample is not affecting the normalisation as long as it remains in the range of that of the genome under investigation.

Ln207-209: “..importantly, are enriched for binding motifs recognized by

transcription factors implicated in early embryonic development (i.e. NFY-A/Dux 22; Ascl1 23, ZFP281 24; Figure 3D).” There is no dux gene in frogs. Since frog uses human gene/protein symbols, zfp281 should be znf281. Most of the znf281 study has been reported in mammalian pluripotency, not Xenopus. Since Nfya and Dux belong to completely different transcription factor (TF) DBD families, its more likely that the motif found corresponds to Nfya, or maybe some other HD protein, but not Dux. Additionally, there is little effort to interpret these findings other than to say 3 out of the 26 motifs might represent TFs known to be involved in early development in some vertebrate. Lastly, what is the statistical significance of the motif analysis? I ask this question because motif analyses frequently use a collection of more narrowly defined DNA regions. However, in this analysis, the authors seem to use much broader DNA fragments. What is the evidence that such an analysis can generate a statistically meaningful output?

We apologize for the confusion arising from this panel. For clarity we have not reported all the TF whose predicted binding motifs are enriched in high HMD fraction of the genome. Instead, we have selected a few representative TF to illustrate the various categories identified.

To avoid confusion we have replaced the Dux/NFY A/B label from that figure with NFY A/B, as this better reflect the frog situation.

The rationale behind this analysis is to survey the TF binding sites that could be controlled by sperm derived methylated histones. We then indicate the instance in which the presence of the corresponding TF in the egg has been documented. FigS7 aim at evaluating whether sperm derived methylated histones could regulate maternal transcription factor activity. In this analysis we explored whether altering methylated histones that decorate a TF binding site or altering the supply of this maternal TF (in that case Ascl1) could lead to misregulation of a similar set of genes.

Regarding the motif analysis, we have not used short sequences centered around identified peaks. Instead we have used sequences corresponding to the full length of the identified peaks (average 571pb and 1770 bp for H3K4me3 and H3K27me3, respectively).

Ln 227-231 Fig4A: By looking at histogram for K27me3, it is unclear whether 1.1, 1.05 and 1.0 are statistically significantly different. Please show p values.

The relative p-values are $< 10e-03$. We now provide the fold enrichment values, ratio fold enrichment values and associated p-values in a separate tab (“enrichment values related to fig 4A&B”) of table S4.

Ln 254-258, Table S4, Fig4C. Is cluster 6 the only cluster with developmentally interesting genes? How many genes are in this category? Perhaps the authors can state it in the text. I also don't fully understand the way this table is organized. For instance, while there are 2450 entries in cluster 6, it is unclear how many of these are the same as another entry. For instance, what is the difference between ZNF800_Kwon201107_XENLA_00069622 and ZNF800_Taira201203_XENLA_tissue_00006407? How about ZNF536_Taira201203_XENLA_tissue_00229075 and ZNF536_Taira201203_XENLA_tissue_00250614?

These are likely to be different genes originating from the duplicated genome of *Xenopus laevis* (*Xenopus laevis* is a pseudotetraploid). Although for example the two ZNF800 are ortholog to the same human genes they are likely to have evolved different function in the frogs (many of the duplicated genes have different expression pattern for example, both spatially and temporally).

Fig4C: I don't think the authors referred to this panel in any significant way and I do not know the significance of the panel.

The purpose was to provide some example of HMD distribution and nucl/subnucl enriched region around some genes. We have now removed this panel as it is indeed not essential.

Ln284-286 “For that purpose, we performed ChIP-seq analysis in early blastula embryos (after ~10 embryonic cell cycles but before the activation of zygotic transcription).” In *Xenopus*, 10 cycles is around the middle of stage 9 and transcription has started earlier based on various RNA-seq data. While this does not change the overall conclusion of the paper, correct stages should be reported.

We apologize for this mistake. We used stage 6/7 blastulae stage embryos that are prior to main ZGA and corresponds to x (6-7) cell division (128-256 cells). Corrected values are provided **page 8 lines 284-286** :” *For that purpose, we performed ChIP-seq analysis using formaldehyde fixed early blastula embryos (after ~ 6/7 embryonic cell cycles but before the activation of zygotic transcription) (Figure 5A).*”

Fig5F: 97% of K4me3 blastula peaks (13,161 vs. 422 peaks) are in sperm. However, only 37% of sperm peaks (13,616 out of 36,015 peaks) are still present in blastula. What is the interpretation of this result?

As we point out in the manuscript the peaks that are found in blastulae are significantly enriched for peaks that are homogeneous in sperm (high HMD). Our interpretation is that peaks that are present in most sperm (homogeneous) hold instructive information for embryo development and are maintained during embryos development leading to ZGA while those peaks that are only present in a subset of sperm are not maintained in early embryos.

Ln300-302 and subsequent paragraphs. I'm confused. Fig6A seems to show that developmental genes at ZGA are not bivalent as bivalency is on "all genes" as these are marked as “K27 homogeneous & retained” and “K4 homogeneous& retained”. I also do not fully understand this entire section. In Fig4B the authors state that bivalent is where the developmental genes are, in cluster 6. Then, in paragraph below, on this page, the authors start again talking about ICeChIP finding. I am unable to follow the argument.

We have added a panel detailing the fate of sperm histone methylation peaks (**new panel C in Figure 6**) that links to the gene set enrichment analysis in **figure 6D (former 6C)**.

We hope that this addition, together with change in the text, improves the clarity of our point (the paragraph entitled *“The epigenetically homogeneous fraction of sperm*

chromatin programs the paternal genome for embryonic expression” page 9 lines 295-296 has been extensively modified).

Ln312-313: “Genes that are misregulated upon demethylation of H3K27me3 (Kdm6b sensitive)” How was this experiment done? The experimental approach is not in materials and methods. However, based on reference 2, I assume that the mRNA injections were done in 1-cell stage embryos. If so, the statement that the authors are looking at interference with H3K27 or H3K4 methylation “at fertilization” is not possible as the injections are occurring post fertilization (and also there must be some lag between mRNA injection and build up of translated protein.)

We have modified the analysis related to figure 6. We explain in more details how the experiments in which sperm histone were demethylated at fertilization were carried out **lines 313-315 page 9**. “*In that studies we identified genes that were misregulated in gastrulae generated from egg that expressed a demethylase targeting either H3K4 (Kdm5b) or H3K27 (Kdm6b) prior to sperm injection*”.

Ln317-318: “Genes sensitive to demethylation of H3K27 or H3K4 at fertilization are enriched for such bivalent genes.” The authors say that developmental genes are bivalent, but that's not what's shown in Fig6A.

Figure 6A illustrates whether different categories of ZGA genes (all ZGA genes, ZGA genes involved in development, or ZGA genes with housekeeping function) are enriched for sperm genes that are homogeneously methylated on H3K4, homogeneously methylated on H3K27, heterogeneously methylated on H3K4, or heterogeneously methylated on H3K27. However in this categorization genes homogeneously methylated on H3K4 includes genes that have no H3K27me3 as well as gene homogeneously methylated on H3K27. We investigate bivalent genes enrichment in **figure 6C&D**.

Ln319-320: “...depletion for the maternally provided factor Ascl1 harbour homogeneously methylated H3K27 on the binding site for this transcription factor (Figure S7). This suggests that sperm-provided modified histones might regulate maternal factor activity”. It seems that the authors are implying that Ascl1 is driving H3K27me3 here. And how does this suggest that sperm-provided modified histones regulate Ascl1? Isn't it more likely that the maternal factor is affecting the retention of H3K27me3 on the sperm? I don't fully understand the authors' model and the description is vague.

FigS7 aims to understand whether sperm derived methylated histones could regulate maternal transcription factor activity. In this analysis we evaluate whether altering methylated histones (that decorate a TF binding site) or altering the supply of this maternal TF (in that case Ascl1) leads to misregulation of a similar set of genes. Since there is a significant overlap between genes whose expression is affected by H3K27me3 demethylation at fertilization and Ascl1 depletion it suggests that the expression of these genes is regulated by both of these factors.

Because of space constraint most of the description is in **Figure S7** legend.

Ln 358-359 “However, in both species the set of genes with high H3K27me3 HMD are related to developmental functions.” Previously, the authors stated that there are no conservation and now it is stated, "however in both". Please clarify your idea.

In both species genes with high HMD are enriched for developmental function. However, the genes that homogeneously harbour this mark are different type of developmental genes in frogs and human.

Ln 360: How is znf281 implicated in early development? What system and how? It seems the protein controls pluripotency in mICe and maybe nodal signaling later.

We have now corrected the text to make a more specific statement: “*Additionally, an enrichment of high HMD peaks for ZFP281 binding sites, a transcription factor implicated in early mammalian embryonic development, is found in both human and Xenopus.*” **Lines 371-373 page 11**

Ln 375-376: “We found that high HMD peaks (HMD>80) are indeed more likely to be associated with such unmethylated sperm DNA than low HMD peaks (0<HMD<80).” Are these data statistically significant? Doesn't the comparison of signal strength (Y axis values) in Fig7E right vs. left suggest that all categories of HMD are more highly associated with no meC than regions with meC? It also seems that HMD 0-20 is as enriched in "no meC"/"meC" as the HMD>80 category. The text related to this figure panel is vague.

Please refer to reply to reviewer 1 point 7

Ln378-379: “Additionally, genes with high HMD of H3K27 in sperm are enriched for genes that have a closed TSS configuration”. The statement, “genes with high HMD are enriched for genes”, do you mean “the gene set with high HMD”?

We thank the reviewer for pointing out this error. Indeed we meant gene set, now corrected.

Ln 382-383: “We conclude that in human sperm, histones near some developmental genes are always retained in their methylated form.” No gene numbers were given. How many genes are like this?

This is mentioned in the text (130 genes for H3K4me3 and 320 genes for H3K27me3) **page 11 lines 362-364:** “*Nonetheless, some human sperm genes have peaks of high density of histone H3K4me3 (130 genes) or H3K27me3 (320 genes) around their TSSs*”

Discussion: There is no attempt to explain how the data in this paper fits together with Akkers et al 2009 paper that talked about acquisition of H3K4me3 and K27me3 post ZGA, which doesn't coincide with the current story. What's the relationship between the observations here and their findings? Sperm marks presumably dissipate during cleavage and then re-occur in late blastula. That is also what was reported in the Cairns zebrafish paper cited herein as ref.7. If so, what would be the notable findings of this paper?

The paper by Akkers *et al.*, Dev cell 2009. mostly deal with embryos at gastrulation, which is not comparable with our data focusing on chromatin after one round of replication (extract) or in embryos before zygotic gene (stage 7~256 cells). Other

papers by the Veenstra lab have reported ChIP-seq analysis on earlier embryos (stage 8/9 *i.e.* at ZGA). For example both Vanheeringen *et al.*, Genome research 2014 (see figure 1) and Hontelez *et al.*, Nat Comm 2015 (see for example supplemental figure 3b) reports clear presence of H3K4me3 peaks in stage 8 blastula embryos. We therefore don't see any major difference with our findings.

Our work reveals the epigenetically homogenous fraction of sperm, its behaviour on replication, and its functional link with gene expression in early embryos.

Ln908-909, “ChIP was performed as described previously with the following modifications. Blastula (stage 7) embryos were generated by in vitro fertilization.” The authors stated that they have used embryos after 10 divisions. If this is what they used, stage 7 is incorrect.

As mentioned in response to the reviewer previous point we have now corrected this mistake. We used stage 6/7 blastulae stage embryos that are prior to main ZGA and corresponds to x (6-7) cell division (128-256 cells). Corrected values are provided **page 8 lines 284-286:** “*For that purpose, we performed ChIP-seq analysis using formaldehyde fixed early blastula embryos (after ~ 6/7 embryonic cell cycles but before the activation of zygotic transcription) (Figure 5A).*”

The figure legend is kept to a minimum. This is acceptable only if the main text describes each figure carefully. However, the current description of individual figure panels is insufficient for the reader to fully comprehend these figures.

We have tried to improve this and provide additional necessary information for the reader to follow the study. Unfortunately, limited information in the core of the manuscript is a caveat often found with work using genome wide analyses. However, we tried to report most of the details missing from the main text in supplementary information.

Sometimes error bars and the statistical significance of data are missing, and it is unclear how many times each experiment was repeated.

When missing, we have indicated in each figures the number of biological replicates used as well as error bars when applicable.

Ln 1068-1070. “Data Availability: The *Xenopus laevis* ChIP-seq data reported in this paper will be deposited on GEO once the manuscript is accepted for publication. Accession number will be provided in this section.” I was unable to determine the quality of their data.

Although we did not include references in the core of the manuscript we provided the journal editorial team with access details of both our sequencing (GEO accession GSE125982: <https://www.ncbi.nlm.nih.gov/geo/query/acc.cgi?acc=GSE125982> Enter token upqfqukyltunbox into the box) and proteomic (to review PXD012853 on pride database use **Username:reviewer98222@ebi.ac.uk** and **Password: eiWPAM5Q**) data. We checked that accession number for both of these data are functional. We apologize if the accession details did not reach this reviewer in time.

Reviewers' comments:

Reviewer #1 (Remarks to the Author):

The authors have appropriately revised the manuscript by replying to all of my comments. The revised manuscript should be accepted.

Reviewer #2 (Remarks to the Author):

This is the revision of Oikawa et al., which have tried to quantify the locus-specific % of H3K4- or H3K27-methylated histones in frog and human sperm by ICeChIP-seq. In addition, authors further investigated whether these methylated histones are maintained during early embryogenesis.

I appreciate that authors have made lots of effort to address reviewers' questions and many of the questions were sincerely responded. However, I still have a few major concerns as listed below, some of which are partly the same to my previous criticisms. I apologize if authors did not understand my intention correctly in my previous comments, and I hope they will respond properly.

1) For the originally submitted manuscript, I pointed out the methodological issue of ICeChIP, which occasionally gives >100 % HMD, and asked the authors for proper explanation or validation by non-sequencing method. In response to my criticism, authors made efforts such as increasing the depth of sequencing and excluding the possibility of off-targeting of antibodies. I agree with the authors' claim that apparent HMD>100 % represent only a minor fraction (< 5%) thus it doesn't affect their claim. However, similarities to the previous ESC study (Grzybowski et al., 2015) and human sperm ChIP-seq (Hommod et al., 2009) seems nonmeaningful as the former was a different type of cells and the latter was not about HMD.

2) Unfortunately, the newly added Fig. S2C made me concern about the comprehensiveness of the entire analyses, as less than 70% of H3 was solubilized by their MNase treatment. Thus, additional western blotting with anti-H3K4me3 and -H3K27me3 is required to make sure that not only a small portion of them was subject to sequencing.

3) Regarding the sperm-sperm heterogeneity in human samples, I was simply suggesting authors to show some data of clinical tests such as cytology, motility, DNA fragmentation score, immunostaining of histones, etc. to give us an idea how homogenous (or heterogeneous) their human sperm samples were because unlike inbred animals, there is no guarantee that every single sperm is (relatively) homogeneous and high quality in one individual sample even after percoll purification. Since it could impact the HMD significantly, I suggest to provide these data once again.

Reviewer #3 (Remarks to the Author):

Major criticisms:

This is an interesting paper and eventually should be published. However, the paper remains somewhat dense and some figures and analyses are confusing. Please clarify the following points and improve the quality of the manuscript.

1. What is the evidence that the isolated *Xenopus* sperm is homogeneous? One explanation for seeing both nucleosomes and subnucleosomes may be because their sperm prep contains both mature and immature sperm as they are using minced testes. Some discussion should be included

in the text that this possibility is unlikely and the population analyzed include only mature *Xenopus* sperm.

2. In general, figure legends for main figures are not detailed enough to understand how experiments and analyses were done. Ideally figure legends should include brief but sufficient information to understand the figures without reading the main text.

3. There is an error that needs attention. In response to referee 2, #8, it states that the authors used InParanoid8 to find orthologs between *Xenopus* and human: "4256 genes are orthologs between the two species (1249 frog genes have a human orthologs, and 2169 human genes have a frog ortholog). The number of ZGA genes that are orthologs between frog and human are only 49." Frog genomes have more orthology to human than reported. Xenbase.org, the frog community resource that also does gene annotation, has an orthology table (<http://www.xenbase.org/other/statistics.do>) that shows the current tally with over 15,000 genes that are orthologous between either *X. laevis* (or *tropicalis*) and human. The real number is larger as this doesn't record many-to-one matches. If an ortholog is present in both species it is recorded as one entry in this count. And gene duplications (e.g., multiple frog nodals to one human nodal) is also recorded as one entry. The orthologs can be found in this file:

<http://ftp.xenbase.org/pub/GenePageReports/XenbaseGeneHumanOrthologMapping.txt> from their FTP site. Therefore, the analyses need to be re-done with this much longer list of orthologs.

4. Lines 207-210 (and elsewhere) discussed putative DNA motifs identified for various potential transcription factors that might be interacting with regions of histone modifications. Please show PWM logos for the motifs mentioned in Fig3D/text. It is also not clear how the motif search was conducted. Since the size of the region used for motif finding is important, it is useful to specify the parameters. I was unable to find the information in the text. If the size of the region is too large, it may not be efficient identify motifs. An interesting exercise may be to use different sizes of regions, 200bp, 500bp, 1kb for instance and see whether the identical motifs could be identified using different length setting.

5. Ln 213-216. "Several gene clusters show co-occurrence of high degree of methylation density for both H3K4 and H3K27 (cluster 2,4,5) and are associated with GO categories related to development, especially when H3K4 methylation cover a broader domain around the TSS (cluster 4) (Table S3)." Was GO term analysis done for each different cluster for Figure 3E? If such analysis is done, do you see differences among different clusters? How would you interpret the data and how does this fit with your model?

6. Lines 232-3 says "there is a tendency for homogeneous retention of H3K4me3..." but the tendency, measured as fold enrichments, looks rather weak at best), being in the 1.1-1.2 fold range. It would be better to more explicitly acknowledge this by saying "there is a weak (or slight) tendency... Alternatively, one may argue that that there may be no difference among these samples. How would you deal with such an argument?

7. Why is Znf281 (frog gene names follow human: <https://www.ncbi.nlm.nih.gov/gene/23528> and not Zfp, which is mouse specific) a focus at several points in the manuscript? First, the paper cited, ref 24, shows this gene's involvement much later than at human ZGA. Second, examining PWMs for Znf281 in CIS-BP (<http://cisbp.ccb.utoronto.ca/TFTools.php>, an excellent resource for TFs and their PWMs), shows that the motif is not very complex and mostly just a string of Cs (or Gs). Using the consensus motifs (which admittedly isn't a PWM!) in a Motif Scan on this site finds that E2f6, Glis2 and Mzf1 might also be candidates for this motif. Having the PWMs, or at least logos, for the motifs you found would provide more convincing evidence for a better match to Znf281, but no data is presented currently.

8. Line 272, I really struggled to understand where the number 75%, for retention of sperm H3K4me3 peaks, comes from. 15930/20085 is 79%, but shouldn't the numbers being discussed here be the fraction of peaks inside the intersection of those in the sperm and replicated sperm, which actually would be $20085/(15930+20085) = 56\%$? And the number for H3K27me3 should be $10775/(10775+45074) = 19\%$, not 24%.

9. Lines 288-290 discuss H3K4me3 peak retention between sperm and blastula but there is no mention of the temporal dynamics of H3K27me3. Is this consistent with published claims that this mark is absent in early blastula/cleavage stages of *Xenopus tropicalis* and appears after ZGA? There is much discussion of bivalently marked regions in the text, but then little about the K27me3

marks behavior from fertilization to blastula. These issues should be more explicitly discussed.

10. Figure 5. While boxplots provide general overview, having some browser tracks showing K4me3 and K27me3 with track height around several genes would be useful to see how they differ among different genes. This can be shown in suppl figures.

11. Text referring to FigS7 says "genes sensitive to both demethylation of H3K27 and depletion for the maternally provided factor Ascl1 harbour homogeneously methylated H3K27 on the binding site for this transcription factor." And this is represented in the intersect in S7C. But where is the actual data supporting this claim? How much motif enrichment was obtained to be able to state this? Do you get the same enrichment when using the genes in the outersect in S7C?

12. I understand the motivation to knockdown Ascl1 to examine its role harbouring homogeneously methylated H3K27. However, it is unclear whether proper control experiment was done? How do you know the knock down is effective? How did you validate the efficacy and the specificity of knockdown? Why ZFP281 was not subjected to the same knockdown?

13. Fig6C isn't explained at all. What are we looking at here? What does the scale bar represent? I guessed that it's percent but that's not explicitly stated. What are the differently sized sectors?

14. I'm confused by the data in Fig7E. The peaks with high histone methylation density (HMD), >80, are somewhat more likely to be found within unmethylated DNA regions in all sperm...and there seems to be a gradation as lower levels of HMD show lower % of peaks in these regions. But the 0-20% HMD show as high a level as the >80% HMD! Can you explain the data more explicitly?

15. What kind of data is being used in Fig7F? Is this ATAC-seq data? Description of this section should be improved.

16. FigS2A legend says ~70% of sperm DNA is released in 70-200bp range. Seems from the figure that it's just over 60%.

17. FigS3H and J, why does the H3K27me3 distribution begin at ~100%? In E/F the distribution of the data doesn't appear like this.

18. This referee, in the first round of reviews, asked "what is the difference between ZNF800_Kwon201107_XENLA_00069622 and ZNF800_Taira201203_XENLA_tissue_00006407? How about ZNF536_Taira201203_XENLA_tissue_00229075 and ZNF536_Taira201203_XENLA_tissue_00250614?" Perhaps the question was badly worded. These could be the two different homeologs of each of these genes. But the problem is that these nomenclatures are opaque. How can I find the sequences for ZNF800_Kwon201107_XENLA_00069622 and ZNF800_Taira201203_XENLA_tissue_00006407 to compare them? I google searched for these and found nothing. Are these nomenclatures created by the authors? Are there Genbank IDs associated with these that are found in a supplementary file to this manuscript?

19. Also a point of confusion is the answer to the question about Ln284-286 on what stage of embryos were used for ChIP-seq. The answer given is stage 6/7, which corresponds to cleavages 6-7. According to Nieuwkoop and Faber staging, stage 6 is 32-cell stage, and stage 6.5 is 64-cell stage (see drawings). So 32-cell is created by cleavage 5, 64-cell by cleavage 6. Anyway, please make certain that you know and report in the paper the correct stage that was used for these experiments as this explanation gives a mixed message that doesn't precisely answer it.

Minor criticisms:

1. The L in laevis should be lower case throughout as it's the species name.
2. Figure 1 panel E, "H3/H4 ratio of signal intensity." Font sizes are not the same.
3. Line 166 says "but happens more homogeneously." The word homogeneous means thoroughly mixed, lacking in heterogeneity. It's all or none. So something can't be "more" homogeneous than something else. It can be less heterogeneous than something else. So a population can be said to be on a spectrum of more or less heterogeneous, or at the extreme end and be homogeneous (100% identical.) Please check whether this is the only place in the manuscript where such a statement needs to be repaired, as there may be many places where something should read more or less heterogeneous instead of homogeneous.
4. Also on line 166 it says parenthetically "17% versus 9%, Figure 2B." Shouldn't this be 19% (=17+2) versus 9%? If you are only considering 70bp subnucleosome and not 110bp subnucleosome, please indicate the reason why.

5. Paragraph starting on line 170 refers to regions +/- 2kb around TSSs as being promoters. Promoters are much smaller than that and such regions can very often also contain promoter proximal enhancers. So perhaps it would be better to not dogmatically call these genomic intervals as "promoters." Maybe "promoter proximal" would be more accurate while still conveying the points being made.
6. Line 967 says semi-synthetic nucleosomes were spiked in and ref 15 cited. Please provide more explanation here. At the very least explain where you bought the reagents used.
7. In the previous review this referee attempted to help clarify a sentence (Ln378-379, but this actually also exists elsewhere in the paper) where it says "genes with high HMD are enriched for genes" and we asked whether instead of saying genes are enriched for genes, it should state "gene sets with X are enriched for genes." The response says that this was corrected, but the same language exists in the current version so it wasn't corrected. Please carefully go over the manuscript and verify your statement.

REPLY to reviewer's comments

manuscript “Epigenetic homogeneity in histone methylation underlies sperm programming for embryonic transcription“

We would like to thank the reviewers for the constructive comments they provide after our revision of the paper. The points that were not properly addressed or arose during revision have been addressed as follow:

Reviewer #1 (Remarks to the Author):

The authors have appropriately revised the manuscript by replying to all of my comments. The revised manuscript should be accepted.

We take note that the two main points initially raised by reviewers 1&2, about antibodies specificity and the range of HMD, have been properly addressed.

Reviewer #2 (Remarks to the Author):

This is the revision of Oikawa et al., which have tried to quantify the locus-specific % of H3K4- or H3K27-methylated histones in frog and human sperm by ICeChIP-seq. In addition, authors further investigated whether these methylated histones are maintained during early embryogenesis.

I appreciate that authors have made lots of effort to address reviewers' questions and many of the questions were sincerely responded. However, I still have a few major concerns as listed below, some of which are partly the same to my previous criticisms. I apologize if authors did not understand my intention correctly in my previous comments, and I hope they will respond properly.

1) For the originally submitted manuscript, I pointed out the methodological issue of ICeChIP, which occasionally gives >100 % HMD, and asked the authors for proper explanation or validation by non-sequencing method. In response to my criticism, authors made efforts such as increasing the depth of sequencing and excluding the possibility of off-targeting of antibodies. I agree with the authors' claim that apparent HMD > 100 % represent only a minor fraction (< 5%) thus it doesn't affect their claim.

We take note that the two main points initially raised by reviewers 1&2, about antibodies specificity and the range of HMD, have been properly addressed

However, similarities to the previous ESC study (Grzybowski et al., 2015) and human sperm ChIP-seq (Hammoud et al., 2009) seems nonmeaningful as the former was a different type of cells and the latter was not about HMD.

The points of these comparisons are as follow:

- 1- To assess antibody specificity we compared the peaks of methylated histone found in human sperm by regular ChIP (Hammoud et al., 2009) to the peaks found in Xenopus sperm in our ICe-ChIP dataset. In that case we do not take**

into account HMD levels and we process our data as regular ChIP. The large number of peaks overlapping between the two studies is used as a proxy of the robustness of our enrichments and confirm reproducibility of the data despite differences in protocols and antibodies used in these two studies. Hence it alleviates concerns about antibodies specificity.

- 2- To get a view of the range of measured values of HMD shown to be biologically relevant (Grzybowski et al., 2015), we compare our HMD range – measured on *Xenopus* sperm - to the one reported for mESC (Grzybowski et al., 2015). The fact that the range of HMD is similar between these two very different cell types and species indicates that the range of HMD described in *Xenopus* sperm is within the range of that of other biological samples. Since H3K4 and H3K27 methylation have been shown to be important for stem cell function, we conclude that the range of HMD observed in *Xenopus* sperm is compatible with a functional role in these latter cells.

As such we believe that these comparisons are meaningful and bring useful information to the reader.

2) Unfortunately, the newly added Fig. S2C made me concern about the comprehensiveness of the entire analyses, as less than 70% of H3 was solubilized by their MNase treatment. Thus, additional western blotting with anti-H3K4me3 and -H3K27me3 is required to make sure that not only a small portion of them was subject to sequencing.

We performed the WB analysis on the pellets and supernatant generated by MNase treatment. This data is now added as Fig S2-G&H. We found that most of the modified histones are found in the supernatant and therefore included in the chromatin immunoprecipitation analysis (Fig S2G). Additionally, the ratio of H3K4me3 or H3K27me3 to H4 is similar in the supernatant and pellet fraction (Fig S2G). This indicates that the partition of chromatin between the two fractions obtained by MNase treatment is not overtly biased toward any particular H3K4me3 or H3K27me3 content.

3) Regarding the sperm-sperm heterogeneity in human samples, I was simply suggesting authors to show some data of clinical tests such as cytology, motility, DNA fragmentation score, immunostaining of histones, etc. to give us an idea how homogenous (or heterogeneous) their human sperm samples were because unlike inbred animals, there is no guarantee that every single sperm is (relatively) homogeneous and high quality in one individual sample even after percoll purification. Since it could impact the HMD significantly, I suggest to provide these data once again.

The method used for sperm gradient purification is standard in the clinical setting, and in accordance with the WHO guidelines (sperm density, total number, motility, morphology and semen volume). In particular microscopical observation indicated a normal morphology and a high motility as indicated in the M&M section. As requested we have now added the recorded parameters : “The recorded parameters for sperm preparations used in this study were within the following range: Progressive motility (PR) = 74-81%; Non-progressive motility (NP) = 10-13%; Immotile (IM) = 9-11%; Total motility (PR + NP) =87-91%.”page 25 lines 866-868 of the M&M section.

Reviewer #3 (Remarks to the Author):

Major criticisms:

This is an interesting paper and eventually should be published. However, the paper remains somewhat dense and some figures and analyses are confusing. Please clarify the following points and improve the quality of the manuscript.

1. What is the evidence that the isolated Xenopus sperm is homogeneous? One explanation for seeing both nucleosomes and subnucleosomes may be because their sperm prep contains both mature and immature sperm as they are using minced testes. Some discussion should be included in the text that this possibility is unlikely and the population analyzed include only mature Xenopus sperm.

We do not use crude minced testis but instead used purified sperm preparation that were obtained by centrifugation on a discontinuous gradient, as described in our previous paper (Teperek et al, Genome Research, 2016) and described in details in M&M section page 24 lines 826-851. Sperm preparation purity was assessed by microscopical observation and typically yielded >99% sperm (see example below from fig S1 from Teperek *et al.*, Genome Research, 2016, purified sperm (A) and spermatid (B))

We are confident that our analysis does not reflect contamination with non mature sperm. Indeed, our analysis of nucleosome and subnucleosome distribution identified genomic locations where most cells in the preparation had either nucleosome or subnucleosome (*i.e.* figure 2D). This indicates that the nucleosome and subnucleosome signal is not emanating from a minor fraction of the cell population analysed.

2. In general, figure legends for main figures are not detailed enough to understand how experiments and analyses were done. Ideally figures legends should include brief but sufficient information to understand the figures without reading the main text.

We have made adjustments to most of the figure legends.

*3. There is an error that needs attention. In response to referee 2, #8, it states that the authors used InParanoid8 to find orthologs between Xenopus and human: “4256 genes are orthologs between the two species (1249 frog genes have a human orthologs, and 2169 human genes have a frog ortholog). The number of ZGA genes that are orthologs between frog and human are only 49.” Frog genomes have more orthology to human than reported. Xenbase.org, the frog community resource that also does gene annotation, has an orthology table (<http://www.xenbase.org/other/statistics.do>) that shows the current tally with over 15,000 genes that are orthologous between either *X. laevis* (or *tropicalis*) and human. The real number is larger as this doesn't record many-to-one matches. If an ortholog is present in both species it is recorded as one entry in this count. And gene duplications (e.g., multiple frog nodals to one human nodal) is also recorded as one entry. The orthologs can be found in this file:*

<http://ftp.xenbase.org/pub/GenePageReports/XenbaseGeneHumanOrthologMapping.txt> from their FTP site. Therefore, the analyses need to be re-done with this much longer list of orthologs.

The difference in orthologs number in our study and that on Xenbase is not an error but is instead likely the result from different threshold used for orthologs identification. Indeed in our analysis we set a very stringent threshold on the inparalog score because we wanted to include only high-confidence orthologs between the two species (see page 39 lines 1350-1353: “In order to look at conservation between frog and human, we ran inparanoid⁴⁶ using the peptide sequences of each of the two organisms. We processed the output of the orthology mapping performed by inparanoid and we extracted orthologs as pairs of proteins in groups with high-score mapping (score ≥ 0.95)”).

The number we reported is based on an inparalog score = 0.95 (inparalog score definition is in doi:10.1093/nar/gki107 see figure 1 caption and explained in the faq page <http://inparanoid.sbc.su.se/cgi-bin/faq.cgi> ; this score is based on reciprocal blast scores). If we don't apply any specific threshold, we observed 13941 ortholog groups resulting in 14072 unique human gene ensemble ids, a number in line with what is reported on Xenbase. With this extended set of orthologs, out of 2864 human ZGA genes, 1483 have a xenopus ortholog.

With this larger ortholog set we reach the same conclusion as with our former set (see table below): When “all peaks” are considered there is a large overlap of genes marked by H3K4me3 and H3K27me3 in human and xenopus sperm. . We also find that with this new ortholog set, the genes that are HMD >80 in both species are less abundant (28.46% versus 69.12% for H3K4me3 and 5% versus 56.21% for H3K27me3). We believed that the set obtained with our stringent threshold provided high confidence orthologs and we therefore suggest to use it in our analysis. We are open to comment/suggestion by the reviewer

Extended set (new analysis)

		All peaks	HMD ≥ 80	
	H3K4me3	69.12 %	18.46%	
	H327me3	56.21 %	5%	

Stringent set (in manuscript)

		All peaks	HMD ≥ 80	
	H3K4me3	72.1 %	32%	
	H3K27me3	58.8%	8%	

We have carried out& added additional analysis related to transcription factor binding motifs enriched in sperm genomic regions with homogeneous methylated histone marking. Answer to reviewer 3 points (4), (7), (11), (12) related to this topic are grouped below.

4. Lines 207-210 (and elsewhere) discussed putative DNA motifs identified for various potential transcription factors that might be interacting with regions of histone modifications. Please show PWM logos for the motifs mentioned in Fig3D/text.

The logos are shown below:

It is also not clear how the motif search was conducted. Since the size of the region used for motif finding is important, it is useful to specify the parameters. I was unable to find the information in the text.

Search was conducted using default parameters: classic mode, DNA as sequence alphabet, Eukaryote DNA and vertebrate (in vivo and in vitro). This information is now added to M&M page 38 lines 1312-1316: “Transcription factor binding motif enrichment As input sequences we used the genomic intervals corresponding to identified peaks of homogeneous histone methylation (HMD>80). The search was conducted using default parameters: classic mode, DNA as sequence alphabet, Eukaryote DNA and vertebrate (in vivo and in vitro).”

If the size of the region is too large, it may not be efficient identify motifs. An interesting exercise may be to use different sizes of regions, 200bp, 500bp, 1kb for instance and see whether the identical motifs could be identified using different length setting.

We understand the concerns raised by the reviewer given that MEME-ChIP is a transcription factor binding motif analysis tool designed and optimized for short sequences, like is the case in TF ChIP-seq. However, unlike in TF ChIP the central part of the modified histone ChIP-seq peak is not necessarily expected to be centred on a TF binding site. For that reason it would be very difficult to extract biological information from such analysis restricted to the central part of methylated histone peaks. Instead, to further explore some of TF highlighted in our MEME-ChIP we used their relative motifs logo and looked for individual occurrences in our lists of peaks. We used FIMO, another tool from the MEME suite. This tool can work on sequences of different lengths without the constraints and assumptions of MEME-ChIP and performs an explicit counting of motifs occurrence, therefore providing an easier way to interpret

output. Finally, we processed the output of the FIMO search by assessing statistical differences in the frequency of motif occurrence in peaks.

We performed this analysis for all the 4 experimental conditions under study and the result is summarized in the following heatmap. Each of the columns in the heatmap represents one specific experimental case. The selected motifs are reported on the row. Additionally, row names contain also the link to the binding motif (Jaspar database). As expected, this heatmap reports a different and orthogonal type of information than what we reported in Figure 3D. It is interesting to note that the fraction of peaks positive for the TF motifs is generally maintained or increased upon egg extract treatment. This also suggests retained regions are those that contain sequences potentially targeted by TFs important for development.

7. Why is *Znf281* (frog gene names follow human: <https://www.ncbi.nlm.nih.gov/gene/23528> and not *Zfp*, which is mouse specific) a focus at several points in the manuscript? First, the paper cited, ref 24, shows this gene's involvement much later than at human ZGA.

Thanks for pointing that out, we replaced *Znf* by *ZFP* page 6 line 211. We point out *ZNF281* because this TF motif is common to high HMD region in both xenopus and human but we did not intend to put special emphasis on this point.

Reference 24 shows expression of *Zfp281* in the inner cell mass of embryo and an effect of *Zfp281* KO on early post-implantation embryos. We refer to this paper to back up our statement page 6 line 210: “transcription factors implicated in early embryonic development”, and we feel that this is not implying ZGA involvement and therefore appropriate.

Second, examining PWMs for *Znf281* in CIS-BP (<http://cisbp.cabr.utoronto.ca/TFTools.php>, an excellent resource for TFs and their PWMs), shows that the motif is not very complex and mostly just a string of Cs (or Gs). Using the consensus motifs (which admittedly isn't a PWM!) in a Motif Scan on this site finds that *E2f6*, *Glis2* and *Mzf1* might also be candidates for this motif. Having the PWMs, or at least logos, for the motifs you found would provide more convincing evidence for a better match to *Znf281*, but no data is presented currently.

We have provided the logo for the ZNF281 motif enriched in xenopus sperm homogeneously methylated regions (new figure 3D) and shown below:

The logos for the ZNF281 enriched in human sperm homogeneously methylated regions:

11. Text referring to FigS7 says “genes sensitive to both demethylation of H3K27 and depletion for the maternally provided factor *Ascl1* harbour homogeneously methylated H3K27 on the binding site for this transcription factor.” And this is represented in the intersect in S7C. But where is the actual data supporting this claim? How much motif enrichment was obtained to be able to state this? Do you get the same enrichment when using the genes in the outersect in S7C?

We have modified Figure S7 to describe better the statistical test supporting our statement. We also included further comparisons as suggested.

We used three set of genes obtained from the comparison shown in Figure S7 Venn diagram: genes DE specifically in *ascl1* KD, genes DE specifically in Kdm6 OE, genes DE in both conditions (intersect).

Using a proportion test, we then ask if there is an enrichment in these sets for *Ascl1* binding motifs when compared to the set of genes that we identified as homogeneously marked by H3K27me3 in sperm (HMD>80). All three sets show enrichment for *Ascl1* binding motif compared to the set of gene homogeneously marked by H3K27me3. We now report p-value related to all three gene sets as a table in figure S7.

	Proportion with Ascl1 motif	Prop.test pval (chi2)	Prop.test pval
(GW) Genes with H3K27me3 HMD>80	12.96%		
(A) Genes DE in Ascl1 KD only and HMD>80	22.62%	41.63	5.514e-11
(B) Genes DE in Ascl1 KD AND Kdm6OE and HMD>80	25.89%	15.38	4.393e-05
(C) Genes DE in Kdm6OE only and HMD>80	39.01%	257	< 2.2e-16

Note: proportion test compares each individual case (A, B, C) to the genome wide proportion (GW).

We refer to this enrichment page 9 lines 318-321: “Additionally, a significant proportion of genes sensitive to both demethylation of H3K27 and depletion for the maternally provided factor Ascl1 harbour homogeneously methylated H3K27 and the binding motif for this transcription factor (Figure S7).” Underlined words are changes from initial text.

12. I understand the motivation to knockdown Ascl1 to examine its role harbouring homogeneously methylated H3K27. However, it is unclear whether proper control experiment was done? How do you know the knock down is effective? How did you validate the efficacy and the specificity of knockdown? Why ZFP281 was not subjected to the same knockdown?

This data is from a published paper (Gao et al., development, 2016) as mentioned in the supplemental figure S7 legend page 19 line 674. The validation for the knock down is described in Fig.2A-C and Fig.S2A,B of this paper. This is however beyond the scope of this paper to test whether the epigenetic make up of sperm is affecting the activity of maternally provided transcription factor. The analysis of this existing Ascl Knock down dataset was motivated by the possibility of evaluating whether evidence could be found that this might be the case. Further work will be needed to formally test this interesting possibility. Our text indicates that while such observations are in agreement with this hypothesis this is not a formal demonstration: “This suggests that sperm-provided modified histones might regulate maternal factor activity. “(page 9 lines 321-322)

5. Ln 213-216. “Several gene clusters show co-occurrence of high degree of methylation density for both H3K4 and H3K27 (cluster 2,4,5) and are associated with GO categories related to development, especially when H3K4 methylation cover a broader domain around the TSS (cluster 4) (Table S3).” Was GO term analysis done for each different cluster for Figure 3E? If such analysis is done, do you see differences among different clusters? How would you interpret the data and how does this fit with your model?

The GO analysis is indeed carried out for each cluster. This is indicated in table S3 legend page 21 line 716-719: “The table contains for each cluster corresponding to figure 3E: (i) the list of genes in that cluster, (ii) the full GO enrichment analysis, and (iii) a filtered GO enrichment analysis (retaining only the GO categories for which a significant p-value is reached in all three statistical tests (classic fischer, Classic KS, and elimKS))”.

One feature of this GO analysis is that clusters showing co-occurrence of H3K4 and H3K27 methylation are associated with genes involved in developmental function, which fit with our latter observation that sperm bivalent genes are involved in programming of embryonic gene expression.

6. Lines 232-3 says “there is a tendency for homogeneous retention of H3K4me3...” but the tendency, measured as fold enrichments, looks rather weak at best), being in the 1.1-1.2 fold range. It would be better to more explicitly acknowledge this by saying “there is a weak (or slight) tendency... Alternatively, one may argue that that there may be no difference among these samples. How would you deal with such an argument?

We agree with the referee that the enrichment is modest. As suggested, we have corrected the text accordingly page 7 lines 233-234: “In particular there is a tendency, albeit weak, for homogeneous retention of H3K4me3 in the context of a nucleosome.” We also point out that the fate of highly methylated histone is not generally affected by the type of particle within which histone H3 is present (figure S5). Although, as shown in figure S5 there is a weak tendency for better retention of highly methylated H3K4me3 when in the context of nucleosome.

8. Line 272, I really struggled to understand where the number 75%, for retention of sperm H3K4me3 peaks, comes from. 15930/20085 is 79%, but shouldn't the numbers being discussed here be the fraction of peaks inside the intersection of those in the sperm and replicated sperm, which actually would be $20085/(15930+20085) = 56\%$? And the number for H3K27me3 should be $10775/(10775+45074) = 19\%$, not 24%.

We apologize for the confusion. The percentage reported in our manuscript are correct (“...75% of sperm H3K4me3 peaks and 24% of H3K27me3 peaks are retained after egg extract treatment”). When considering the peaks overlap between sperm and replicated sperm there are cases where one peak in one condition overlaps with two peaks in the other condition (this can be seen on the peaks shown in reply to point 10 from this reviewer). For that reason the number of peak overlapping is different depending on the direction of change considered: the number of “sperm peaks retained after replication” is not equal to the number of “replicated sperm peaks that also had a peak in sperm”. We wrongly reported the later case in the venn diagram (figure 5B) when the percentage reported in the text corresponds to the former. The peak overlap in the two scenario are shown below (in the new figure 5 we have replaced the number from the original venn diagram (here shown on right) by the appropriate version (here on left)).

27300/(8720+27300) > 75.8% Sperm peaks are retained after replication (left Venn diagram)

20085/(15930+20085) > 55.7% Replicated peaks that were present in sperm (right Venn diagram)

9. Lines 288-290 discuss H3K4me3 peak retention between sperm and blastula but there is no mention of the temporal dynamics of H3K27me3. Is this consistent with published claims that this mark is absent in early blastula/cleavage stages of *Xenopus tropicalis* and appears after ZGA? There is much discussion of bivalently marked regions in the text, but then little about the K27me3 marks behavior from fertilization to blastula. These issues should be more explicitly discussed.

We have added a section to discuss H3K27me3 temporal dynamic during embryogenesis page 13, lines 431-440:

“In the case of *Xenopus laevis* sperm a direct transmission mechanism of H3K27me3 is also unlikely. Indeed we observe that a much larger fraction of H3K27 methylation than that of H3K4 methylation is lost after replication (Figure 5B), in agreement with H3K27me3 ChIP-seq data obtained in *Xenopus tropicalis* that detected very low level of this mark in gastrulae⁵¹. We hypothesize that a relay mechanism (placeholder) or other histone marks associated with H3K27me3 carry the info into the embryo. Indeed recent analysis suggest a central role for H2Aub rather than H3K27me3 in polycomb mediated gene repression^{52,53}. Such histone modification could be the basis for the transmission of polycomb related epigenetic cue to the embryos.”

10. Figure 5. While boxplots provide general overview, having some browser tracks showing K4me3 and K27me3 with track height around several genes would be useful to see how they different among different genes. This can be shown in suppl figures.

As requested we have added supplemental data showing track height in two genomic regions. This is included as a new panel (A) in figure S5.

(A) Browser tracks around Pax2 and SP2 showing H3K4me3 and H3K27me3 signal in sperm and in replicated sperm. Boxes below the track indicate the position of identified peaks. The dashed rectangles highlight regions where peaks are lost after replication in extract.

13. Fig6C isn't explained at all. What are we looking at here? What does the scale bar represent? I guessed that it's percent but that's not explicitly stated. What are the differently sized sectors?

We have modified the figure 6C legend page 16 lines 540-545: “Pie chart indicating the percentage of the homogeneously methylated histone present in sperm that are retained after egg extract-mediated replication. Three type of sperm peaks are considered: homogeneous for H3K27me3 only, homogeneous for H3K4me3 only, and homogeneous for H3K27me3 and H3K4me3 (bivalent). The area of the coloured sector (together with the colour) indicates the fraction of the overlap between each pair-wise comparison”.

14. I'm confused by the data in Fig7E. The peaks with high histone methylation density (HMD), >80, are somewhat more likely to be found within unmethylated DNA regions in all sperm...and there seems to be a gradation as lower levels of HMD show lower % of peaks in these regions. But the 0-20% HMD show as high a level as the >80% HMD! Can you explain the data more explicitly?

This is indeed the case that a similar fraction of high (>80) and low (<20) HMD fragments are associated with unmethylated DNA (~70%). We observe a significant enrichment for HMD>80 peaks with region of unmethylated DNA when compared to peak in the HMD 20-80 range but not when comparing to those in the low range (HMD 0-20)(See p-value graph below). One explanation for this phenomenon could be that we are focusing on low level of H3K4 or H3K27 methylation but are blind to other chromatin modifications that might drive the exclusion of DNA methylation.

15. What kind of data is being used in Fig7F? Is this ATAC-seq data? Description of this section should be improved.

The data used is from Li, L. et al. Nat Cell Biol, 2018, as indicated in the manuscript and indeed corresponds to ATAC-seq data.

We now mention that the closed/open configuration was determined by ATAC-seq analysis page 11 lines 392-394 in the main text: “Additionally, the set of genes with high HMD on H3K27 in sperm is enriched for genes that have a closed TSS configuration in most cells of the human embryos undergoing ZGA, as shown by ATAC-seq³⁰ (Figure 7F).”

and page 16 lines 557-561 in the figure legend:

“(F) Barplots indicating -log10(pvalue) for enrichment of sperm TSSs with HMD>80 for H3K4 or H3K27 in set of genes with TSSs showing different chromatin accessibility level in 8 cell embryos (open& closed corresponds to TSSs open or closed in all cells of a 8 cell embryos, whereas divergent corresponds to TSSs either open or closed in different cells (ATAC-seq data from³⁰)).”

16. FigS2A legend says ~70% of sperm DNA is released in 70-200bp range. Seems from the figure that it's just over 60%.

We have corrected the legend page 17 lines 582-583:

“After nuclease digestion, over 60% of the sperm DNA is released as fragments in the 70-200 bp size range”

17. FigS3H and J, why does the H3K27me3 distribution begin at ~100%? In E/F the distribution of the data doesn't appear like this.

We thank the reviewer for spotting the inconsistency in FigS3H: indeed there was a bug in the visualization and data points outside of the y-axis range had been cut out from the plot. We have now corrected this.

18. This referee, in the first round of reviews, asked “what is the difference between ZNF800_Kwon201107_XENLA_00069622 and ZNF800_Taira201203_XENLA_tissue_00006407? How about ZNF536_Taira201203_XENLA_tissue_00229075 and ZNF536_Taira201203_XENLA_tissue_00250614?” Perhaps the question was badly worded. These could be the two different homeologs of each of these genes. But the problem is that these nomenclatures are opaque. How can I find the sequences for ZNF800_Kwon201107_XENLA_00069622 and ZNF800_Taira201203_XENLA_tissue_00006407 to compare them? I google searched for these and found nothing. Are these nomenclatures created by the authors? Are there Genbank IDs associated with these that are found in a supplementary file to this manuscript?

We apologize for our inappropriate answer. We hope that the file now included as Supplementary file 1 answers the question. This file enables search for the nucleotide sequences corresponding to e.g. “ZNF800_Kwon201107_XENLA_00069622”.

Supplementary file 1: FASTA file listing all xenopus transcripts used in this analysis

19. Also a point of confusion is the answer to the question about Ln284-286 on what stage of embryos were used for ChIP-seq. The answer given is stage 6/7, which corresponds to cleavages 6-7. According to Nieuwkoop and Faber staging, stage 6 is 32-cell stage, and stage 6.5 is 64-cell stage (see drawings). So 32-cell is created by cleavage 5, 64-cell by cleavage 6. Anyway, please make certain that you know and report in the paper the correct stage that was used for these experiments as this explanation gives a mixed message that doesn't precisely answer it.

As mentioned in the paper, we have used embryos that are prior to MBT. We selected stage 7 embryos based on microscopical observation. Based on reviewer comment we checked recent papers referring to the number of cells present in embryos at this particular stage. Based on the following paper [https://www.cell.com/current-biology/pdfExtended/S0960-9822\(14\)01360-8](https://www.cell.com/current-biology/pdfExtended/S0960-9822(14)01360-8) (i.e figure 1C) a Nieuwkoop and Faber stage 7 corresponds to 8 cleavage i.e. 256 cells.

We therefore propose to use the following:

-In the main text: "For that purpose, we performed ChIP-seq analysis using formaldehyde fixed early blastula embryos (after ~ 8 embryonic cell cycles but before the activation of zygotic transcription)" **page 8 lines 285-287 so that the actual cell number are available to the non-specialist reader.**

-In the M&M section we indicate Nieuwkoop and Faber staging so that researcher from the Xenopus community have the information about the stage selected. "ChIP was performed as described previously^{38,39} with the following modifications. Blastula (stage 7) embryos were generated by in vitro fertilization" page 29 lines 1012-1013.

-In Figure 5A: we indicate that embryos were collected after ~8 replication cycles.

Minor criticisms:

1. The *L* in *laevis* should be lower case throughout as it's the species name

This has been corrected throughout the manuscript.

2. Figure 1 panel E, "H3/H4 ratio of signal intensity." Font sizes are not the same.

The font sizes on y-axis were changed to that used in figure 1A.

3. Line 166 says "but happens more homogeneously." The word homogeneous means thoroughly mixed, lacking in heterogeneity. It's all or none. So something can't be "more" homogeneous than something else. It can be less heterogeneous than something else. So a population can be said to be on a spectrum of more or less heterogeneous, or at the extreme end and be homogeneous (100% identical.) Please check whether this is the only place in the manuscript where such a statement needs to be repaired, as there may be many places where something should read more or less heterogeneous instead of homogeneous.

Thanks for pointing this misuse of "homogeneous". We have replaced our former "more homogeneous" statements by "less heterogeneous" page 5 line 166; page 7 line 250; and page 8 line 276.

4. Also on line 166 it says parenthetically "17% versus 9%, Figure 2B." Shouldn't this be 19% (=17+2) versus 9%? If you are only considering 70bp subnucleosome and not 110bp 19 We are indeed considering both the 70 and 110 bp nucleosomes and have therefore added (=17+2) page 5 line 166 to make this point clear.

5. Paragraph starting on line 170 refers to regions +/- 2kb around TSSs as being promoters. Promoters are much smaller than that and such regions can very often also contain promoter proximal enhancers. So perhaps it would be better to not dogmatically call these genomic intervals as “promoters.” Maybe “promoter proximal” would be more accurate while still conveying the points being made.

Since there is no consensus on how to define promoters genome-wide, we decided to state our definition on the first use of the word: “the region surrounding their TSS +/- 2 kb (thereafter named promoter)” page 5 line 172.

6. Line 967 says semi-synthetic nucleosomes were spiked in and ref 15 cited. Please provide more explanation here. At the very least explain where you bought the reagents used.

We have added additional details on the way spiked-in nucleosomes were obtained and used, page 28&29 lines 994-1001:

“We used the semisynthetic standards used in Grzybowski *et al.*, Mol Cell, 2015 and provided by the Ruthenburg’s lab. We spiked in the standards aiming for the second lowest concentration of the barcoded nucleosomes to be at the same concentration as the genome count (to that end we simply multiply the amount of the nuclei in the sample with the number of genome copies per nucleus (2.5 for dividing diploid; 2 for stationary diploid; 1 for haploid). We then add the amount of the ladder equivalent to the member representing the 5% or 10% of the ladder so that some standards will be below and above the expected genome coverage).”

7. In the previous review this referee attempted to help clarify a sentence (Ln378-379, but this actually also exists elsewhere in the paper) where it says “genes with high HMD are enriched for genes” and we asked whether instead of saying genes are enriched for genes, it should state “gene sets with X are enriched for genes.” The response says that this was corrected, but the same language exists in the current version so it wasn’t corrected. Please carefully go over the manuscript and verify your statement.

Sorry for this mistake. We have corrected genes to gene sets in the following instances: Page 9 lines 302&316; page 10 line 323; page 11 lines 368&392.

REVIEWERS' COMMENTS:

Reviewer #2 (Remarks to the Author):

I appreciate that the authors patiently took care of the questions I raised. Although the accuracy and reliability of ICeChIP are still needed to be verified elsewhere, the revised manuscript is now worthy to be published in Nature Communications.

Reviewer #3 (Remarks to the Author):

Most of the points raised by this reviewer has been addressed satisfactorily and the paper is acceptable.